



# Data assimilation for continuous global assessment of severe conditions over terrestrial surfaces

Clément Albergel[1], Yongjun Zheng[1], Bertrand Bonan[1], Emanuel Dutra[2], Nemesio Rodríguez-Fernández[3], Simon Munier[1], Clara Draper[4], Patricia de Rosnay[5], Joaquin Muñoz-Sabater[5], Gianpaolo Balsamo[5], David Fairbairn[5], Catherine Meurey[1], Jean-Christophe Calvet[1]

[1] CNRM, Université de Toulouse, Météo-France, CNRS, Toulouse, France

[2] Instituto Dom Luiz, IDL, Faculty of Sciences, University of Lisbon, Portugal

[3] CESBIO, Université de Toulouse, CNRS, CNES, IRD, Toulouse, France

[4] CIRES/NOAA Earth System Research Laboratory, Boulder, CO, USA

[5] European Centre for Medium-Range Weather Forecasts, Shinfield Road, Reading RG2 9AX, UK

**\* Correspondence: clement.albergel@meteo.fr**

**Abstract-**This study demonstrates that LDAS-Monde, a global and offline Land Data Assimilation System (LDAS), that integrates satellite Earth observations into the ISBA (Interaction between Soil Biosphere and Atmosphere) Land Surface Model (LSM), is able to detect, monitor and forecast the impact of extreme weather on land surface states. LDAS-Monde jointly assimilates satellite derived Earth observations of surface soil moisture (SSM) and Leaf Area Index (LAI). It is run at global scale forced by ERA5 (LDAS_ERA5), the latest atmospheric reanalysis from the European Centre for Medium Range Weather Forecast (ECMWF) over 2010-2018 leading to a 9-yr, ~0.25° x 0.25° spatial resolution reanalysis of Land Surface Variables (LSVs). This reanalysis is then used to compute anomalies of land surface states, in order to (i) detect regions exposed to extreme weather such as droughts and heatwave events and (ii) address specific monitoring and forecasting requirements of LSVs for those regions. In this study, LDAS_ERA5 analysis is first successfully evaluated worldwide using several satellite-based datasets (SSM, LAI, Evapotranspiration, Gross Primary Production and Sun Induced Fluorescence), as well as in situ measurements (SSM, evapotranspiration and river discharge). The added value of assimilating the soil moisture and LAI is demonstrated with respect to a model simulation (open-loop, with no assimilation). Since the global LDAS_ERA5 has relatively coarse resolution, two higher spatial resolution experiments over two areas particularly affected by heatwaves and/or droughts in 2018 were run: North Western Europe and the Murray-Darling basin in South Eastern Australia. These experiments were forced with ECMWF Integrated Forecasting System (IFS) high resolution operational analysis (LDAS_HRES, ~0.10° x 0.10° spatial resolution) over 2017-2018, and both open-loop and analysis experiments compared once again. Since the IFS is a forecast system, it also allows LDAS-Monde to be used in forecast mode, and we demonstrate the added value of initializing 4- and 8-day LDAS-





35    HRES forecasts of the LSVs, from the LDAS-HRES assimilation run, compared to the open-loop
      experiments. This is particularly true for LAI that evolves on longer time space than SSM and is
      more sensitive to initial conditions than to atmospheric forcing, even at an 8-day lead time. This
      confirms that slowly evolving land initial conditions are paramount for forecasting LSVs and that
      LDAS-systems should jointly analyse both soil moisture and vegetation states. Finally evaluation of
40    the modelled snowpack is presented and the perspectives for snow data assimilation in LDAS-
      Monde are discussed.

      1    Introduction

      Extreme weather and climate events like heatwaves and droughts are likely to increase in frequency
      and/or magnitude (IPCC, 2012, Ionita et al., 2017). Amongst all the natural disasters, droughts are
the most detrimental (Bruce, 1994; Obasi, 1994; Cook et al., 2007; Mishra and Singh, 2010; WMO
      2017) and about one-fifth of damages caused by natural hazards can be attributed to droughts
      (Wilhite 2000). They also cost society billions of dollars every year (WMO 2017). It is therefore of
      paramount importance to implement tools that can monitor and warn about drought conditions
      (Svoboda, 2002; Luo and Wood, 2007; Blyverket et al., 2019) as well as their impact on land
surface variables (LSVs) and society (Di Napoli et al., 2019). A major scientific challenge in
      relation to the adaptation to climate change is to observe and simulate how land biophysical
      variables respond to those extreme events (IPCC, 2012).

      Droughts can be described as a deficit of water caused by a lack of precipitation. However its
      concept is broader and they are generally classified according to which part of the hydrological
cycle suffers from a water deficit (IPCC, 2014; Barella-Ortiz and Quintana-Seguí, 2018). Drought
      types are all related to precipitation deficit and most severe in areas of rain-fed crops agriculture
      with no irrigation. They include meteorological droughts (lack of precipitation), agricultural
      droughts (deficit of water in the soil), hydrological droughts (deficit of streamflow, water level in
      rivers) and environmental droughts (a combination of the previous droughts types). Because of the
effect of precipitation deficit propagating through the whole hydrological system, it can be stated
      that drought types are related (Wilhite, 2000). Complex interactions between continental surface
      and atmospheric processes have to be combined with human action in order to represent the wide
      ranging impacts of droughts on land surface conditions (Van Loon, 2015). As a consequence, Land
      Surface Models (LSMs) driven by high-quality gridded atmospheric variables and coupled to river-
routing system are key tools to address these challenges (Dirmeyer et al., 2006; Schellekens et al.,
      2017). Initially developed to provide boundary conditions to atmospheric models, the role of LSMs
      has evolved and they can now be used to monitor and forecast land surface conditions (Balsamo et



al., 2015; Balsamo et al., 2018; Schellekens et al., 2017). Additionally, the representation of LSVs by LSMs can be improved through the integration of Earth Observations (EOs) (e.g. Reichle et al.,

2007; Lahoz and de Lannoy, 2014; Kumar et al., 2018; Albergel et al., 2017, 2018a, 2019; Balsamo et al., 2018) as well as by coupling them with other models of the Earth system (e.g., de Rosnay et al., 2013, 2014; Kumar et al., 2018, Balsamo et al., 2018; Rodríguez-Fernández et al., 2019; Muñoz-Sabater et al., 2019). Satellite products are particularly relevant for such application. Satellite EOs related to the terrestrial hydrological, vegetation and energy cycles are now

unrestrictedly available at a global scale with high spatial resolution (at kilometric scale and below) and with long-term records (e.g., Lettenmaier et al., 2015, Balsamo et al., 2018). Combining EOs and LSMs through Land Data Assimilation Systems (LDASs) leads to enhanced initial land surface conditions which, in turn, lead to improved forecasts of weather patterns, sub-seasonal temperature and precipitation, agricultural and vegetation productivity, seasonal streamflow, floods and

droughts, as well as the carbon cycle (Bamzai and Shukla, 1999; Schlosser and Dirmeyer, 2001; Bierkens, M. and van Beek, 2009; Koster et al., 2010; Bauer et al., 2015; Massari et al, 2018; Albergel et al., 2018a, 2019, Rodríguez-Fernández et al., 2019; Muñoz-Sabater et al., 2019). Amongst the current land-only LDAS activities are several NASA-led (National Aeronautics and Space Administration) projects. Amongst them are the Global Land Data Assimilation System

(GLDAS, Rodell et al., 2004) run at global scale. The North American Land Data Assimilation System (NLDAS, Xia et al., 2012a, b) and the National Climate Assessment-Land Data Assimilation System (NCA-LDAS, Kumar et al., 2016, 2018, 2019) are run over the continental United States of America and the Famine Early Warning Systems Network (FEWS NET) Land Data Assimilation System (FLDAS, McNally et al., 2017) is run e.g. over Western, Eastern and Southern

Africa. Finally, the Carbon Cycle Data Assimilation System (CCDAS, Kaminski et al., 2002), the Coupled Land Vegetation LDAS (CLVLDAS, Sawada and Koike, 2014, Sawada et al., 2015), the Data Assimilation System for Land Surface Models using CLM4.5 proposed by Fox et al., 2018, the SMAP (Soil Moisture Active Passive) level 4 system (Reichle et al., 2019) as well as LDAS-Monde (Albergel et al., 2017, 2018, 2019) developed by the research department of Météo-France are

additional initiatives of combining EOs and LSMs through data assimilation. Few studies however have included the assimilation of multiple EOs and considered global applications (Kumar et al., 2018, Albergel et al., 2019). A more detailed description of the various existing LDASs is available in Kumar et al., 2018, Albergel et al., 2019 and references therein.

After several applications at regional and continental scales (Albergel et al., 2017, 2018, 2019,

Leroux et al., 2018, Tall et al., 2019, Blyverket et al., 2019, Bonan et al., 2019), LDAS-Monde was run at global scale forced by the latest atmospheric reanalysis from the European Centre for



Medium Range Weather Forecast (ECMWF), ERA5, over 2010-2018 leading to a 9-yr, 0.25° x 0.25° spatial resolution reanalysis of the LSVs (LDAS_ERA5). In this study, stemming from previous works referenced above, this global, offline, joint integration of Surface Soil Moisture

(SSM) and Leaf Area Index (LAI) EOs into the ISBA (Interaction between Soil Biosphere and Atmosphere) LSM (Noilhan and Planton, 1989, Noilhan and Mahfouf, 1996) are presented:

• An evaluation at global scale using diverse and complementary datasets such as evapotranspiration from the GLEAM project (Miralles et al., 2011, Martens et al., 2017), Gross Primary Production (GPP) from the FLUXCOM project (Tramontana et al., 2016, Jung et al.,

2017), Solar Induced Fluorescence (SIF) from the GOME-2 (Global Ozone Monitoring Experiment-2) scanning spectrometer (Munro et al., 2006, Joiner et al., 2016) and snow cover data from the Interactive Multi-sensor Snow and Ice Mapping System (or IMS, https://www.natice.noaa.gov/ims/, last access June 2019). It is also validated using reference observations including in situ evapotranspiration from the FLUXNET 2015 synthesis data set

(http://fluxnet.fluxdata.org/, last access June 2019), soil moisture from the International Soil Moisture Network (ISMN, https://ismn.geo.tuwien.ac.at/en/, last access June 2019) as well as river discharge from several networks across the world.

• An estimation of the mean LSVs climate over 2010-2018, used as reference for computing anomalies of the land surface conditions to (i) detect regions severely exposed to extreme weather

such as drought and heatwave events in 2018 and (ii) trigger more detailed monitoring and forecasting activities of the LSVs for those regions at higher spatial resolution.

The paper is organised in four sections as it follows: section 2 details the various components constituting LDAS-Monde: the ISBA LSM, the data assimilation scheme and the EOs assimilated as well as the different atmospheric forcing datasets used, followed by the experimental and

evaluation setup. Section 3 describes and discusses the impact of the analysis on the representation of the LSVs. The selection of 2 case studies over regions particularly affected by extreme events during 2018 and their detailed monitoring at higher spatial resolution combined with land surface forecasting activities is also presented. Finally section 4 provides conclusions and prospects.

## 2 Material and methods

The following subsections briefly describe the main components of LDAS-Monde: the ISBA LSM, its data assimilation scheme and two other key elements of the setup: atmospheric forcing and assimilated satellite derived observations. The experimental setup and the evaluation datasets used in this study are also presented.



### 2.1 LDAS-Monde

#### 2.1.1 ISBA Land Surface Model

Embedded within the SURFEX (SURFace EXternalisée, Masson et al., 2013, version 8.1) modelling platform developed by the research department of Météo-France (CNRM, Centre National de Recherches Météorologiques), LDAS-Monde (Albergel et al., 2017) allows the joint integration of satellite derived SSM and LAI into the $CO_2$-responsive (Calvet, et al., 1998, 2004,

Gibelin et al., 2006), multilayer diffusion scheme (Boone et al., 2000, Decharme et al., 2011) version of the ISBA LSM (Noilhan and Planton, 1989, Noilhan and Mahfouf, 1996) using a simplified version of an Extended Kalman Filter (SEKF, e.g. Mahfouf et al., 2009, Barbu et al., 2011, Fairbairn et al., 2017). It can be coupled to the ISBA-CTRIP hydrological model (ISBA-CTRIP for ISBA-CNRM, Total Runoff Integrating Pathways) as detailed in Decharme et al., (2019).

In such a configuration, ISBA is able to represent the transfer of water and heat through the soil based on a multilayer diffusion scheme, as well as plant growth and leaf-scale physiological processes. ISBA models key vegetation variables like LAI and above ground biomass, the diurnal cycle of water, carbon and energy fluxes. It computes a soil-vegetation composite using a single-source energy budget. In the $CO_2$-responsive versions of ISBA, photosynthesis is in control of the

evolution of vegetation variables. It makes vegetation growth possible as a result of an uptake of $CO_2$. Oppositely, a deficit of photosynthesis triggers higher mortality rates. Ecosystem respiration (RECO) is represented by the $CO_2$ being released by the soil-plant system and GPP by the carbon uptake related to photosynthesis. Finally, the net ecosystem exchange (NEE) consists of the difference between GPP and RECO. Each ISBA grid cell can be composed of up to 12 generic land

surface types, bare soil, rocks, and permanent snow and ice surfaces as well as nine plant functional types (needle leaf trees, evergreen broadleaf trees, deciduous broadleef trees, C3 crops, C4 crops, C4 irrigated crops, herbaceous, tropical herbaceous and wetlands). The ECOCLIMAP-II land cover database (Faroux et al., 2013) provides ISBA parameters for all of them.

ISBA multilayer diffusion scheme's default discretization is 14 layers over 12 m depth. The

following configuration is used in this study: thickness (depth) of each layers are (from top to down), 1 cm (0-1 cm), 3 cm (1-4 cm), 6 cm (4-10 cm), 10 cm (10-20 cm), 20 cm (20-40 cm), 20 cm (40-60 cm), 20 cm (60-80 cm), 20 cm (80-100 cm), 50 cm (100-150cm), 50 cm (150-200cm), 100 cm (200-300 cm), 200 cm (300-500 cm), 300 cm (500-800 cm) and 400 cm (800 to 1200 cm), see also Figure 1 of Decharme et al., 2011. Snow is represented using the ISBA 12-layers explicit snow

scheme (Boone and Etchevers, 2001, Decharme et al., 2016).

#### 2.1.2 CTRIP river routing system





The ISBA-TRIP river routing system is bale to simulate continental scale hydrological variables based on a set of three prognostic equations. They correspond to (i) the groundwater, (ii) the surface stream water and (iii) the seasonal floodplains. It converts the runoff simulated by ISBA into river discharge. ISBA-CTRIP river-routing network has a spatial resolution of 0.5° x 0.5° globally and is coupled daily with ISBA through the OASIS3-LCT coupler (Voldoire et al., 2017). ISBA provides to CTRIP updated fields of runoff, drainage, groundwater and floodplain recharges. In turn, CTRIP provides ISBA with water table depth, floodplain fraction as well as flood potential infiltration so that ISBA can simulate capillarity rise, evaporation and infiltration over flooded areas. A comprehensive overview of ISBA-CTRIP is available in Decharme et al., (2019).

### 2.1.3 Data assimilation

The SEKF used in LDAS-Monde is a 2-step sequential approach in which a forecast step is followed by an analysis step. The forecast step propagates the initial state of the model (being a short term forecast from the ISBA LSM) and then, the analysis step corrects this forecast by assimilating observations. Flow dependency between the model control variables and the observations are generated using finite differences from perturbed simulations. The analysis involves the computation of a Jacobian matrix having as many rows as assimilated observation types (here two: SSM and LAI) and as many columns as model control variables requested (here eight: soil moisture from the second to the eight layers of soil, 1-100cm, and LAI). Additionally to a control run, computing the Jacobian matrix requires perturbed runs, one for each control variable. Typically, for those runs, initial state of the control variable is perturbed by about 0.1% (see Albergel et al., 2017; Rüdiger et al., 2010). The length of the LDAS-Monde assimilation window is 24-hours. A mean volumetric standard deviation error is specified proportional to the soil moisture range (the difference between the volumetric field capacity and the wilting point, calculated as a function of the soil type, as given by Noilhan et Mahfouf, 1996) and scaled by a factor 0.04 for SSM in its model equivalent (the second layer of soil between 1 and 4 cm), scaled with and 0.02 for deeper layers (layers of soil 3 to 8, 4-100cm). The observational SSM error follows the same rule scaled by 0.05 and is consistent with errors typically expected for remotely sensed SSM (e.g., de Jeu et al., 2008, Gruber et al, 2016). Soil moisture errors for both the model and the observations are assumed to be proportional to the soil moisture range (being defined as the difference between the volumetric field capacity and the wilting point, calculated as a function of the soil type, as given by Noilhan et Mahfouf, 1996). The standard deviation of errors for the observed LAI is assumed to be 20% and a similar assumption is made for the standard deviation of errors of the modelled LAI values higher than 2 $m^2m^{-2}$. For modelled LAI values lower than 2 $m^2m^{-2}$, a constant error of 0.4


m²m⁻² is assumed (Barbu et al., 2011). More details can be found in Albergel et al, 2017 or Tall et al., 2019.

## 2.2 Atmospheric forcing

The lowest model level (about 10-meters above ground level) of air temperature, wind speed, specific humidity and pressure and the downwelling fluxes of shortwave and longwave radiations as
well as precipitations (partitioned in solid and liquid phases) are needed to force LDAS-Monde. In this study, LDAS-Monde is driven by several near-surface meteorological fields from ECMWF, its most recent atmospheric reanalysis (ERA5), as well as its high resolution operational weather analysis and forecasts (HRES). ERA5 (Hersbach et al., 2018, 2019 submitted) is the fifth generation of European reanalyses produced by the ECMWF. This atmospheric reanalysis a key element of the
Copernicus Climate Change Service (C3S, EU-funded) and is available from 1979 onward (data is released about 2 months behind real time). ERA5 has hourly output analysis, 31 km horizontal dimension and 137 levels in the vertical dimension. Although being quite new, ERA5 quality has already been evaluated in the scientific literature. Urraca et al. (2018) have compared incoming solar radiation from both ERA5 and the former ERA-interim reanalysis (Dee et al., 2011) at a
global scale and found evidence that ERA5 outperforms ERA-Interim. In another study, Beck et al. (2019) have highlighted the good performance of ERA5 precipitation with respect to a set of 26 gridded (sub-daily) precipitation data sources by comparing them to Stage-IV gauge-radar data over the CONUS domain (CONtinental United States of America). Tall et al. (2019) have used in situ measurements of precipitation at more than 100 stations spanning all over Burkina-Faso in western
Africa as well as incoming solar radiation from 4 in situ stations to evaluate the quality of ERA5 over ERA-Interim with positive outcomes for ERA5 as well. They have also evaluated both reanalysis datasets through their impact on the representation of LSVs when used to force the ISBA LSM, again demonstrating a clear advantage for ERA5. Similar work has been done by Albergel et al. (2018a), over North America, they found enhanced performances in the representation of
evaporation, snow depth, soil moisture as well as river discharge when the ISBA-LSM was forced by ERA5 compared to ERA-Interim. At the time of the study, ERA5 underlying model and data assimilation system (Cycle 41r2) are very similar to that of the operational weather forecast, HRES, which has production cycles ranging from 41r2 to 45r1 during the study period(it is 46r1 from June 2019, more information at https://www.ecmwf.int/en/forecasts/documentation-and-support/changes-
ecmwf-model, last access July 2019). The main difference between ERA5 and HRES over the considered period is the horizontal resolution, 9 km in HRES. Atmospheric forcing is interpolated from the native grids of ERA5 and HRES to regular grids of 0.25° × 0.25° and 0.1° × 0.1°, using a

bilinear interpolation from the native reanalysis grid to the regular grid. The four neighbours in the source grid fitting latitude and longitude were linearly interpolated. ERA5 and HRES were used in

Albergel et al. (2019) to force LDAS-Monde in order to study the impact of the 2018 summer heatwave in Europe. Authors have highlighted that the HRES configuration exhibits better monitoring skills than the coarser resolution ERA5 configuration.

From the forecast initialized at 00:00 UTC, HRES is also available with a 10-day lead time, but with changes in the temporal resolution. HRES forecast step frequency is hourly up to time step 90

(i.e. day 3), 3-hourly from time-step 90 to 144 (i.e. day 6) and 6-hourly from time-step 144 to 240 (i.e. day 10). In this study, for forecast experiments (see section 2.4 for details on the experimental setup) HRES forecasts with a 10-day lead time are used to initialize forecasts of the LSVs from LDAS_HRES open-loop and analysis configurations in order to evaluate the impact of the initialisation on the forecast of LSVs. The original 3-hourly time steps are used up to day 6 (time

step 144), the 6-hourly time steps from day 6 to 10 are interpolated to 3-hourly frequency to avoid discontinuities.

### 2.3  Assimilated satellite Earth Observations

Two types of satellite derived variables are assimilated in LDAS-Monde, ASCAT Soil Wetness Index (SWI) and LAI GEOV1. They are both freely available through the Copernicus Global Land

Service (CGLS, https://land.copernicus.eu/global/index.html, last access June 2019). They are illustrated by Figure 1.

ASCAT stands for Advanced Scatterometer, it is an active C-band microwave sensor that is onboard the European MetOp polar orbiting satellites (METOP-A, from 2006, B from 2012 and also C from 2018). From ASCAT radar backscatter coefficients, it is possible to derive information on SSM

following a change detection approach (Wagner et al., 1999, Bartalis et al., 2007). The recursive form of an exponential filter (Albergel et al., 2008), is then applied to estimate the Soil Wetness Index (SWI) using a timescale parameter, T (varying between 1 day and 100 days) and ranging between 0 (dry) and 100 (wet). T is a surrogate parameter for all the processes potentially affecting the temporal dynamics of soil moisture (like, soil hydraulic properties and thickness of the soil

layer, evaporation, run-off and vertical gradient of soil properties such as texture and density). In this study, CGLS SWI-001 (i.e. produced with a T-value of 1 day) is used as a proxy of SSM (Kidd et al., 2013). Grid points with an average altitude exceeding 1500 m above sea level as well as those with more than 15 % of urban land cover were rejected as those conditions are known to affect the retrieval of SSM. Prior to their assimilation, SSM has to be converted from the observation space to

the model space. This is done through a linear rescaling as proposed by Scipal et al. (2007), where





the observations mean and variance are matched to the modelled soil moisture mean and variance from the second layer of soil (1-4 cm depth). This rescaling known as Cumulative Distribution Function (CDF) matching is run at seasonal scale using a 3-month moving window as suggested by Draper et al., (2011), Barbu et al., (2014).

The LAI GEOV1 observations are based on data from from both SPOT-VGT and then PROBA-V satellites. They span from 1999 to present, have a 1km x 1km spatial resolution and are produced daily according to the methodology developed by Baret et al. (2013). As in previous studies (e.g, Barbu et al., 2014, Albergel et al., 2019), observations are interpolated by an arithmetic average to the model grid points (0.25° or 0.10° in this study), if at least 50 % of the model grid points are

observed (i.e. half the maximum amount). LAI GEOV1 observations have a temporal frequency of 10 days at best (in the presence of clouds, no observation are available). LAI data are masked in the presence of snow.

### 2.4      Experimental setup

LDAS-Monde is first run at a global scale, at 0.25° x 0.25° spatial resolution, forced by ERA5

atmospheric reanalysis and assimilating SSM and LAI EOs from 2010 to 2018 (LDAS_ERA5 hereafter). LDAS-ERA5 was spun-up by running year 2010 twenty times. LDAS_ERA5 analysis as well as its model counterpart (open-loop, i.e. no data assimilation) are presented and evaluated in this study.

This 9-yr global reanalysis was then used to provide a climatology for estimating anomalies of the

land surface conditions. Significant anomalies were used to trigger more detailed monitoring as well as forecasting activities for a region of interest. 19 regions across the globe known for being potential hot spots for droughts and heat waves were selected. They are listed in Table I and presented in Figure 2. Monthly anomalies of LDAS_ERA5 analysis of SSM and LAI for those 19 regions are assessed for 2018 (with respect to the 2010-2018 period) and regions presenting

significant level of anomalies were selected and further investigated. For them, LDAS-Monde has been driven by HRES atmospheric analysis leading to a 0.1° x 0.1° reanalysis of the LSVs from April 2016 to December 2018 (LDAS_HRES herafter). The period 2017-2018 is presented, HRES is available at this spatial resolution from April 2016, only, and the time period from April to December 2016 is used as a short spinup. LDAS_HRES complements the coarser spatial resolution

LDAS_ERA5. HRES forecasts with a 10 day lead time are also used initialised by either LDAS_HRES open-loop or analysis (LDAS_Fc hereafter) in order to assess the impact of the initialisation on the forecast. A summary of the experimental setup is given in Table II.





### 2.5    Evaluation datasets and metrics

This study uses several satellite-derived estimates of EOs as well as in situ measurement data.
LDAS_ERA5 analysis impact is assessed with respect to the open-loop model run (i.e. no
assimilation). The two assimilated datasets, CGLS SSM and LAI, were used to verify to which
extent the assimilation system was able to produce analyses closer to them (i.e. suggesting a healthy
behaviour from the data assimilation system). Then several independent spatially distributed
datasets: (namely) evapotranspiration from the GLEAM project (Miralles et al., 2011, Martens et
al., 2017, version 3b entirely satellite driven), Gross Primary Production (GPP) from the
FLUXCOM project (Tramontana et al., 2016, Jung et al., 2017), Sun Induced Fluorescence (SIF)
from the GOME-2 (Global Ozone Monitoring Experiment-2) scanning spectrometer (Munro et al.,
2006, Joiner et al., 2016) and snow cover data from the Interactive Multi-sensor Snow and Ice
Mapping System (or IMS, https://www.natice.noaa.gov/ims/) were used in the evaluation process.
Ground based measurements of soil moisture from the International Soil Moisture Network (ISMN,
Dorigo et al., 2011, 2015) were used, also, along with several river discharge observations (see
Table    S1)    and    evapotranspiration    from    FLUXNET-2015    synthesis    data    set
(http://fluxnet.fluxdata.org/data/fluxnet2015-dataset/, last access: June 2019). Most of these ground
stations are located in Europe and North America and they were already used in previous studies
(e.g. Albergel et al., 2017, 2018a,b, Leroux et al., 2019) to assess the LDAS-Monde quality.
Therefore LDAS-Monde evaluation using ground measurements is discussed in the result section
while figures are reported as supplementary materials of this study. Evaluation datasets are listed in
Table III along with the metrics used (correlation, Root Mean Square Differences -RMSD- and
unbiased RMSD -ubRMSD- and bias). For in situ datasets, a Normalized Information Contribution
(NIC, Eq.(1)) measure is applied to the correlation values to quantify the improvement or
degradation due to the specific configuration.

$$\text{NIC}_R = \frac{R_{(\text{Analysis})} - R_{(\text{Model})}}{1 - R_{(\text{Model})}} \times 100 \qquad\qquad\qquad \text{Eq.(1)}$$

NIC scores were classified according to three categories: (i) negative impact from the analysis with
respect to the open-loop with values smaller than -3 %, (ii) positive impact from the analysis with
respect to the open-loop with values greater than +3 % and (iii) neutral impact from the analysis
with respect to the open-loop with values between -3 % and 3 %.
The Nash-Sucliffe Efficiency score (NSE, Eq.(2), Nash and Sutcliffe, 1970) is used to evaluate
LDAS_ERA5 experiments ability to represent the monthly discharge dynamics.





$$\text{NSE} = 1 - \frac{\sum\limits_{mt=1}^{T} \left(Q_s^{mt} - Q_o^{mt}\right)^2}{\sum\limits_{mt=1}^{t} \left(Q_s^{mt} - \overline{Q_s^{mt}}\right)^2} \qquad\qquad \text{Eq.(2)}$$

where $Q_s^{mt}$ is the monthly river discharge from LDAS_ERA5 (analysis or open-loop) at month $mt$,

and $Q_o^{mt}$ is the observed river discharge at month $mt$. NSE can vary between $-\infty$ and 1. An exact

match between model predictions and observed data is defined as a value of 1, whereas a value of 0

means that the model predictions have the same accuracy as the mean of the observed data. Finally

negative values represent when the observed mean is a better predictor than the model simulation.

NIC presented in Eq.(1) has also been applied to NSE scores to assess the added value of

LDAS_ERA5 analysis over its open-loop counterpart.

### 3    Results

#### 3.1    Global assessment of LDAS_ERA5

##### 3.1.1    Gridded datasets

Figure 3 presents mean RMSD values between the observations and LDAS_ERA5 for the open

loop (Figure 3a), and for the analysis (Figure 3b) for LAI over 2010-2018. Because LAI

observations are ingested into the model, the assimilation reduces the LAI RMSD values almost

everywhere. It can be noted that rather large LAI RMSD values ($> 1.5$ $m^2m^{-2}$) can remain in some

areas after the assimilation, especially in densely forested areas. Figure 4 illustrates latitudinal plots

of LAI, SSM, GPP and evapotranspiration for LDAS_ERA5 before assimilation (the open-loop)

and after assimilation (the analysis) along with observations. The number of points considered per

latitudinal stripes of 0.25° is represented, also. From Figure 4a it is possible to appreciate the

positive impact of the analysis compared to the open loop, with the former being closer to the

observations. Improvement from the analysis occurs from nearly 80°North to about 55° South, areas

around the equator are particularly improved. A smaller impact than for LAI is obtained for SSM,

GPP and EVAP, hardly visible at this scale. The mean latitudinal results show a consistent

difference in terms of GPP and Evapotranspiraton between the LDAS_ERA5 and the observational

products. These differences are systematic with higher values in tropical regions. Figure 5

represents latitudinal plots of score differences (correlations and RMSD) for LAI, SSM, GPP, EVAP

and SIF (Figure 5i, correlation only). Score differences are computed as follow, analysis minus

open-loop using monthly averaged over 2010-2018 for LAI and SSM, 2010-2013 for GPP, 2010-

2016 for EVAP and 2010-2015 for SIF. For SIF only differences in correlation are represented as it





is used to evaluate GPP variability as in Leroux et al., 2018. For each panel of Figure 5, the vertical dashed line represents the 0-value. Therefore, for plots of correlation differences, positive values

indicate an improvement from the analysis with respect to the open-loop simulation. Similarly, for plots of RMSD differences negative values indicate an improvement from the analysis with respect to the open-loop simulation. LAI and SSM being assimilated variables, the analysis leads to a clear improvement in both correlation and RMSD. Such improvement is expected and reflects the healthy behaviour of the assimilation system. Both variables are improved at almost all latitudes with the

exception around 45°S for LAI correlation values (very few land points). For SSM a noticeable improvement in both correlation and RMSD is found around 20°N corresponding mainly to an improvement in the Sahara desert (not shown). GPP is also improved across almost all latitude with a particularly positive impact below 20°N which is also true for EVAP. This variable is less impacted by the analysis and some parts of the world show a decrease in e.g. RMSD values. Panels

of Figure 6 illustrate histograms of score differences (correlation and RMSD, analysis minus open-loop) for LAI, SSM, GPP, EVAP and SIF. Number of available data as well as the percentage of positive and negative values are reported. For correlations (RMSD) differences, positive (negative) values indicate an improvement from the analysis over the open-loop. It complements Figure 5. Regarding LAI the analysis improves 96.9% of the grid points for correlations and 99.9% for

RMSD. As for SSM, correlation values are improved for 92.8% of the grid points, it is 92.4% for RMSD. When using independent datasets such as GPP and SIF, one may also notice an improvement from the analysis, correlation (RMSD) are better for 81.1% (74.1%), 79.7% (NA) of the grid points. Results using the GLEAM dataset for evapotranspiration are more contrasted with 63.6% (48.9%) of the grid points showing an improvement from the analysis and it is worth

mentioning that 24.9% (39.6%) of the grid point shows a degradation. However GLEAM only estimates (root-zone) soil moisture and terrestrial evaporation, while ISBA in LDAS_ERA5 is a physically-based land surface model, accounting for more processes linked to vegetation.

Finally, Figure 7 and Figure 8 illustrate snow cover evaluation. LDAS_ERA5 snow cover was evaluated against the IMS snow cover (as e.g. in Orsolini et al., 2019). The IMS snow cover product

combines ground observations and satellite data from microwave and visible sensors (using geostationary and polar orbiting satellites) to provide snow cover information in all weather conditions. The IMS product is available daily for the northern hemisphere. Figure 7 shows the averaged northern hemisphere snow cover fraction for the 2010-2018 period and is complemented by all panels of Figure 8 showing (i) maps of IMS snow cover (top row) for 3 seasons, September-

October-November (SON), December-January-February (DJF) and March-April-May (MAM), respectively, (ii) maps of snow cover from LDAS_ERA5 open-loop (second row), (iii) maps of





snow cover differences between the open-loop and IMS data and (iv) maps of snow cover differences between the analysis and the open-loop. LDAS_ERA5 open-loop compares very well with the IMS snow-cover data in the accumulation season from September to February (Figure 7

and panels d) to I) of Figure 8), only with an overestimation over the Tibetan Plateau. The latter issue over Tibet in ERA5 is not new, and consistent with previous studies like Orsolini et al., 2019. An early melt in Spring compared to observations is noted in LDAS_ERA5 and could be related with the snow cover parametrization in ISBA. As expected, the analysis has an almost neutral impact on snow as both SSM and LAI observations are filtered out from frozen/snow condition

(Figure 7 and panels (j), (k) and (l) of Figure 8). This clearly shows however an area of potential improvement of data assimilation using satellite data such as the IMS one (as in e.g. de Rosnay et al., 2014).

### 3.1.2    Ground-based datasets

LDAS_ERA5 analysis and open-loop are also evaluated using in situ measurements of

evapotranspiration, river discharge and surface soil moisture across the world. Daily in situ measurements of evapotranspiration from the Fluxnet2015 synthesis data set (http://fluxnet.fluxdata.org/, last access June 2019) are first used in this study. Stations with at least two years of data (after 2010) are retained leading to a pool of 85 stations available for this evaluation (note that none of these stations include 2015), they are listed in Table S2. LDAS_ERA5

ability to represent evapotranspiration is evaluated using correlation (R), RMSD and ubRMSD as well as bias (LDAS_ERA5 minus observations). Median R, RMSD, ubRMSD and bias for LDAS_ERA5 analysis (open-loop) are 0.73 (0.72), 28.74 (29.60) w.m$^{-2}$, 27.37 (26.92) wm$^{-2}$ and 4.64 (4.40) wm$^{-2}$, respectively. These numbers depict a small advantage of the analysis over the open-loop configuration.

Figure S1(a) represents the added value of the analysis based on NIC$_R$ (Eq.(1)), large blue circles represent a positive impact from the analysis (20 stations) at NIC$_R$ greater that +3 (i.e. R values are better when the analysis is used than when the model is used) while large red circles represent a degradation from the analysis (5 stations) at NIC$_R$ smaller than -3. Stations with a rather neutral impact (60 stations) at NIC$_R$ between [-3;+3] are not reported for sake of clarity. Figure S1 (b), (c),

(d) and (e) are scatter-plots of R, ubRMSD, absolute bias and RMSD between LDAS_ERA5 open-loop and the 85 stations from the Fluxnet2015 (y-axis) against LDAS_ERA5 analysis and the same pool of stations (x-axis). 56 stations (out of 85) have better R values considering the analysis. They are 41 for ubRMSD, 47 for RMSD and 44 for absolute bias.



Over 2010-2017, river discharge from LDAS_ERA5 analysis and open-loop runs were compared to

daily streamflow data from 7 large networks across the world (see Table S1). As in Albergel et al., 2018a, data were selected for sub-basins with rather large drainage areas (10,000 km$^2$ or greater) due to the low resolution of CTRIP (0.5 x 0.5°) and with observation time series of 4 years or more. Results are illustrated by Figures S2 and S3. Figure S2 represents NSE scores and as NSE values below -2 were discarded, it leads to a subset of 982 stations available. Most of them are located in

North America and Europe while a few are available in South America and Africa. Figure S2 is complemented by Figure S3. Panel a of Figure S3 represents the NIC scores applied to NSE scores and emphasizes the added value of LDAS_ERA5 analysis over the open-loop. 74% of this subset of stations presents a rather neutral impact from the analysis (at NIC ranging between -3 and +3) while 26% (254 stations) presents an impact greater or smaller than 3%. When the analysis impacts the

representation of river discharge, this impact tends to be positive with 74% (189 stations) having a NIC score greater than 3% while only 26% (65 stations) presents NIC score smaller than -3%. This results are supported by panels (b) and (c) of Figure S3, also (density of NSE scores for LDAS_ERA5 analysis and open-loop, scatter-plot of NSE scores for LDAS_ERA5 analysis and open-loop, respectively).

In situ measurements of surface soil moisture from 20 networks across 14 countries available from the ISMN are also used to evaluate the performance of SSM analysis. They represent more than 900 stations with at least 2 years of daily data over 2010-2018. Sensors at 5 cm depth are compared with soil moisture from LDAS_ERA5 third layer of soil (4-10 cm). Beside 11 stations located in 4 countries of Western Africa (Benin, Mali, Sénégal and Niger) and 19 stations in Australia, most of

the station are located in North America and Europe, see Table S3. Using this pool of stations, averaged statistical metrics (ubRMSD, R and bias) are similar for both LDAS_ERA5 analysis and open-loop even if local differences exist. For the analysis, averaged R values is 0.67 (it is 0.66 for the open-loop) with averaged-network values going up to 0.88 (SOILSCAPE network, 49 stations in the USA) and always higher than 0.55 except for one network, ARM (10 stations in the USA)

presenting an averaged R value of 0.29. Averaged ubRMSD and bias (LDAS_ERA5 minus in situ) are 0.058 m$^3$m$^{-3}$ and 0.079 m$^3$m$^{-3}$ for the analysis, 0.059 m$^3$m$^{-3}$ and 0.078 m$^3$m$^{-3}$ for the open-loop, respectively. Results for each network are summarized in Table S2. NIC (Eq.1) has also been applied to R values, 64% of the pool of stations present a neutral impact from the analysis (at NIC ranging between -3 and +3), 12% present a negative impact (at NIC < -3) and 24% present a

positive impact at (NIC>+3). NIC scores are also presented by Figure S4 at global scale ((a) panel) and with a zoom over the continental USA ((b) panel).





For evapotranspiration, river discharge and surface soil moisture it can be stated that there is an advantage from LDAS_ERA5 analysis with respect to its open-loop counterpart. Even if the distribution of the averaged statistical metrics can be rather similar for both (particularly true for

surface soil moisture evaluation), there are significant differences for some sites, which shows the added value of the analysis with respect to the open-loop.

### 3.2    Monitoring and forecasts for areas under severe/extreme conditions

For each individual region presented in Table I and Figure 2, monthly anomalies (scaled by the standard deviation) of analysed SSM (second layer of soil, 1-4cm) and LAI for 2018 were assessed

with respect to the 2010-2018 period. The anomalies (see Figure 9) highlight three regions, two presenting strong negative anomalies for both SSM and LAI for almost all 2018 (north western Europe, WEUR, and the Murray-Darling basin, MUDA, in south eastern Australia) and one presenting strong positive anomalies of SSM and LAI in Eastern Africa (EAFR). WEUR and MUDA regions were affected by a severe heatwave and a drought in 2018 impacting LSVs

analysed by LDAS_ERA5. According to Figure 9, monthly anomalies of SSM and LAI for MUDA are negative through all the year 2018 with 7 and 6 months presenting LAI and SSM anomalies below -1 standard deviation (stdev), respectively. WEUR has negative SSM anomalies from May to December 2018 with values going below -2 stdev. LAI was severely impacted as well with July to October 2018 presenting negative anomalies below -2 stdev. For WEUR, 5 months present LAI and

SSM anomalies below -1 stdev. EAFR has experienced 3 and 7 months with positive anomalies for SSM and LAI in 2018 above 1 stdev (8 and 7 months consecutively present positive anomalies for SSM and LAI respecively).

According to the National Oceanic and Atmospheric Administration (NOAA), Europe experienced its warmest summer since continental records began in 1910 at +2.16°C (Global Climate Report,

https://www.ncdc.noaa.gov/sotc/global/ last access April 2019). In Europe, temperature for the whole summer 2018 was above climatology. The summer 2018 heatwave in Europe was already reported in the scientific literature (e.g. Magnusson et al., 2018, Albergel et al., 2019, Blyverket et al., 2019). In its 70th Special Climate Statement, the Australian Bureau of Meteorology (BoM) has reported a very hot and dry summer 2018 in eastern Australia (BoM, 2019). Like much of Australia,

the Murray Darling Basin has experienced a remarkably dry and hot weather during 2018 (http://www.bom.gov.au/state-of-the-climate/, last visited: April 2019). The annual maximum temperature for the Murray Darling Basin as a whole was more than two degrees above average during 2018. The northern Murray–Darling Basin in particular was severely affected with inflows to all catchments persistently well below average. Finally, the East Africa Seasonal Monitor based on



the Famine Early Warning System Network (FEWS) confirms above-average rainfall amounts as well as significantly greener than normal vegetation conditions (e.g., https://reliefweb.int/report/somalia/east-africa-seasonal-monitor-july-27-2018, last visited: April 2019). As this study focuses on monitoring and forecasting the impact of severe conditions on LSVs, WEUR and MUDA are selected for further investigation.

3.2.1    Case studies for assessing LDAS-Monde medium resolutions (0.25° x 0.25°)

Figure 10 illustrates seasonal cycles of observed LAI (Figure 10a) and SWI (Figure 10e), LDAS_ERA5 analysis and open-loop LAI (Figure 10b) and SWI (Figure 10f) for the WEUR domain. The last year (2018) is compared to an average of the previous years (2010-2017). From Figure 10a one may see the heatwave impact with a sharp drop in observed LAI values from June to

November 2018 (solid green line). Such low LAI values have never been observed over the eight previous year (dashed green line for the 2010-2017 averaged along with the 2010-2017 minimum and maximum observations in shaded green). A similar behaviour is also visible in the ASCAT SWI dataset in Figure 10e with the lowest values ever reached in this 2010-2018 period. Over WEUR, LDAS_ERA5 open-loop overestimates LAI in the second part of the year as already highlighted by

several studies (e.g. Albergel et al., 2017, 2019). LDAS_ERA5 analysis has a positive impact, reducing LAI values, as seen on Figure 10b (LAI open-loop in blue, analysis in red) and on Figure 10c representing RMSD seasonal cycles. LDAS_ERA5 analysis also leads to an improvement in correlations for LAI (see Figure 10d). Similar conclusions can be drawn for SSM (Figure 10e to h). Note that for data assimilation and statistical scores, ASCAT SWI estimates were converted into the

model space, in $m^3m^{-3}$, as detailed in section 2.3. Finally looking at the MUDA area (panels of Figure 11) one may appreciate similar positive impact from the analysis over the open-loop simulation. Almost all 2018 presents the lowest anomaly values for both SSM and LAI. For both MUDA and WEUR the smaller differences between LDAS_ERA5 analysis and open-loop in 2018 than in 2010-2017 (Figure 10 b and f, Figure 11 b and f) also suggest that both extreme events were

well captured in the atmospheric forcing used to drive LDAS_ERA5 while the statistical scores presented in Figure 10 c, d, g, h as well as in Figure 11 c, d, g, h also suggest an improvement from the analysis over the open-loop simulation.

3.2.2    Case studies for assessing LDAS-Monde high resolutions (0.1° x 0.1°)

For these two specific areas, LDAS-Monde was also run forced by HRES (LDAS_HRES) at 0.1° x

0.1° spatial resolution over April 2016 to December 2018. Additionally to LDAS_HRES analysis, forecast experiments with a lead time of 4-days and 8-days, initialised by either LDAS_HRES analysis or open-loop are presented for 2017-2018 (for SSM and LAI) in order to assess the impact of the initial conditions on the forecast of LSVs. Upper panels of Figure 12 and Figure 13, illustrate



seasonal RMSD (Figure 12a, 13a) and correlation (Figure 12b, 13b) values between SSM from the
second layer of soil (1–4 cm) from LDAS-Monde forced by HRES (LDAS_HRES, open-loop and
analysis) and ASCAT SSM estimates over 2017-2018. Scores between SSM from the second layer
of soil of LDAS_HRES 4-day forecast (LDAS_fc4, initialised by either the open-loop or analysis)
and 8-day forecast (LDAS_fc8, initialised by either the open-loop or analysis) and ASCAT SSM
estimates are reported, also. From the upper panels of those figures one may notice a small
improvement from the analysis (solid red line) over the open-loop simulation (solid blue line),
slightly decreasing RMSD values and increasing correlations values. However no improvement (nor
degradation) is visible from the 4-d and 8-d forecasts experiments initialised by LDAS_HRES
analysis over those initialised by LDAS_HRES open-loop, they  display very similar scores.
LDAS_HRES SSM is of better quality than LDAS_fc4 and LDAS_fc8. Note however that for the
MUDA area, a 4-d forecast of surface soil moisture initialised by the analysis is of better quality
than a 4-d forecast initialised by the open-loop. Those results suggest that this fast evolving model
variable (SSM between 1 cm and 4 cm depth) is more sensitive to the atmospheric forcing than to
initial conditions (at least within the forecast range presented in this study) and it can be assumed
that the 4-day and 8-day atmospherical forecast from HRES is of poorer quality that its first day
analysis. Results for LAI are different than for SSM (lower panels of Figure 12 and Figure 13).
Firstly, there is a large improvement from the analysis (solid red line) over the open-loop (solid blue
line), particularly in the LAI decaying phase (Boreal and Austral autumns mainly). Secondly,
LDAS_HRES open-loop, LDAS_fc4 and LDAS_fc8 initialised by LDAS_HRES open-loop present
very similar skills, so do LDAS_fc4 and LDAS_fc8 initialised by LDAS_HRES analysis. They
outperform however skills of LDAS_HRES open-loop, LDAS_fc4 and LDAS_fc8 initialised by
LDAS_HRES open-loop. This suggests that LAI is more sensitive to its initial conditions than to
the atmospheric forcing (at least within the forecast range presented in this study) and that
forecasting  LAI is also a matter of initial conditions. This is true for these two contrasted areas,
WEUR and MUDA. These results are corroborated by Figures 14 (for WEUR) and 15 (for MUDA),
top rows illustrate SSM and bottom rows LAI. Figures 14(a) and 15(a)  show RMSD values
between LDAS_HRES open-loop SSM (1-4 cm) and ASCAT SSM over 2017-2018 for the WEUR
and MUDA domains, respectively. Due to the CDF matching applied to  ASCAT estimates, RMSD
values are rather small. For the WEUR (MUDA) domain they range from 0 to 0.048 $m^3m^{-3}$ (0 to
0.040 $m^3m^{-3}$). Figures 14(b) and 15(b) represent maps of RMSD differences between LDAS_HRES
analysis (open-loop) and ASCAT SSM estimates over 2017-2018 for the WEUR and MUDA
domains, as well. Both maps are dominated by negative values (in blue) indicating that RMSD
values are smaller (better) when using LDAS_HRES analysis than when using LDAS_HRES open-





loop. It is also worth-mentioning than no positive differences (i.e. a degradation from the analysis) are present in those maps. RMSD differences for the WEUR domain range from -0.004 $m^3m^{-3}$ to

0.004 $m^3m^{-3}$ meaning that the analysis is improving them by about 8 %. For the MUDA domain, they are improved by about 15%. Figures 14(c), (d) 15(c),(d) are also maps of RMSD differences, they consider forecast experiments (LDAS_fc4, LDAS_fc8). It appears that for both domains, the impact from the initialisation is rather small with few coloured areas, strengthening previous results suggesting that to forecast SSM variable, forcing quality is more important than initial conditions. It

is different for LAI, RMSD values for LDAS_HRES open-loop are ranging between 0 and 1.6 $m^2m^{-2}$ over WEUR, 0 and 1 $m^2m^{-2}$ over MUDA (Figures 14(e) and 15(e)). RMSD values are improved by up to 37 % over WEUR and up to 60% over MUDA by the analysis (Figures 14(f) and 15(f)). Improvement from the analysis over the open-loop experiment is consistent through all the WEUR domain while it is mainly the south eastern part of the MUDA domain that is improved (the north western part has low RMSD values as the open-loop).

western part has low RMSD values as the open-loop).

## 4    Discussion and conclusion

This study has demonstrated that combining a LSM, satellite EOs and atmospheric forcing through LDAS-Monde has a great potential to represent the impact of extreme weather (heatwaves and droughts) on land surface conditions. LDAS-Monde is now ready for use in various applications

such as (i) reanalyses of land Essential Climate Variables (ECVs), (ii) monitoring of water resources, drought and vegetation, and (iii) detection of severe conditions over land and initialisation of LSVs forecast. It has been applied in this study to past events of 2018 with respect to a short period of time (2010-2018) as a demonstrator but will be extended to longer time period. LDAS-Monde operational use in near real time has the capacity to serve as an emergency

monitoring system for the LSVs. Using atmospheric reanalysis like ERA5 for force LDAS-Monde guarantees a high level of consistency because of its fixed configuration. ERA5 coarse spatial resolution makes it affordable to run long term, large scale LDAS-Monde experiments. With ERA5 available now back to 1979 and covering near real-time needs with the ERA5T (https://climate.copernicus.eu/climate-reanalysis), an LDAS_ERA5 configuration would be able to

provide a long term, near real time coarse resolution (0.25° x 0.25°) climatology as reference for anomalies of the land surface conditions. Significant anomalies could then be used to trigger more focussed "on-demand" simulations for regions experiencing extreme weather. In that case LDAS-Monde could be run forced by e.g. ECMWF operational high resolution product (0.10° x 0.10°) in monitoring and forecast (up to 10-d ahead) modes, as was presented here for two regions in North

Western Europe and South Eastern Australia. Our results showed a very small impact of initial



conditions on the forecasts of SSM. This was expected due to the reduced memory of the top soil surface (0-1cm), which is dominated by meteorological variability. However, the LAI initialisation had significant impact on the LAI forecast skill. This was also expected due to the memory of vegetation evolution. Despite the expected behaviour of these two LSVs in forecasting, our results

show that LDAS-Monde system is capable of propagating the initial LAI conditions, which is relevant not only for LSV medium-range forecasting but with potential for longer lead-times. The strong impact of LAI initialisation on the forecast does not seem to propagate to surface soil moisture and further studies are necessary to test the impact of initial conditions to more variables from LDAS-Monde (including soil moisture in deeper layers, evapotranspiration). Another

possibility would be to force LDAS-Monde using ECMWF ensemble forecasts, although the ensemble system has coarser spatial-resolution (~0.20° x 0.20°), it offers a 15-day forecast and a 51 member ensemble, which can introduce forcing uncertainty into the LSVs. The maximum range of the soil and vegetation forecast could even reach up to six months if using seasonal atmospheric forecasts as forcing.

LDAS-Monde has well identified areas of developments that can further improve the representation of LSVs. For instance, it does not consider snow data assimilation yet and it has been shown in this study than if the snow accumulation seems to be represented correctly in the system, it suffers from an early snow-melt in spring. To overcome this issue, two possibilities will be explored. Firstly using a recently developed ISBA parametrisation, MEB for Multiple Energy Budget which is

known to lead to a better representation of the snowpack (Boone et al., 2017), in particular in the densely forested areas of the Northern Hemisphere where large differences between LDAS-Monde and the IMS snow cover were found in spring (Figure 8(i), Aaron Boone CNRM, personal communication June 2019) and (ii) adapting the current data assimilation scheme of LDAS-Monde to permit assimilation the IMS snow cover data (as done e.g. at ECMWF, de Rosnay et al., 2014).

The Current SEKF data assimilation scheme is also being revisited. Even though it has provided good results, one of its limitations is the computation of a Jacobian matrix which requires one model run for each control variable, requiring significant computational resources with increased number of control variables. That is why more flexible Ensemble based approaches like the Ensemble Square Root Filter (EnSRF) have recently been implemented (Fairbain et al., 2015,

Bonan et al., 2019). Bonan et al., 2019 have evaluated performances from the EnSRF and the SEKF over the Euro-Mediterranean area. Both data assimilation schemes have a similar behaviour for LAI while for SSM, EnSRF estimates tend to be closer to observations than those from the SEKF. They have also conducted an independent evaluation of both assimilation approaches using satellite estimates of evapotranspiration and GPP as well as measures of river discharges from gauging





stations. They have found that the EnSRF leads to a systematic (moderate) improvement for
       evapotranspiration and GPP and a highly positive impact on river discharges, while the SEKF lead
       to more contrasting performance. As for applications in hydrology, the 0.5° x 0.5° spatial resolution
       TRIP river network is currently being improved to 1/12° x 1/12° globally. Also, the added value of
       LDAS-Monde compared to already existing datasets has to be evaluated and current work at Météo-
France is investigating its quality against state of the art reanalyses such as those from NASA at
       either global scale (GLDAS, MERRA-2, The Modern-Era Retrospective Analysis for Research and
       Applications, Version 2, Reichle et al., 2017, Draper et al., 2018) or regional scale (NCALDAS over
       the continental USA, FLDAS over Africa). Finally, first attempts to go to higher spatial resolution
       over smaller areas like the AROME domain (Applications de la Recherche à l'Opérationnel à Méso-
Echelle, https://www.umr-cnrm.fr/spip.php?article120, last access July 2019) of Météo-France
       (centred over France) at kilometre scale and assimilating kilometric and sub-kilometric scale
       satellite retrieval of SSM and LAI (from CGLS) are very promising.

**Code availability.** LDAS-Monde is a part of the ISBA land surface model and is available as open
       source via the surface modelling platform called SURFEX. SURFEX can be downloaded freely at
       http: //www.umr-cnrm.fr/surfex/ using a CECILL-C Licence (a French equivalent to the L-GPL
       licence; http://www.cecill.info/licences/Licence_CeCILL-C_V1-en.txt). It is updated at a relatively
       low frequency (every 3 to 6 months). If more frequent updates are needed, or if what is required is
not in Open-SURFEX (DrHOOK, FA/LFI formats, GAUSSIAN grid), you are invited to follow the
       procedure to get a SVN account and to access real-time modifications of the code (see the
       instructions at the first link). The developments presented in this study stemmed on SURFEX
       version 8.1. LDAS-Monde technical documentation and contact point are freely available at: https://
       opensource.umr-cnrm.fr/projects/openldasmonde/files


       **Data availability:** upon request by contacting the corresponding author.

       **Author Contributions:** Conceptualization, CA, JCC.; Investigation, CA, YZ, SM, NRF;
       Methodology, CA; Writing—original draft, CA; Writing—review and editing, All

       **Funding:** This research was funded by IRT Antoine de Saint-Exupéry Foundation, grant number
CDT-R056-L00-T00 (POMME-V project), the Climate Change Initiative Programme Extension,
       Phase 1 - Climate Modeling User Group ESA/contract No 4000125156/18/I-NB





.

**Acknowledgments:** Results were generated using the Copernicus Climate Change Service Information, 2017. The Authors would like to thanks the Copernicus Global Land Service for providing the satellite derived Leaf Area Index and Surface Soil Moisture.


**Conflicts of Interest: The authors declare no conflict of interest**



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

| Region name | abbreviation | LON-W | LON-E | LAT-S | LAT-N | Number of monthly SSM anomalies below -1 (above 1) stdev | Number of monthly LAI anomalies below -1 (above 1) stdev |
|---|---|---|---|---|---|---|---|
| **Western-Europe** | **WEUR** | **-1** | **15** | **48** | **55** | **5(1)** | **5(0)** |
| Western Mediterranean | WMED | -10 | 15 | 35 | 45 | 0(7) | 4(4) |
| Eastern Europe | EEUR | 15 | 30 | 45 | 55 | 2(1) | 0(2) |
| Balkans | BALK | 15 | 30 | 40 | 45 | 3(3) | 1(4) |
| Western Russia | WRUS | 30 | 60 | 55 | 67 | 0(1) | 1(3) |
| Lower Volga | LVOL | 30 | 60 | 45 | 55 | 2(1) | 2(1) |
| India | INDI | 73 | 85 | 12 | 27 | 3(0) | 2(1) |
| Southwestern China | SWCH | 100 | 110 | 20 | 32 | 0(2) | 0(6) |
| Northern China | NRCH | 110 | 120 | 30 | 40 | 0(3) | 0(4) |
| **Murray-Darling** | **MUDA** | **140** | **150** | **-37** | **-26** | **6(0)** | **7(0)** |
| California | CALF | -125 | -115 | 30 | 42 | 2(0) | 5(0) |
| Southern Plains | SPLN | -110 | -90 | 25 | 37 | 0(3) | 0(4) |
| Midwest | MIDW | -105 | -85 | 37 | 50 | 1(2) | 1(3) |
| Eastern North | ENRT | -85 | -70 | 37 | 50 | 0(3) | 0(7) |
| Nordeste | NDST | -44 | -36 | -20 | -2 | 0(3) | 1(2) |
| Pampas | PAMP | -64 | -58 | -36 | -23 | 2(2) | 2(0) |
| Sahel | SAHL | -18 | 25 | 13 | 19 | 2(0) | 1(2) |
| **East Africa** | **EAFR** | **38** | **51** | **-4** | **12** | **2(3)** | **1(7)** |
| Southern Africa | SAFR | 14 | 26 | -35 | -26 | 2(0) | 2(1) |






Table II: Set up of the experiment used in this study. LDAS_ERA5 and LDAS_HRES have an analysis (assimilation of surface soil moisture, SSM, and leaf area index, LAI) and a model equivalent (open-loop, no assimilation), LDAS_fc4 and LDAS_fc8 are model runs initialized by either LDAS_HRES open-loop or analysis. N/A stands for not applicable.

| Experiments (time period) | Model version | Atmospheric forcing | Domain & spatial resolution | DA method | Assimilated observations | Observations operators | Control variables |
|---|---|---|---|---|---|---|---|
| LDAS_ERA5 (2010 to 2018) | ISBA Multi-layer soil model $CO_2$-responsive version (Interactive vegetation) | ERA5 | Global, ~0.25 °x 0.25° | SEKF | SSM (ASCAT) | Second layer of soil (1-4cm) | Layers of soil 2 to 8 (1-100cm) |
| LDAS_HRES (04/2016 to 12/2018) | | IFS-HRES | North Western Europe (**WEUR**) and Murray-Darling River basin (**MUDA**) (see spatial extend in Table I) ~0.10° x 0.10° | | LAI (GEOV1) | LAI | LAI |
| LDAS_fc4 (2017 to 2018) | | | | N/A | N/A | N/A | N/A |
| LDAS_fc8 (2017 to 2018) | | | | | N/A | N/A | N/A |





Table III: Evaluation datasets and associated metrics used in this study.

| Datasets used for the evaluation | Source | Metrics associated |
|---|---|---|
| In situ measurements of soil moisture (ISMN Dorigo et al., 2011, 2015) | https://ismn.geo.tuwien.ac.at/en/ | R, unbiased RMSD and bias |
| In situ measurements of river discharge | See Table S1 | Nash Efficiency (NSE), Normalized Information Contribution (NIC) based on NSE, |
| In situ measurements of evapotranspiration (FLUXNET-2015) | http://fluxnet.fluxdata.org/data/fluxnet2015-dataset/ | R, unbiased RMSD, Bias, NIC on R values |
| Satellite derived surface soil wetness index (ASCAT, Wagner et al., 1999, Bartalis et al., 2007) | http://land.copernicus.eu/global/ | R and RMSD |
| Satellite derived Leaf Area Index (GEOV1, Baret et al., 2013) | http://land.copernicus.eu/global/ | R and RMSD |
| Satellite-driven model estimates of land evapotranspiration (GLEAM, Martens et al., 2017) | http://www.gleam.eu | R and RMSD |
| Upscaled estimates of Gross Primary Production (GPP, Jung et al., 2017) | https://www.bgc-jenna.mpg.de/geodb/projects/Home.php | R and RMSD |
| Solar Induced Fluorescence (SIF) from GOME-2 (Munro et al., 2006, Joiner et al., 2016) | See references | R |
| Interactive Multi-sensor Snow and Ice Mapping System (or IMS) snow cover | https://www.natice.noaa.gov/ims/ | Differences |


**Figures**


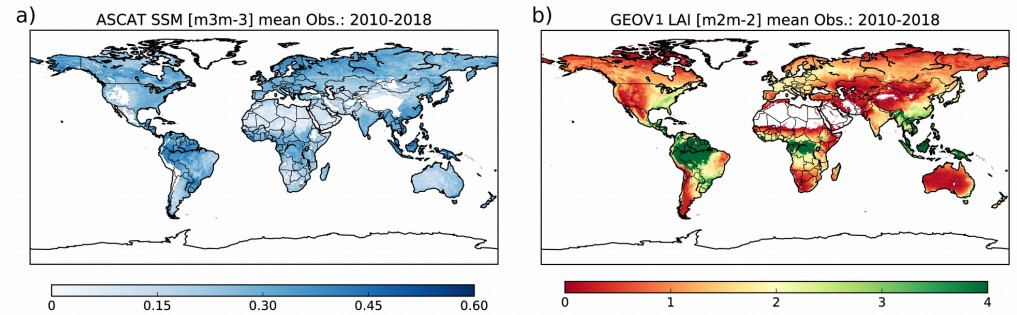

*Figure 1: (a) Surface soil moisture (SSM) from the Copernicus Global Land Service (CGLS) for pixels with less than 15% of urban areas and with an elevation of less than 1500 m above sea level, (b) GEOV1 leaf area index (LAI) from CGLS, for pixels covered by more than 90 % of vegetation, averaged over 2010 to 2018. SSM is obtained after rescaling the ASCAT Soil Wetness Index (SWI) to model climatology, see Section 2.3.*

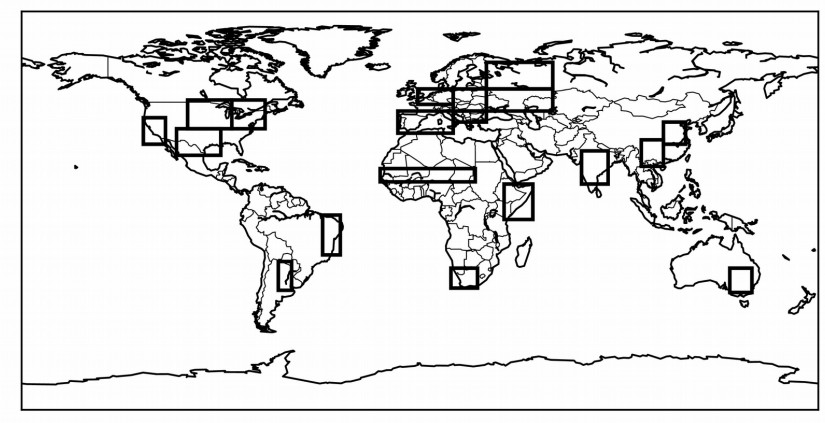

*Figure 2: Selection of 19 regions across the globe known for being potential hot spots for droughts and heat waves, see section on experimental setup.*




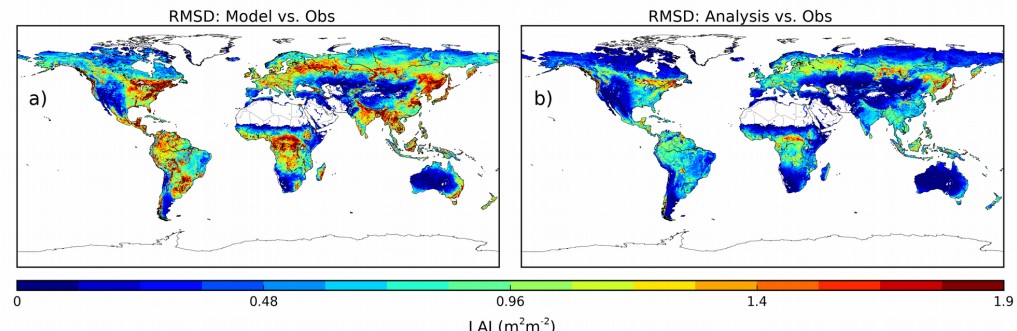

*Figure 3: RMSD values between observed Leaf Area Index (LAI) and LDAS_ERA5 (a) before assimilation and (b) after assimilation of surface soil moisture (SSM) and LAI.*




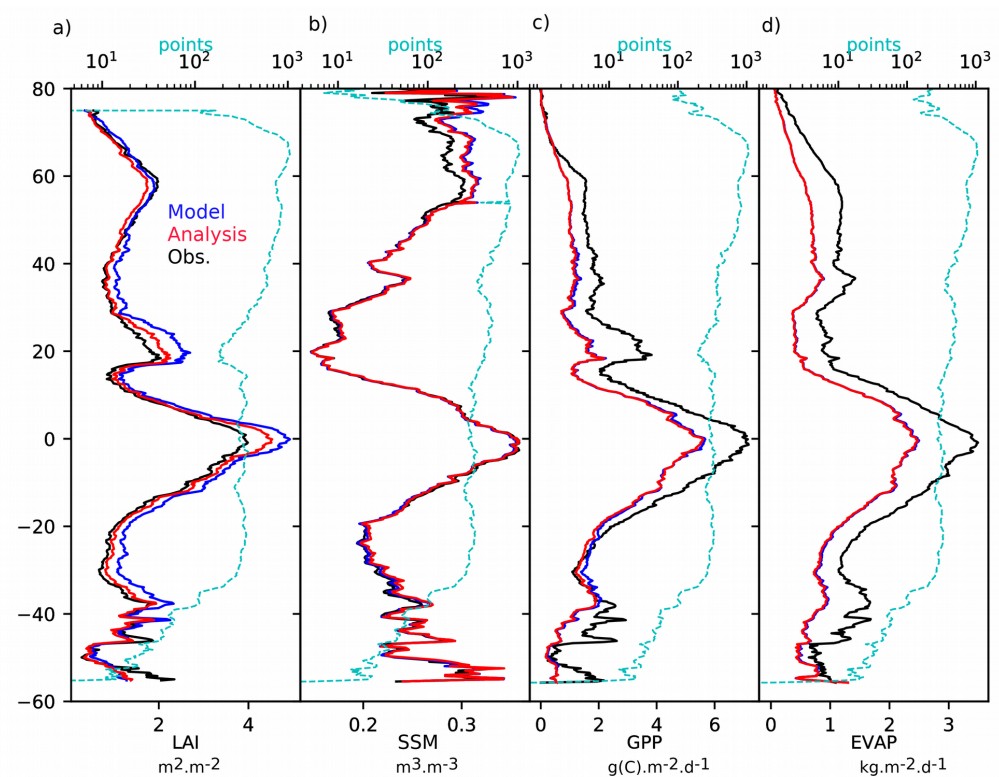

*Figure 4: Latitudinal plots of (a) Leaf Area Index (LAI), (b) Surface Soil Moisture (SSM), (c) Gross Primary Production (GPP) and (d) Evapotranspiration for LDAS_ERA5 before assimilation (Model, blue solid line) and after assimilation (Analysis, red solid line) as well as observations (black solid line). Cyan dashed line represents the number of points considered per latitudinal stripes of 0.25°.*



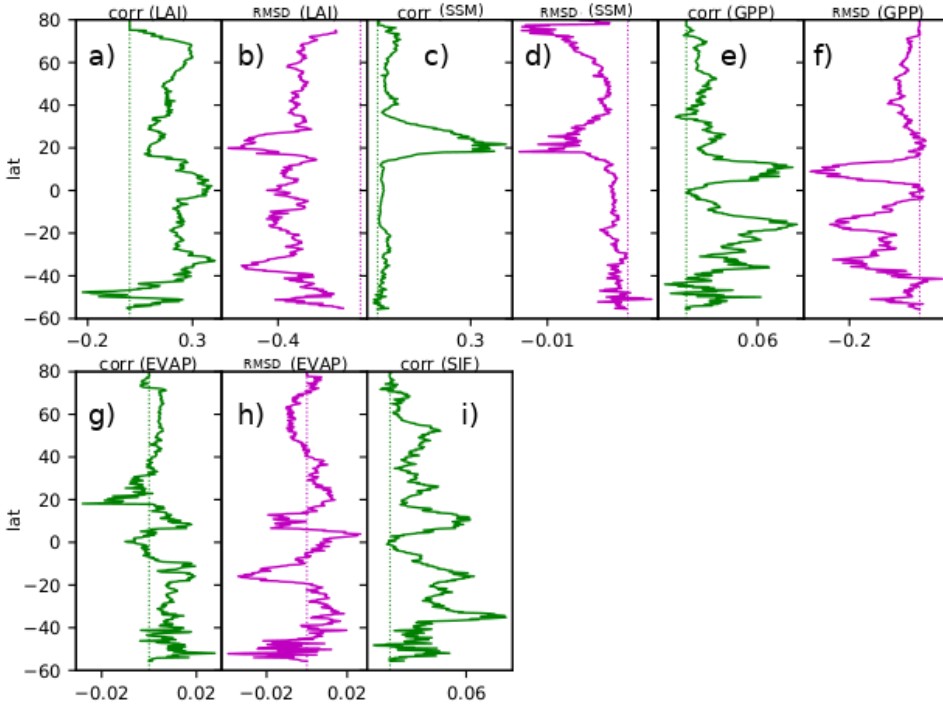

*Figure 5: Latitudinal plots of score differences (analysis minus open-loop) for a) correlation, b) RMSD for Leaf Area Index (LAI), c) correlation ,d) RMSD for Surface Soil Moisture (SSM 1-4 cm), e) correlation ,f) RMSD for Gross Primary Production (GPP), g) correlation, h) RMSD for evapotranspiration (EVAP) and I) correlation for Sun-Induced Fluorescence (SIF). Scores were computed based on monthly average over 2010-2018 for LAI and SSM, 2010-2013 for GPP, 2010-2016 for EVA and 2010-2015 for SIF. For SIF only differences in correlation are represented. Dashed lines represent the zero lines (equal scores for open-loop and analysis).*



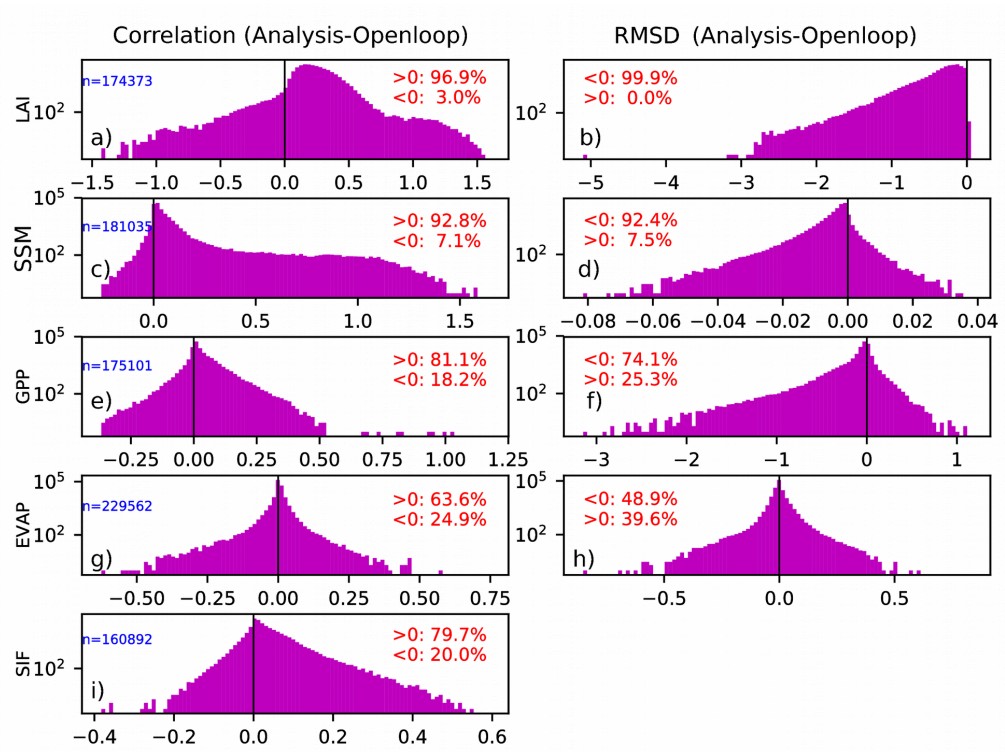

*Figure 6: Histograms of score differences (correlation and RMSD, analysis minus open-loop) for a),b) Leaf Area Index (LAI), c),d) Surface Soil Moisture (SSM 1-4 cm), e),f) Gross Primary Production (GPP), g),h) evapotranspiration (EVAP) and i) Sun-Induced Fluorescence (SIF). Scores were computed based on monthly average over 2010-2018 for LAI and SSM, 2010-2013 for GPP, 2010-2016 for EVAP and 2010-2015 for SIF. For SIF only differences in correlation are represented. Number of available data (in blue) as well as the percentage of positive and negative values (in red) are reported. Note that for sake of clarity, the y-axis is logarithmic.*








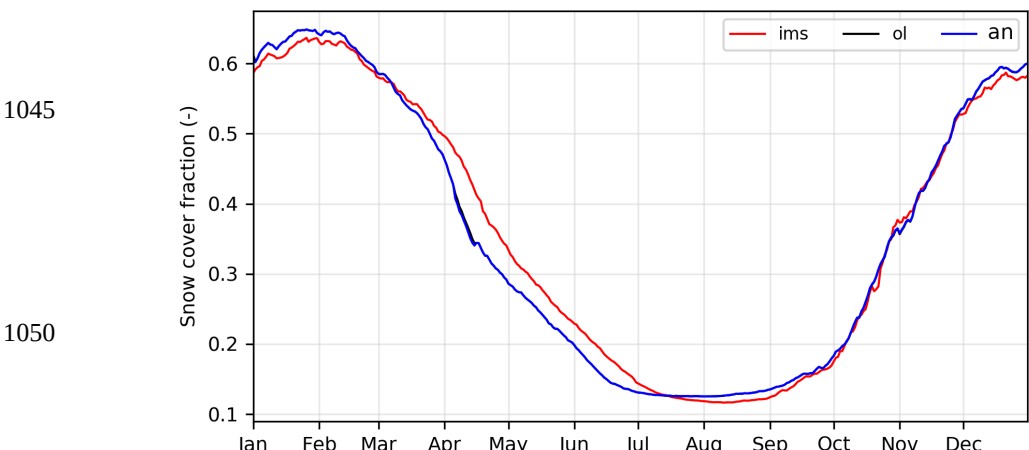


*Figure 7: Mean northern hemisphere snow cover fraction over 2010-2018 for the Interactive Multi-sensor Snow and Ice Mapping System data (ims, in red), LDAS-Monde open-loop (ol, in black) and analysis (an, in blue) forced by ERA-5 atmospheric reanalysis.*




*Figure 8: a), b) and c) Maps of snow cover fraction from the Interactive Multi-sensor Snow and Ice Mapping System data (ims) for September-October-November (SON), December-January-February (DJF) and March-April-May (MAM), respectively for the period 2010-2018. d), e) and f) same as a), b) and c) for LDAS-Monde open-loop (ol) forced by ERA-5. g), h) and i) maps of snow cover differences, ims-ol for SON, DJF and MAM, respectively, for the 2010-2018 period. j), k) and l), same as g), h) and I) between LDAS-Monde analysis and open-loop (an-ol).*






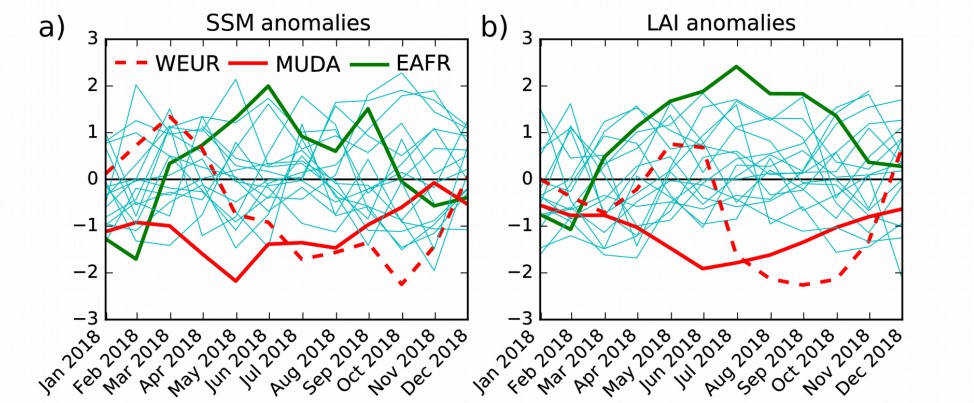

*Figure 9: 2018 monthly anomalies scaled by standard deviation of analysed (a) Surface Soil Moisture (SSM, 1-4 cm) and (b) Leaf Area Index (LAI), with respect to 2010-2018, for the 19 regions presented in Table 1 and Figure 2.*





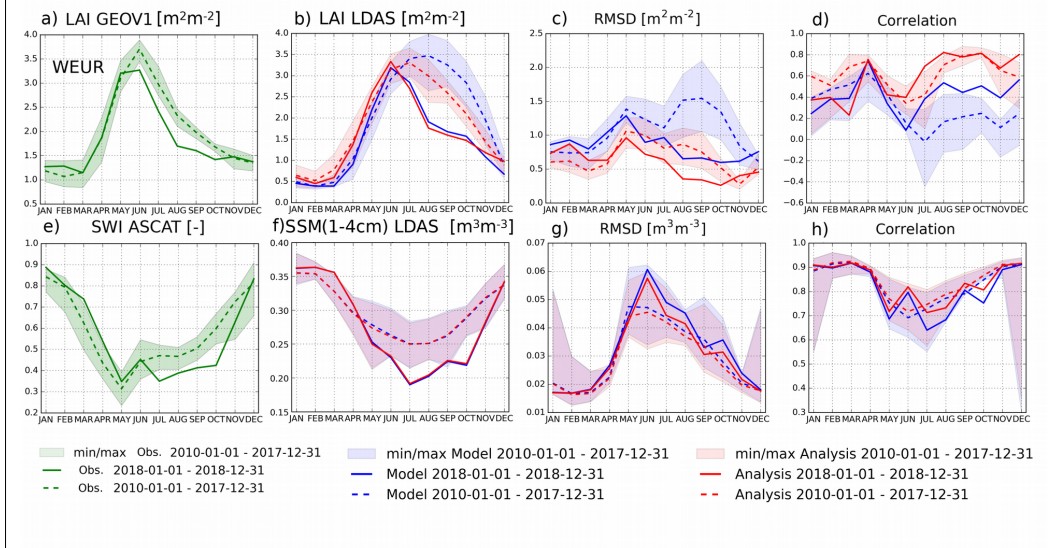

*Figure 10: Seasonal cycles of a) observed Leaf Area Index (LAI) from the Copernicus Global Land Service (GEOV1, CGLS), b) LAI from the open-loop (in blue) and the analysis (in red), c) LAI RMSD values between either the open-loop or the analysis and the LAI GEOV1 for the Western Europe (WEUR) area (see Table I for geographical extent). d) same as (c) for correlation values. e), ASCAT Soil Wetness Index (SWI) from CGLS, f), g) and h) same as b), c) and d) for Surface Soil Moisture (SSM). Note that in g) and h) ASCAT SWI has been converted to SSM using the CDF-matching technique discussed in section 2.3 on assimilated Earth Observations dataset. For each panels dashed line represents the averaged over 2010-2017 along with the minimum and maximum values, the solid line is for year 2018.*






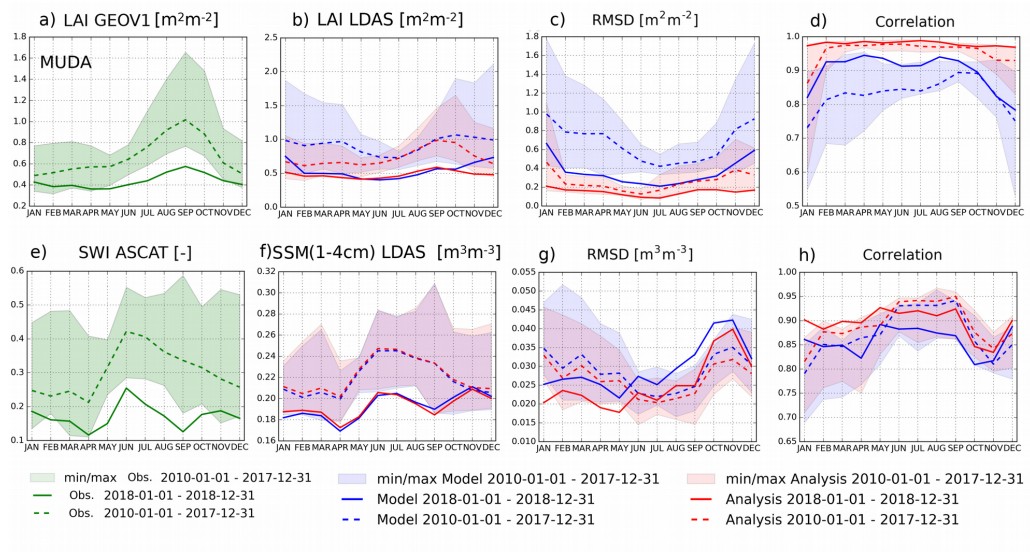

*Figure 11: Same as Figure 10 for the Murray-Darling river (MUDA) area in south eastern Australia.*



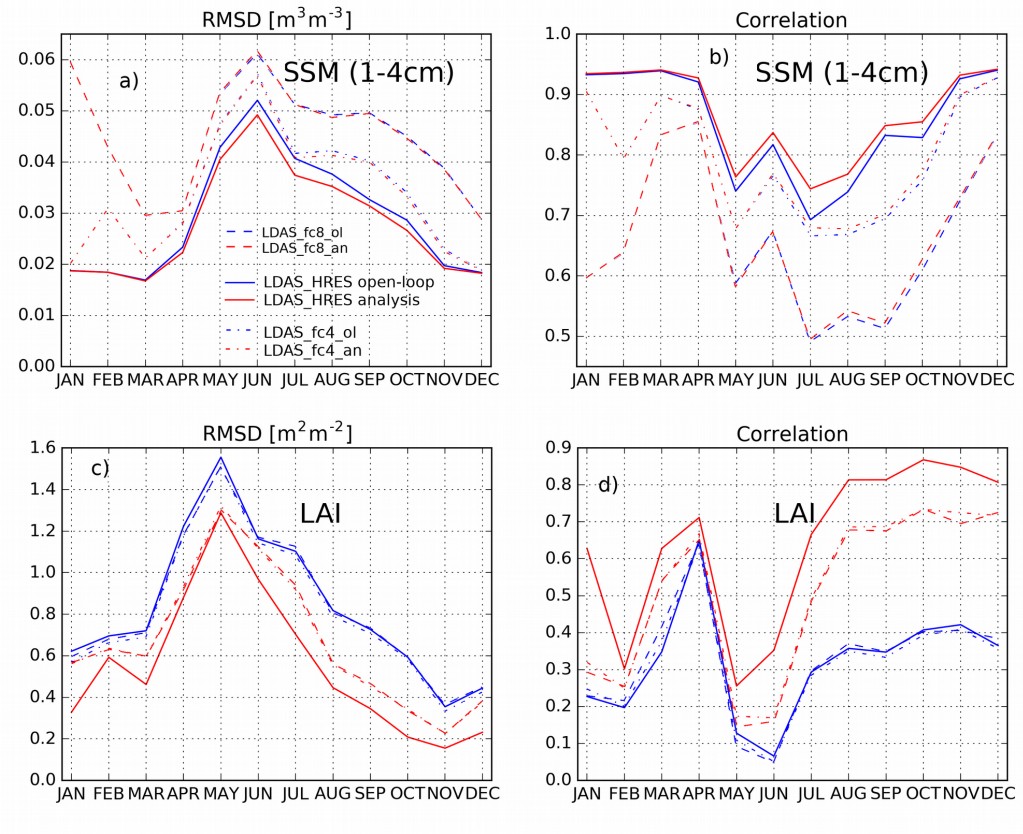

*Figure 12: Upper panel, seasonal (a) root mean square differences (RMSD) and (b) correlation values between surface soil moisture (SSM) from the second layer of soil (1–4 cm) from the model forced by HRES (LDAS_HRES, open-loop in blue solid line, analysis in red solid line) and ASCAT SSM estimates from the Copernicus Global Land Service project over 2017-2018 over the WEUR area. Scores between SSM from the second layer of soil of LDAS_HRES 4-day (dashed/dotted blue – when initialised by the open-loop- and red – when initialised by the analysis- lines) and 8-day (dashed blue and red lines) forecasts and ASCAT SSM estimates are also reported. Lower panel (c) and (d) , same as upper panel between modeled/analyzed Leaf Area index (LAI)  and GEOV1 LAI estimates from the Copernicus Global Land Service project.*



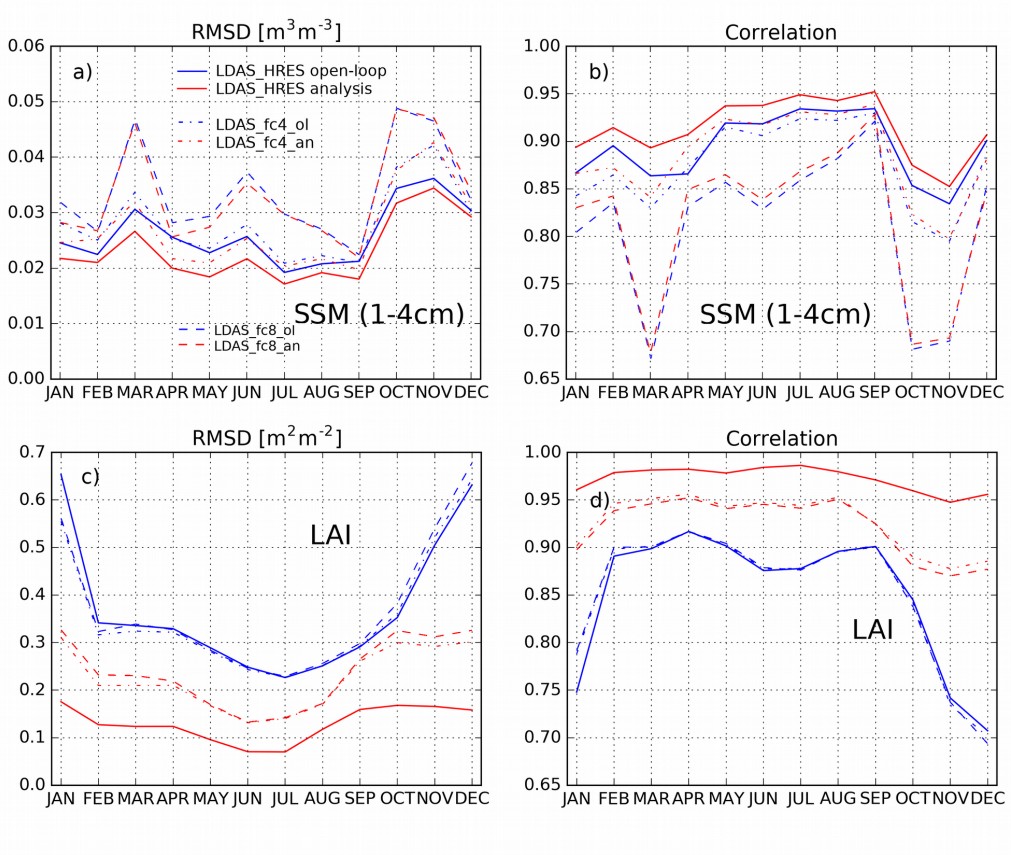

*Figure 13: Same as Figure 12 for the Murray-Darling river (MUDA) area in southeastern Australia.*





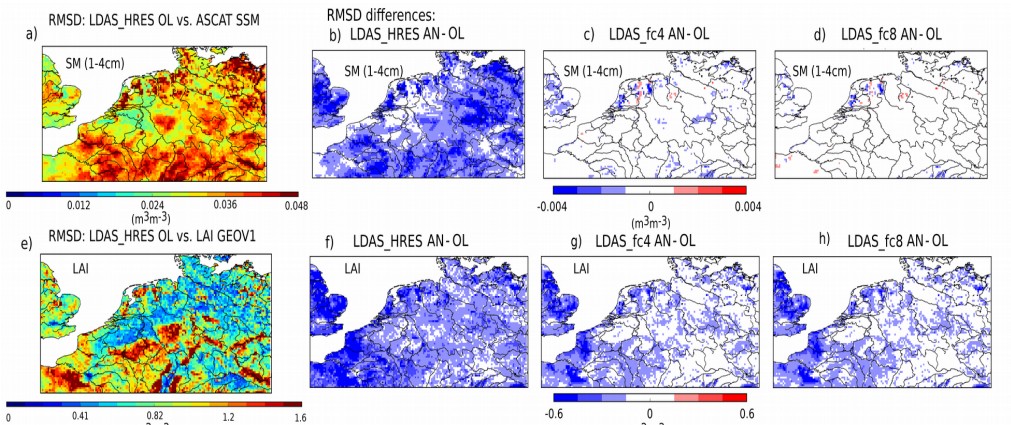

*Figure 14: Top row, (a) RMSD values between LDAS_HRES open-loop and ASCAT SSM estimates from the Copernicus Global Land Service (CGLS) over 2017-2018 for the WEUR domain, (b) RMSD differences between LDAS_HRES analysis (open-loop) and ASCAT SSM. (c), (d) and (e) Same as (b) between LDAS_fc4 initialised by the analysis (open-loop) and LDAS_fc8. Bottom row, same as top row for Leaf Area Index (LAI) from the different experiments and LAI GEOV1.*





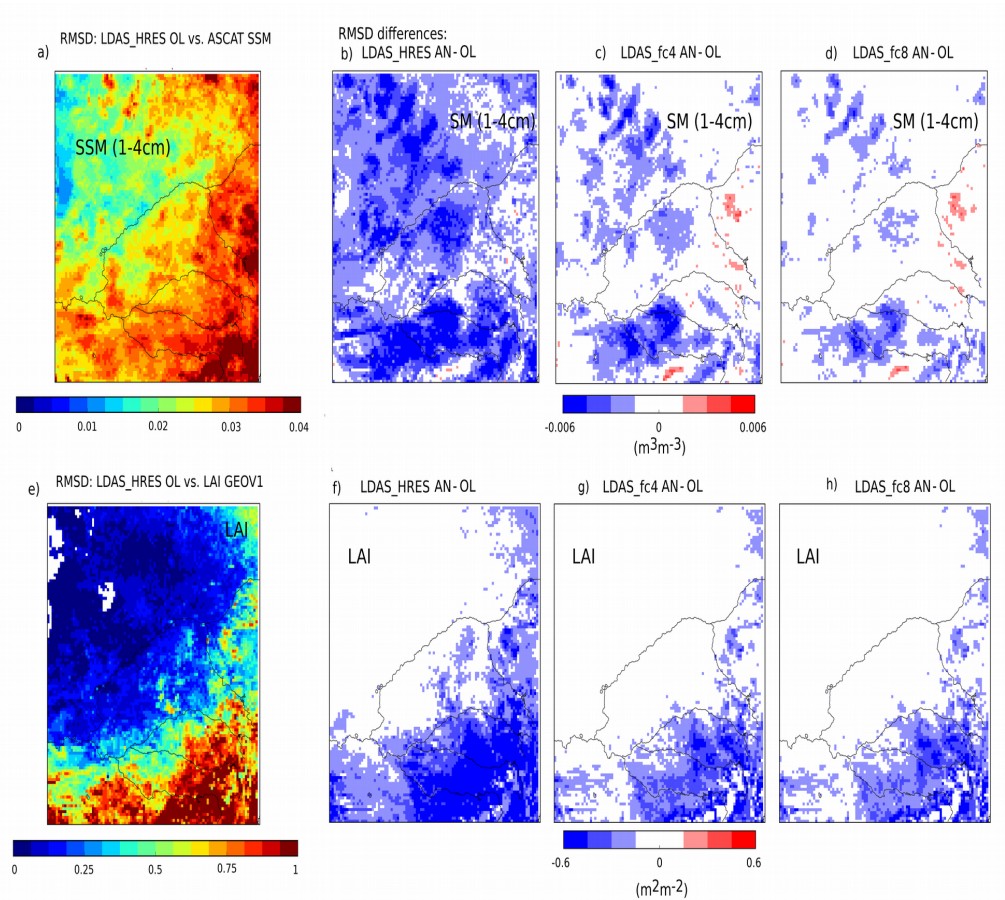

*Figure 15: Same as Figure 14 or the Murray-Darling river (MUDA) area in southeastern Australia.*