# Peer review of "Data assimilation for continuous global assessment of severe conditions over terrestrial surfaces"

_Hydrology and Earth System Sciences, 2019_

## Referee Comment (RC1) · Anonymous Referee #1 · 21 Nov 2019

General comments:

The paper by Albergel et al., describes the application of the LDAS-Monde land data assimilation system (LDAS) to monitor and forecast the impact of extreme events (droughts) on land surface variables, such as surface soil moisture (SSM) and leaf area index (LAI). More specific, the study compares the skill of the open-loop (model only) to that of the analysis (assimilating satellite derived SSM and LAI). This is done globally (to detect anomalies) and at regional scale (using two case studies) for a more comprehensive look at the potential to forecast extreme events. The experiments are validated using different satellite based datasets and also in situ observations. In ad-

dition the authors look at the potential to include snow in the LDAS-Monde data assimilation system. The paper is well written and organized. Furthermore, the paper demonstrates the challenges and opportunities in setting up a global LDAS. It would have been interesting to see a comparison of analysis vs. open-loop root-zone soil moisture skill (compared to the International Soil Moisture Network), as this could have a longer memory than the surface zone soil moisture, however, this is not crucial for the conclusions of this study. The authors are to be congratulated on setting up a global "near real time" LDAS for monitoring and forecasting of land surface variables and I will recommend that this manuscript is published in HESS after considering the following minor comments:

Specific comments:

L107-117: Is it necessary to include such details about the datasets in the introduction?

L180: Please specify what you mean by flow dependency between the prognostic variables and the observations.

L198-200: Difficult to interpret the difference in LAI error when you use a mix of percent and m2/m2. Please could you clarify this?

L251: Could you please include why you don't consider assimilating surface soil moisture observations from the Soil Moisture and Ocean Salinity (SMOS) and/or from the Soil Moisture Active Passive (SMAP) satellite missions? As these satellites are expected to be more sensitive to surface soil moisture than the C-band observations from ASCAT. Furthermore, as I understand ASCAT data are already assimilated in the production of the ERA5 dataset. Will the LDAS-Monde assimilation not lead to a Âńdouble counting or usage of the ASCAT data and what are the potential consequences for your analyses results?

L268: Please could you specify the difference between linear rescaling and CDF-matching (if any)? To my understanding linear rescaling is correction of the mean

and standard deviation, while CDF-matching corrects the whole CDF (i.e., all moments of the probability distribution function), hence linear rescaling is not the same as CDF-matching.

L292-294: Could you please discuss how this short spinup period could affect your results?

L383: Could you please provide more details on how this can explain the differences seen between ISBA and GLEAM?

Technical corrections:

L18: Acronym ERA5 is not defined. What is LDAS_ERA5? An experiment?

L23: Please remove "successfully"

L25: Evapotranspiration and evapotranspiration, please decide on capital letter or not.

L54: "However its concept is broader" This is a bit unclear, maybe: "The concept of drought is broad and they are generally....".

L56: This was a bit unclear, maybe: "and they have severe impacts in regions with rain-fed crops and no irrigation".

L61: Suggestion: "that all drought types..."

L63: Suggestion: "in order to fully understand the...."

L64: Acronym is already defined in the abstract.

L69: Earlier Earth observations now Earth Observations, define acronym earlier.

L73: Please specify what application? Coupling of LSVs with other models of the Earth system? And how is this done?

L75: Please remove "unrestrictedly"

L77: LDAS acronym already defined in the abstract.

L78: "could lead to..."

L83: "several are NASA lead...."

L84: "Amongst" multiple times? Maybe rephrase to: "Examples of such activities are the Global Land Data Assimilation System (GLDAS) which is run at a global scale. While the...."

L93: "the LDAS-Monde"

L95: Please rephrase. Maybe "Few studies have, however, included..."

L100: "the LDAS-Monde"

L101: ECMWF acronym already defined

L103: This is already specified in the abstract, perhaps remove/rephrase in the abstract to avoid duplication.

L104-106: Acronyms are already defined in the abstract. Maybe cut the definition of the acronyms in the abstract to avoid the abstract to become very technical?

L113: "accessed"

L115: "accessed"

L116: "accessed"

L119: Please remove "severely"

L128: "prospects for future work"

L158: Please specify what kind of parameters.

L159: "The ISBA..."

L161: "bottom"

L167: What is "bale"?

L184: ".....eight soil layer (1-100 cm) and..."

L184: "In addition to a control run..."

L186: "., the initial...."

L193: "(soil layer 3 to 8, 4 – 100 cm)."

L204: Replace "and" with a comma.

L205: Change "precipitations" to "precipitation"

L209: "is a..."

L210: Remove "EU-funded" or be more specific.

L212: Change "dimension" to "resolution"

L213: Please rephrase to avoid the use of "quite new" and "already evaluated". Maybe: "Several studies have validated the ERA5 datasets, for example...."

L214: Remove "former"

L221: ", also with positive outcomes for ERA5."

L226: "this study"? And ", the ERA5...."

L230: "accessed"

L231: "...HRES and 31 km in ERA5".

L231: "The atmospheric forcing..."

L232: For HRES this is not a "reanalysis grid"

L232-234: Please rephrase, what do you mean here?

L250: "accessed"

L256: Remove extra spacing

L256: SWI is already defined.

L264: "the assimilation"

L277: Using model or and observed snow dataset?

L290: "For these regions..."

L292-294: Please rephrase. Maybe: "The HRES is available at the 0.1x0.1 resolution from April 2016, thus we do a spinup covering April to December 2016 and present results for 2017 to 2018."

L295: "used, and initialized..."

L302: Please change wording.

L303: Independent from what? The analysis or each other? Please specify

L304-310: Is defined in the abstract, consider removing this technical information from the abstract.

L311: Remove "also"

L314: "and they have been used in previous studies to...."

L316: "Therefore, the LDAS-Monde ...."

L327: Nash-Sutcliffe

L341: open-loop

L347: Please change "appreciate"

L348: open-loop

Figure 4: Perhaps indicate in the figure caption (or in the figure) which observational dataset is being used for comparison e.g., GLEAM for evap etc.

L356: "averages"

[Figure]

L364: Please rephrase "healthy behaviour"

L367: "latitudes"

L369: Please clarify "decrease in e.g. RMSD values".

Figure 5: Please fix the comma in the caption e.g., "Latitudinal plots of score differences (analysis minus open-loop) for Leaf Area Index (LAI), c) correlation ,d) RMSD...." what is the space after correlation?

L371: "The number..."

Figure 6: Plot title "Correlation (Analysis – openloop)" to "Correlation (Analysis minus open-loop)".

L377: Maybe add "(SIF RMSD N/A)".

L387-393: Very long sentence, please rephrase.

L395: Please remove "latter"

L396: Please clarify that this is down to the ERA5 forcing and not the snow module in ISBA.

L397: Spring to spring

L398: What is it in the snow cover parameterization of ISBA that causes this?

L400: Please make it more clear that it is the LDAS-Monde snow product that could benefit from this and not data assimilation in general.

L407: "accessed"

L409: "The LDAS...."

L413: Well 1/2 of the metrics are positive and the other 1/2 is negative?

L428: Please comment why NSE values below -2 were discarded.

L429: 982 out of how many stations?

L431: "Panel a)...."

L437: "These results..."?

Figure S2 caption: "are available and with a drainage...."

Figure S2: Do you have any idea why the NSE seems to be better over Europe than for example the United States?

Figure S3 caption: The analysis is green in the figure, while the text says that it is red.

L441: "...the SSM analysis."

L447: "averaged R values are...."

L455: Spacing: "(NIC > +3)"

L455: "in" instead of "by"?

L460: Please rephrase "significant differences" can also be negative, hence no added value from the analysis.

Figure 9 caption: Please specify the acronyms and their corresponding color in the figure.

L471: "...through the whole 2018...."

L475: "EAFR experiences.."?

L480: "accessed"

L481: "is already"?

L488: "...Murray Darling..." to follow what is used earlier in the text.

L489: References needed

L495: Please clarify the title "medium resolutions" what? I assume forecasts.

L501: "years"

L511: Please rephrase "one may appreciate..."

L512: "Almost all months...."?

L518: "high resolutions.." of what? Maybe: "Case studies for assessing the LDAS-Monde high resolution (0.1x0.1) model runs."

L539: "poorer" to "lower" and please make it clear what you mean by "... that its first day analysis".

L543: Maybe clarify this by inserting (blue solid line) behind the different model runs. As there is a lot of information from L543 to 546 concerning the different runs.

Figure 14: Please increase the size of this figure, to around the same size as Fig. 15.

L569: "south eastern" in the text and "southeastern" in the Figure 15 caption.

L580: "to force"?

L581: Could you specify what you mean by "fixed configuration"

L581-582: "The ERA5 coarse spatial resolution makes it affordable to run long term and large scale LDAS-Monde experiments".

L583: "available from 1979...."

L583: Please define and explain ERA5T.

L585: "..long term and near real time...."

L587: "focused"?

L587: Change "weather" to "conditions"?

L592: Small detail, but don't you compare to (1-4 cm) and not (0-1 cm)?

L599: "...layers and evapotranspiration..."

L607: Change "than if" to "that the..."

L607: "..., however, it suffers from a too early...."

L635: "accessed"

---

## Referee Comment (RC2) · Anonymous Referee #2 · 30 Nov 2019

GENERAL COMMENTS

This a very interesting and well written paper that describes global and regional-scale experiments with the new global data assimilation scheme (LDAS-Monde) set up at Meteo France. The experiments presented in the paper concern the assimilation of remotely sensed soil moisture and leaf area index. The regional experiments were carried out a finer grid to study extreme events. This paper seems to represent a major milestone in the development of LDAS-Monde (which is in my view a very important undertaking), and hence I recommend publishing the paper after minor revisions.

SPECIFIC COMMENTS

[Figure]

Line 167: What do you mean by "... is bale to ..."?

Lines 188ff: The procedure described here results in an observational error field mainly related to soil properties, while the real retrieval errors are mostly dependent on vegetation density. Please discuss implications.

Line 198: Is "20 %" a relative error?

Section 2.2: Note that ASCAT SSM data are already assimilated in ERA5. Please discuss implications.

Line 248: SWI is the Soil Water Index

Section 2.3: Describe also the masking of SSM

Line 493: Only this sub-study focusses on severe conditions, but not "this study" overall.

---

## Referee Comment (RC3) · Anonymous Referee #3 · 30 Nov 2019

The paper evaluates LDAS-Monde using a range of different data products and illustrates how LDAS-Monde results can be used for applications, such as e.g. drought monitoring. Overall, the LDAS-Monde system is great, but the paper needs a thorough revision. In particular, there are too many 'trivial' results, which are repeatedly highlighted. These need to be removed (i.e. if we assimilate an observation, then obviously, the results will be closer to that observation) in favour of a discussion of the evaluation against independent observations. Further comments are listed below.

Section 2.1.3

- Are the perturbations chosen to get an optimal data assimilation system? Please

discuss.

- How are the cross correlations between the errors in the various soil layers defined, and the error correlations between LAI and soil moisture?

Section 2.3

- ASCAT has an approximate resolution of 25 km. How are these coarse data assimilated/downscaled into the 0.1˚o model simulations?

- 'CDF matching ' refers to rescaling of the entire CDF, and is not a correct terminology when only rescaling the mean and variance.

- How exactly are the LAI data 'interpolated' from 1 km to 0.25 degree? Do you mean interpolation to bridge cloudy pixels and then aggregation (upscaling)?

- Is the LAI also 'converted from the observation space to the model space' as is done for soil moisture? Please describe how? If there is no such rescaling, then the results may be trivial, i.e there will be more impact of a non-rescaled LAI assimilation than when doing a gentle nudging with rescaled soil moisture. However, since you use a KF variant, there probably is some rescaling for both (otherwise the KF assumptions would be violated).

Section 2.4

- How exactly is the 'climatology' defined? Is it seasonally varying, how much smoothing is applied, etc?

- The spinup period for the 0.1˚o simulation seems unrealistically short. How was it initialized? Could you cycle over the short April-December period multiple times?

- Table II: An observation operator is a function, not a variable; also explain what you mean by control variable (updated variables) for readers who are new to the field. In fact, the control vector enters the observation operator, which in turn selects a subset of relevant variables to produce the observation prediction.

Section 2.5

- Why is there no skill evaluation in terms of anomalies? Would be interesting.

- Which variable in LDAS-Monde output is related to SIF and how?

Section 3

- Overall, it is a bit disconcerting that trivial design results are shown repeatedly. Assimilate a variable, and sure, the model will get closer the assimilated observations. The results need to be thoroughly revised (both text and figures) to eliminate the trivial results. They can be mentioned once, but then the focus needs to be on the independent evaluation. It is also not correct to say that results "improve" if they simply get closer to the assimilated observations (e.g. L. 375, L. 516, . . .). This holds both for the global assessment and for the case studies, e.g. all of L. 505-512 is 'trivial' and can be removed.

- The snow cover results (Fig 7-8) can be removed. It is too trivial that there would be no impact on snow cover by assimilating soil moisture or LAI. Or else, explain in detail how either variable would affect the snow cover.

- The independent validation (e.g against in situ SSM) shows no substantial improvement in any of the metrics due to data assimilation. Have the in situ data been thoroughly filtered to remove bad points? Why exactly do the authors see an advantage of LDAS_ERA5 for these variables relative to the open loop (L. 458)? There is some added value, but there is also significant degradation, i.e. I would say it is an equal game here.

- L. 535 & L. 545: 'more sensitive to' is perhaps not the correct wording? Sensitivity would be quantified by something like the Jacobian. There is simply a larger update in LAI than in SSM by design, and this propagates in time differently due to the difference in memory for both variables (at this point in the paper, I am actually suspecting that LAI is assimilated with a bias, see comment above).

[Figure]

- Could you evaluate the impact of LAI and SSM assimilation in terms of runoff for the high-resolution simulation?

The text needs careful revision. There are too many incorrect sentences, it is annoying. Here are only some random examples:

L. 104: "this... are": rewrite sentence

L. 168: "system is bale...": what does this mean?

L. 186: "1%": of what?

L. 191: ", scaled with and 0.02": makes no sense

L. 195 and further is a repetition of the text around L. 188: combine and condense

L. 198: "...is assumed to be 20% and a similar assumption is made...": split sentence. 'Similar', but not identical? Please be precise and say which error is applied.

L. 207: replace 'as well as' by 'or' to avoid the impression that multiple forcing data sets are combined in a way.

L. 281: LDAS-ERA5; later on LDAS_ERA5

L. 412: w.m-2 –> W.m-2

L. 615: The *C*urrent

L. 632: references seem off, GLDAS , MERRA-2?

---

## Referee Comment (RC4) · Anonymous Referee #4 · 7 Dec 2019

The paper primarily discusses the verification of output from the global LDAS-Monda land surface data assimilation system for 2010-2018 at 0.25-degree spatial resolution. Satellite observations of surface soil moisture (SSM) from ASCAT and leaf area index (LAI) from GEOV1 are assimilated into the ISBA land surface model. The verification assesses the skill of the assimilation estimates against the assimilated observations and against independent in situ, model and satellite datasets. The skill of the assimilation estimates is compared to that of a model-only "open-loop" simulation (without data assimilation). Moreover, higher-resolution simulations with and without data assimaltion along with 4-day and 8-day forecasts of land surface conditions are presented for 2017-2018 over two regional domains, and the output was assessed in the context

of extreme land surface conditions. The authors find that the assimilation generally improves over the open-loop simulation, particularly for the more slowly-varying LAI estimates.

The manuscript is obviously of interest to HESS readers. I'm confident that the data assimilation system, the simulations, and the verification results are technically sound. However, I found the presentation of the material to be somewhat lacking. The paper is a bit tedious to read in that a number of results are discussed at length where the assimilation and the open-loop are essentially the same (e.g., for snow variables) or where the improvements from the assimilation are probably not meaningful, at least not in the sense of statistical significance. Perhaps worse still, there is no quantitative information about errors in the assimilation estimates, e.g, confidence intervals or other suitable uncertainty estimates. In my view, the discussion also over-emphasizes the comparison of the assimilation estimates against the assimilated observations, which is not independent verification (as the authors point out clearly in one part of the paper but mostly sweep under the carpet in another part). Finally, I was rather disappointed, given the illustrious author list, with the fairly careless pre-submission proof-reading.

I think the paper can be an important contribution and can eventually be suitable for publication in HESS, but at this point I recommend MAJOR revisions with consideration of the comments below.

Major comments:

1) Six of the thirteen figures that show results (i.e., not counting "data & methods" figures 1 and 2) are about evaluating the skill of the assimilation estimates *exclusively* against the assimilated observations (Figs 3 and 9-13). Comparisons against the assimilated observations are also included in Figs 4-6 (along with other variables) and Figs 14-15 (along with forecast estimates of SSM and LAI). While I agree that it is important to verify that the assimilation system works as intended, the authors over-emphasize the comparison against the assimilated observations.

2) The two figures about snow (Figs 7 & 8) could be simplified considerably because there is no meaningful difference between the assimilation estimates and the open-loop estimates, which is a rather trivial result (as the authors discuss).

3) There are no graphics in the main text (only in the supplement) about the validation of the results against *independent* in situ measurements (section 3.1.2). This independent validation should be reflected more prominently in the main paper.

4) The claim about "improvement" of the assimilation estimates vs. the open-loop estimates from the independent validation against in situ soil moisture estimates in section 3.1.2 (∼line 460) is on shaky footing. For none of the networks listed in Table S3 is there a difference of more then 0.02 in the R values between the assimilation and the open loop. In some cases, the 0.02 difference is negative (ie., degradation). For most networks the R difference is 0 or 0.01, that is, there really isn't a meaningful change. Here, and also for at least the other in-situ based results, it is imperative that the authors provide some estimates of whether the differences are meaningful (e.g., by including statistical confidence intervals), and then honestly discuss the results. The claim in line 460 about significant improvements at some sites may be true, but given the network-average neutral results there must then also be sites with a significant degradation, which is not mentioned in the paper.

5) The editing of the paper is rather careless. There are many small mistakes, and the organization of the text is lacking. Examples include the following:

a) The Introduction lacks a clear statement of the paper's objectives. The text in Lines 107-121 simply states what will be presented (with lots of references and details). It's hard to tell what the objectives might be.

b) There are several instances in the Results section of text that belongs in the Methods section, incl: Lines 384-387 - IMS snow cover product description Lines 405-409 - Fluxnet description Lines 440-447 - ISMN description

c) Section 3.2.2 is a *single* paragraph that stretches over nearly two pages. Really? There are several other paragraphs of excessive length.

d) Graphics:

Figure 1a: Use different color for zero values and no-data value. (currently, both are white, making it unclear whether there are data in, e.g., the western US, or whether those are screened, perhaps because of topography.

Figure 3: The label of the colorbar should read "RMSD of LAI [m2 m-2]", not just "LAI [m2 m-2]"

Figure 5: Units are missing for RMSD panels. (This is particularly important because this information is needed to judge whether the differences are in fact meaningful.)

Figure 6: Three panels only have a single tick & tick label on the y-axis. At least two are required to interpret the axis scale.

Figure 7: The color choices should be made consistent with Fig 4.

Figure 9: I could not find out what the thin cyan lines depict.

Figures 10+11: add "LAI" to plot title of c) and d); add "SSM" to plot title of g) and h)

Figure S2: NSE should vary from -infinity to 1. The colorbar is from -20 to 20, and darker blue values would clearly be greater than 1. Either the colorbar is wrong or the values show something other than NSE.

Table S3: The column headings on the 2nd page of the table still include French words.

6) In section 3.2.2, the authors no longer make it clear that the verification is against the assimilated datasets. While verification of forecast data against the assimilated dataset can be viewed as independent validation because the verification data have not (yet) been assimilated, there is an important distinction here between SSM and LAI. For SSM, the assimilation is done after rescaling (cdf-matching), which removes bias. For

[Figure]

LAI, however, the assimilation uses the raw LAI observations (I think). That is, the assimilation removes bias in the modeled LAI (w.r.t. the observed LAI). This technical difference between SSM and LAI assimilation, combined with the longer memory of LAI compared to SSM, should contribute to the results in section 3.1.2. Put differently, the LAI results of section 3.1.2 are not likely to hold if an independent LAI dataset had been used for validation that is itself biased against the assimilated LAI observations. (Different LAI datasets may not be as biased against each other as typical satellite SSM datasets, but there are considerable biases between LAI products.)

7) Figure 3c suggests that the change in GPP is negligible, at least in the zonal mean sense although Figure 4f suggests that GPP does change in terms of RMSD. Given the considerable change in the (zonal mean) LAI (Fig 3a), I would have expected a lot more change in the mean GPP. I suspect that the disconnect between the LAI and GPP changes is rooted in how these variables are connected in ISBA and how exactly the assimilation system goes about updating LAI. This rather counter-intuitive result requires clarification in the paper.

8) Fig 5h: The changes in EVAP are with +/- 0.02 (mm/d???). If my guess about the units is correct, this would amount to only a few mm per year, which is well within the uncertainty of in situ measurements. That is, the EVAP changes are not likely to be meaningful in a practical sense. This should be discussed more explicitly.

Minor comments:

9) Line 167: typo "bale" –> "able"

10) Line 209: "fifth generation of European reanalyses produced by ECMWF" I recommend phrasing this differently to avoid the misunderstanding that the reanalyses are just for the European domain. E.g.,: "fifth generation of global reanalyses produced by ECWMF"

11) Lines 293-295: How did you address the heterogeneity within the 0.25-deg grid

cells during spin-up? It is not obvious that the short spin-up period from April 2016 suffices for properly spinning up grid cells with strong heterogeneity at the sub-0.25-degree scale.

12) Line 379: Do you mean a decrease in RMSD or a decrease in skill?

13) Line 412: If I'm reading this correctly RMSD decreases while both bias and ubRMSD increase. This is quite counter-intuitive and requires a rather odd distribution of the metrics across the sites or networks included in the average. In any case, since bias and ubRMSD get worse, I do not think that the statement about "a small advantage of the analysis over the open-loop" is justified.

14) Line 429: "NSE values below -2 were discarded" requires a justification, otherwise it reads like cherry-picking.

15) Line 535: "the analysis is of better quality" Given the numbers, I see at best "slightly better quality"

16) Line 592: "surface (0-1 cm)" In section 3.2.2 the discussion was about the "(1-4 cm)" layer. Which is it?

---

## Author Comment (AC1) · 22 Jan 2020

Dear Reviewer 1 many thanks for reviewing the manuscript and for highlighting its relevance and interest. Your comments and suggestions led to an improved version of the manuscript. Attached is a point by point answer to your specific comments, all your editorial and technical comments were accounted for in the revised version of the manuscript.

Response to Reviewer 1 are structured as follow: (1) 1.X: comments from Reviewer 1, (2) Response to 1.X: author's response and author's changes in manuscript when any. For sake of clarity, line and page numbering from the revised version are used.

[Figure]

Please also note the supplement to this comment:
https://www.hydrol-earth-syst-sci-discuss.net/hess-2019-534/hess-2019-534-AC1-
supplement.pdf

**Supplement:**

Response to Reviewer 1 are structured as follow: (1) 1.X: comments from Reviewer 1, (2) Response to 1.X: author's response and author's changes in manuscript when any. For sake of clarity, line and page numbering from the revised version are used.

**Reviewer#1**

Dear Reviewer#1 many thanks for reviewing the manuscript and for highlighting its relevance and interest. Your comments and suggestions led to an improved version of the manuscript. Below is a point by point answer to your specific comments, all your editorial and technical comments were accounted for in the revised version of the manuscript.

**1.1 [It would have been interesting to see a comparison of analysis vs. open-loop root-zone soil moisture skill (compared to the International Soil Moisture Network), as this could have a longer memory than the surface zone soil moisture, however, this is not crucial for the conclusions of this study.]**

**Response to 1.1**

Thank your for your highly relevant comment. Following it and similar comments from the other Reviewers, it has been decided to revisit the soil moisture evaluation part of the study:
(1) we have added an evaluation of soil moisture from LDAS-Monde fourth layer of soil (10 to 20 cm) against in situ measurements of soil moisture at 20 cm depth when available (10 networks and 685 stations),
(2) for surface soil moisture (SSM), correlation values (R) were calculated for both absolute and anomaly time-series in order to remove the strong impact from the SSM seasonal cycle on this specific metric,
(3) a 95% Confidence Interval (CI) has been added to R values.
(4) we have added the number of stations for which correlations differences are significant (significant improvement or degradation from the analysis) as well as a map over North America for illustration.

It involves several changes in the revised version of the manuscript, they are listed below.

Methodology section, 2.5 Evaluation datasets and metrics

P.11, Lines 358-365: "In situ measurements of surface soil moisture from 19 networks across 14 countries available from the ISMN are also used to evaluate the performance of the soil moisture analysis. They represent 782 stations with at least 2 years of daily data over 2010-2018. Sensors at 5 cm depth (SSM) are compared with soil moisture from LDAS_ERA5 third layer of soil (4-10 cm), sensors at 20 cm depth with the fourth layer of soil (10-20 cm, 685 stations from 10 networks). Beside 11 stations located in 4 countries of Western Africa (Benin, Mali, Sénégal and Niger) and 21 stations in Australia, most stations are located in North America and Europe, see Table S3."

P.12, Lines 374-377: "For global estimates, Normalized RMSD (NRMSD, Eq.(2)) was used, also. Finally, for surface soil moisture, R was calculated for both absolute and anomaly time-series in order to remove the strong impact from the SSM seasonal cycle on this specific metric (see e.g. Albergel et al., 2018a, 2018b)."

Result section, 3..1.2 Ground-based datasets

P.17-18, Lines 548-580: "The statistical scores for soil moisture from LDAS_ERA5 open-loop and analysis (third and fourth layers of soil, 4-10 cm depth, 10-20 cm depth, respectively) over 2010-

2018 when compared with ground measurements from the ISMN (5 cm depth and 20 cm depth) are presented in Table S2 for each individual network. Averaged statistical metrics (ubRMSD, R, $R_{anomaly}$ and bias) are similar for both LDAS_ERA5 analysis and open-loop even if local differences exist. For the analysis, averaged R ($R_{anomaly}$) values along with its 95% Confidence Interval (CI) using in situ measurements at 5 cm (782 stations from 19 networks) are 0.68±0.03 (0.53±0.04) (0.67±0.03(0.53±0.04) for the open-loop) with averaged-network values going up to 0.88±0.01 (0.58±0.04) for the analysis (SOILSCAPE network, 49 stations in the USA) and always higher than 0.55 except for one network, ARM (10 stations in the USA) presenting an averaged R value of 0.29±0.05. Averaged ubRMSD and bias (LDAS_ERA5 minus in situ) are 0.060 $m^3m^{-3}$ and 0.077 $m^3m^{-3}$ for the analysis, 0.060 $m^3m^{-3}$ and 0.076 $m^3m^{-3}$ for the open-loop, respectively. NIC (Eq.1) has also been applied to R values, 65% of the pool of stations present a neutral impact from the analysis (511 stations at NIC ranging between -3 and +3), 12% present a negative impact (91 stations at NIC < -3) and 23% present a positive impact at (180 stations at NIC > +3).

The number of stations where R differences between the analysis and the openloop are significant (i.e. their 95% CI are not overlapping) is 186 out of 782 (about 26%). There is an improvement from the analysis w.r.t. the openloop for 128 stations (out of 186, i.e. about 69%) and a degradation for 58 stations (about 31%). Figure 7 illustrates R differences between the analysis and the openloop runs. When differences (analysis minus openloop) are not significant stations are represented by a small dot. When they are significant, large circles have been used, blue for positive differences (an improvement from the analsysis) and red for negative differences (a degradation from the analysis). For most of the stations where a significant difference is obtained, it represent an improvement from the analysis.

Averaged analysis R (95%CI), bias and ubRMSD for the fourth layer of soil (685 stations from 10 networks) are 0.65±0.03, 0.049 $m^3m^{-3}$ and 0.055 $m^3m^{-3}$, respectively. For the open-loop, they are 0.064±0.03, 0.048 $m^3m^{-3}$ and 0.056 $m^3m^{-3}$, respectively. For soil moisture at that depth, about 60% of the stations present a neutral impact from the analysis (410 stations at NIC ranging between -3 and +3), 28% a positive impact (189 stations at NIC > +3) and 12% a negative impact (86 stations at NIC < -3). Although differences between the openloop run and the analysis are rather small, these results underline the added value of the analysis with respect to the model run. Figure S6 represents the distribution of the scores values for LDAS_ERA5 open-loop and analysis using boxplots centred on the median value. They look very similar and from this figure, it is difficult to see either improvement or degradation from the analysis."

[Figure]

*Figure 7: Map of correlations (R) differences (analysis minus openloop) for stations available over North America. Small dots represent stations where R differences are not significant (i.e. 95% confidence intervals are overlapping), large circles where differences are significant.*

[Figure]

*Figure S6: a) Boxplots representing the distribution of the correlation values on absolute time-series and anomaly time-series ("Ano") between the stations with in situ measurements of soil*

*moisture either 5cm depth or 20 cm depth and soil moisture from LDAS_ERA5 openloop and analysis over 2010-2018 (third and forth layer of soil, respectively). Correlation values are presented for surface soil moisture (5 cm depth measurements against third layer of soil), only. Distribution are centred on the median values. b) Distribution of the Bias values between the stations with in situ measurements of soil moisture either 5cm depth or 20 cm depth and soil moisture from LDAS_ERA5 openloop and analysis over 2010-2018 (third and forth layer of soil, respectively).c) Same as b) for ubRMSD.*

**1.2 [L107-117: Is it necessary to include such details about the datasets in the introduction?]**

**Response to 1.2**

We agree that a lot of information is provided in this bullet. However we believe acronyms should be detailed and appropriate references should be used the first time they appear in the text.

**1.3 [L180: Please specify what you mean by flow dependency between the prognostic variables and the observations.]**

**Response to 1.3**

This sentence has been rephrased: "Flow dependency between the model control variables and the observations are generated using finite differences from perturbed simulations" is now (P.6, Lines 192-195): "The flow-dependency (dynamic link) between prognostic variables and the observations is ensured in the SEKF through the observation operator Jacobians, which propagate information from the observations to the analysis via finite-difference computations (de Rosnay et al., 2013)"

**1.4 [L198-200: Difficult to interpret the difference in LAI error when you use a mix of percent and m2/m2. Please could you clarify this?]**

**Response to 1.4**

We agree that this sentence could be improved. Setting up the observed and modelled LAI standard deviation to 20 % of the LAI value is an empirical option coming from previous studies by Jarlan et al. (2008) and Rudiger et al. (2010), which have underlined the need for a variable LAI error definition. Barbu et al. (2011) further explored the impact of LAI model and background errors on the assimilation results by using diagnostics on model and observation errors (e.g. Desroziers and Ivanov, 2001) on different setups (see figure 2 of Barbu et al., 2011). They found that for small LAI values, it is necessary to use a fixed error standard deviation. This value was set to 0.04 $m^2m^{-2}$ for LAI values lower than 2 $m^2m^{-2}$ and is also used in this study.

The following sentence: "The standard deviation of errors for the observed LAI is assumed to be 20% and a similar assumption is made for the standard deviation of errors of the modelled LAI values higher than 2 m2m−2. For modelled LAI values lower than 2 $m^2m^{-2}$, a constant error of 0.4 m2m−2 is assumed (Barbu et al., 2011). More details can be found in Albergel et al, 2017 or Tall et al., 2019." as been reformulated and is now (P.7, Lines 220-224): "Based on previous results from Jarlan et al., 2008, Rüdiger et al., 2010, Barbu et al., 2011, observed and modelled LAI standard deviation errors are set to 20 % of the LAI value itself for values higher than 2$m^2m^{-2}$. For LAI values lower than 2 $m^2m^{-2}$, a fixed value of 0.04 $m^2m^{-2}$ has been used. More detailed can be found in Barbu et al., 2011 (section 2.3 on data assimilation scheme and figure 2)."

Reference (not added to the manuscript):

Desroziers, D. and Ivanov, S.: Diagnosis and adaptive tuning of observation-error parameters in a variational assimilation, Q. J. Roy. Meteorol. Soc., 127, 1433–1452, 2001.

Reference (added to the manuscript):
Jarlan, L., Balsamo, G., Lafont, S., Beljaars, A., Calvet, J.-C., and Mougin, E.: Analysis of leaf area index in the ECMWF land surface model and impact on latent heat on carbon fluxes: Application to West Africa, J. Geophys. Res., 113, D24117, doi:10.1029/2007JD009370, 2008.

Reference (already in the manuscript):

Rüdiger, C.; Albergel, C.; Mahfouf, J.-F.; Calvet, J.-C.; Walker, J.P. Evaluation of Jacobians for leaf area index data assimilation with an extended Kalman filter. J. Geophys. Res. 2010.

**1.5 [L251: Could you please include why you don't consider assimilating surface soil moisture observations from the Soil Moisture and Ocean Salinity (SMOS) and/or from the Soil Moisture Active Passive (SMAP) satellite missions? As these satellites are expected to be more sensitive to surface soil moisture than the C-band observations from ASCAT.**

**Response to 1.5**

We find it difficult at this stage to include why a specific dataset has not been used. The development of LDAS-Monde at CNRM has been made possible through different externally funded project including the Copernicus Gobal Land Service providing, amongst other datasets, the ASCAT Soil Wetness Index used in this study. ASCAT is from 2007 onward an operational product obtained from sensors onboard the METOP satellites and has been used at CNRM for many years. However, it is true than any satellite surface soil moisture products can be assimilate into LDAS-Monde. At CNRM, Albergel et al., 2017 have assimilated the ESA CCI (European Space Agency, Climate Change Initiative) combined surface soil moisture product (e.g. Dorigo et al., 2015), Parrens et al., 2014 have assimilated SMOS surface soil moisture. Future work will assimilate the most recent ESA CCI surface soil moisture dataset (v4.5) up to 2018. It includes the SMOS data.

Albergel, C., Munier, S., Leroux, D. J., Dewaele, H., Fairbairn, D., Barbu, A. L., Gelati, E., Dorigo, W., Faroux, S., Meurey, C., Le Moigne, P., Decharme, B., Mahfouf, J.-F., and Calvet, J.-C.: Sequential assimilation of satellite-derived vegetation and soil moisture products using SURFEX_v8.0: LDAS-Monde assessment over the Euro-Mediterranean area, Geosci. Model Dev., 10, 3889–3912, https://doi.org/10.5194/gmd-10-3889-2017, 2017.

Dorigo, W.A., A. Gruber, R.A.M. De Jeu, W. Wagner, T. Stacke, A. Loew, C. Albergel, L. Brocca, D. Chung, R.M. Parinussa and R. Kidd: Evaluation of the ESA CCI soil moisture product using ground-based observations, Remote Sensing of Environment, http://dx.doi.org/10.1016/j.rse.2014.07.023, 2015.

Parrens, M., Mahfouf, J.-F., Barbu, A. L., and Calvet, J.-C.: Assimilation of surface soil moisture into a multilayer soil model: design and evaluation at local scale, Hydrol. Earth Syst. Sci., 18, 673–689, https://doi.org/10.5194/hess-18-673-2014, 2014.

**1.6 [Furthermore, as I understand ASCAT data are already assimilated in the production of the ERA5 dataset. Will the LDAS-Monde assimilation not lead to a "double" counting or usage of the ASCAT data and what are the potential consequences for your analyses results?]**

**Response to 1.6**

Thank you for your comment. ASCAT soil moisture is indeed assimilated in the ERA5 LDAS. However, previous studies showed that its impact is confined to the soil and that it is neutral on the

IFS atmospheric analysis and forecasts (de Rosnay et al 2014, Munoz-Sabater et al 2019). In our study we use the ERA5 atmospheric analysis as forcing but we do not use any of the ERA5 soil analysis variables as input of our system. So, we consider the ASCAT SM contribution to the ERA5 atmospheric forcing to be negligible.

Reference (already in the manuscript):
de Rosnay, P.; Balsamo, G.; Albergel, C.; Muñoz-Sabater, J.; Isaksen, L. Initialisation of land surface variables for numerical weather prediction. Surv. Geophys., 35, 607–621, doi: 10.1007/s10712-012-9207-x, 2014.
Reference (not added to the revised version of the manuscript):
Muñoz-Sabater, J. , Lawrence, H. , Albergel, C. , de Rosnay, P. , Isaksen, L. , Mecklenburg, S. , Kerr, Y. and Drusch, M. (2019), Assimilation of SMOS brightness temperatures in the ECMWF Integrated Forecasting System. Q J R Meteorol Soc. Accepted Author Manuscript. doi:10.1002/qj.3577

**1.7 [L268: Please could you specify the difference between linear rescaling and CDF-matching (if any)? To my understanding linear rescaling is correction of the mean and standard deviation, while CDF-matching corrects the whole CDF (i.e., all moments of the probability distribution function), hence linear rescaling is not the same as CDF-matching.]**

**Response to 1.7**
We use in this paper a seasonal linear rescaling. Linear rescaling was introduced by Scipal et al. (2008) and has been shown giving results that are very similar to an exact CDF matching. Nevertheless, to avoid any confusion, we have rewritten the sentence as follows (P.9-10, Lines 294-301): "This is done through a linear rescaling as proposed by Scipal et al. (2007), where the observations mean and variance are matched to the modelled soil moisture mean and variance from the second layer of soil (1-4 cm depth). This rescaling gives in practice very similar results to CDF (cumulative distribution function) matching. The linear rescaling is performed on a seasonal basis (with a 3-month moving window) as suggested by Draper et al., (2011), Barbu et al., (2014)." Further mentions of CDF matching in the manuscript have been replaced by "seasonal linear rescaling".

**1.8 [L292-294: Could you please discuss how this short spinup period could affect your results?]**

**Response to 1.8**

Nine months can be perceived as a short period to spin up the system. Unfortunately, HRES atmospheric forcing is only available from April 2016 and the LDAS-HRES experiment ends in December 2018. We have considered this 9 months period for the spin up in order to have the longest possible time series for land surface variables, thus giving more strength to statistics. We could have considered a longer period for spin up (April 2016 to December 2017) and studied only 2018. This gives very similar results on surface soil moisture and LAI (not shown). While not being fully spun-up, results obtained with LDAS-HRES can be considered as representative of the system response to data assimilation. Note that most initial values of the LDAS-HRES run are taken from the ECOCLIMAP-II database. For instance, initial LAI is set from a 1999-2005 climatology derived from MODIS.

Another possibility to initialise LDAS-HRES could have been to downscale the state of LDAS-ERA5 run in April 2016 to 0.10°x0.10° spatial resolution. LDAS-ERA5 runs have been set to an equilibrium spinning up 20 times the first year (2010).

The following sentence: "The period 2017-2018 is presented, HRES is available at this spatial resolution from April 2016, only, and the time period from April to December 2016 is used as a short spinup." has been modified and is now (P.10, Lines 327-332): "HRES is available at a 0.1° x 0.1° resolution only from April 2016. April to December 2016 is used as a short period for spinup and results are presented for the period 2017-2018. Although a 9-month spinup period can be seen as rather short, evaluating LDAS-HRES on either 2017-2018 or 2018 (using instead a 21-month spinup) leads to similar results on surface soil moisture and LAI (not shown). While the system is not fully spun-up, it can be considered as representative of the system response to data assimilation."

**1.9 [L383: Could you please provide more details on how this can explain the differences seen between ISBA and GLEAM?]**

**Response to 1.9**

GLEAM is an hydrological model and vegetation is mainly driven by observations. On the contrary, ISBA also represents plant growth and leaf-scale physiological processes, models key vegetation variables like LAI and above ground biomass (see section 2.1.1 on ISBA Land Surface Model). Within GLEAM, each grid cell comprises four different land-cover types: (1) bare soil, (2) low vegetation (e.g. grass), (3) tall vegetation (e.g. trees), and (4) openwater (e.g. lakes). Except for the fraction of open water, these fractions are sourced from the Global Vegetation Continuous Fields product (MOD44B), based on observations from the Moderate Resolution Image Spectroradiometer (MODIS). While in ISBA, each grid cell can be composed of up to 12 generic land surface types, bare soil, rocks, and permanent snow and ice surfaces as well as nine plant functional types (needle leaf trees, evergreen broadleaf trees, deciduous broadleef trees, C3 crops, C4 crops, C4 irrigated crops, herbaceous, tropical herbaceous and wetlands). Those types depart from prevalent land cover products such as CLC2000 (Corine Land Cover) and GLC2000 (Global Land Cover) by splitting existing classes into new classes that possess a better regional character by virtue of the climatic environment (latitude, proximity to the sea, topography).

Work is undergoing at CNRM to better understand the differences between terrestrial evaporation from ISBA and GLEAM. In particular, the different components of terrestrial evaporation, i.e. transpiration, bare soil evaporation and, interception loss are investigated.

Paragraph: "However GLEAM only estimates (root-zone) soil moisture and terrestrial evaporation, while ISBA in LDAS_ERA5 is a physically-based land surface model, accounting for more processes linked to vegetation."
is now (P.14-15, Lines 458-471):
"However GLEAM is an evaporation model designed to be driven by remote sensing observations only. GLEAM only estimates (root-zone) soil moisture and terrestrial evaporation while the $CO_2$-responsive version of ISBA in LDAS_ERA5 is a physically-based land surface model, accounting for more processes linked to vegetation (see section 2.1.1). It has to be noted that the auxiliary dataset used to e.g. represent the different land cover types are different also. Within GLEAM, the land cover types fractions are sourced from the Global Vegetation Continuous Fields product (MOD44B), based on observations from the Moderate Resolution Image Spectroradiometer (MODIS). Four land cover types are considered, bare soil, low vegetation (e.g. grass), tall vegetation (e.g. trees), and openwater (e.g. lakes). In ISBA the 12 land cover types fraction depart from prevalent land cover products such as CLC2000 (Corine Land Cover) and GLC2000 (Global Land Cover). It can potentially impact the distribution of the terrestrial evaporation between GLEAM and ISBA."

Further work at CNRM will focus on understanding the differences between ISBA and GLEAM, in particular investigating the sub-components of terrestrial evaporation."

---

## Author Comment (AC2) · 22 Jan 2020

Dear Reviewer 2 many thanks for reviewing the manuscript and for highlighting its relevance and interest. Your comments and suggestions led to an improved version of the manuscript. Attached is a point by point answer to your specific comments, all your editorial and technical comments were accounted for in the revised version of the manuscript.

Response to Reviewer 2 are structured as follow: (1) 2.X: comments from Reviewer 2, (2) Response to 2.X: author's response and author's changes in manuscript when any. For sake of clarity, line and page numbering from the revised version is used.

[Figure]

Please also note the supplement to this comment:
https://www.hydrol-earth-syst-sci-discuss.net/hess-2019-534/hess-2019-534-AC2-supplement.pdf

**Supplement:**

Response to Reviewer 2 are structured as follow: (1) 2.X: comments from Reviewer 2, (2) Response to 2.X: author's response and author's changes in manuscript when any. For sake of clarity, line and page numbering from the revised version is used.

**Reviewer#2**

**[...] This paper seems to represent a major milestone in the development of LDAS-Monde (which is in my view a very important undertaking), and hence I recommend publishing the paper after minor revisions.**

Dear Reviewer#2 many thanks for reviewing the manuscript and for highlighting its relevance and interest. Your comments and suggestions led to an improved version of the manuscript. Below is a point by point answer to your specific comments, all your editorial and technical comments were accounted for in the revised version of the manuscript.

**2.1 [Line 167: What do you mean by "...is bale to ..."?]**

**Response to 2.1**

Thanks for pointing out this typo, it should read "[…] *is able to* [...]" it is now corrected in the revised version of the manuscript.

**2.2 [Lines 188ff: The procedure described here results in an observational error field mainly related to soil properties, while the real retrieval errors are mostly dependent on vegetation density. Please discuss implications.]**

**Response to 2.2**

You are right that vegetation has a role in ASCAT SSM observational error. The observational SSM error we use is consistent with errors typically expected for remotely sensed SSM (e.g., de Jeu et al., 2008, Gruber et al, 2016). Most of the in-situ measurements sites used in typical evaluation studies are indeed representative of grassland. Going from radar backscatter measurements (ASCAT level1 data, σ°) to SSM (ASCAT level2 data) using the change detection approach developed at TUWIEN implies a lot of assumptions in particular on vegetation variability: only seasonal variability is accounted for (e.g. Wagner et al., 1999, Bartalis et al., 2007). That is why we have an undergoing work at CNRM trying to directly assimilate σ° (Shamambo et al., 2019). Assimilating σ° also raises the question of how to specify observation, background, and model error covariance matrices. The last decade has seen the development of techniques to estimate those matrices. Approaches based on Desroziers diagnostics (Desroziers et al., 2005) are affordable for land data assimilation systems from a computational point of view and could provide insightful information on the various sources of the data assimilation system.

The following paragraph has been added in the discussion and conclusion section (P.24, Lines 788-796):
"CNRM is also investigating the direct assimilation of ASCAT radar backscatter (Shamambo et al., 2019), it is supposed to tackle the way vegetation is accounted for in the change detection approach used to retrieve SSM with an improved representation of its effect. Assimilating ASCAT radar backscatter also raises the question of how to specify observation, background, and model error covariance matrices, so far mainly relying on soil properties (see section 2.1.3 on data assimilation). The last decade has seen the development of techniques to estimate those matrices. Approaches based on Desroziers diagnostics (Desroziers et al., 2005) are affordable for land data assimilation

systems from a computational point of view and could provide insightful information on the various sources of the data assimilation system".

References (* denotes new references added to the manuscript):

Bartalis, Z.; Wagner, W.; Naeimi, V.; Hasenauer, S.; Scipal, K.; Bonekamp, H.; Figa, J.; Anderson, C.: Initial soil moisture retrievals from the METOP-A advanced Scatterometer (ASCAT). Geophys. Res. Lett., 34, L20401, doi: 10.1029/2007GL031088., 2007.

Desroziers, G.; Berre, L.; Chapnik, B.; Poli, P. Diagnosis of observation, background and analysis-error statistics in observation space. Q. J. Roy. Meteor. Soc. **2005**, 131, 3385–3396.

de Jeu, R.A.; Wagner, W.; Holmes, T.R.H.; Dolman, A.J.; Van De Giesen, N.C.; Friesen, J. Global soil moisture patterns observed by space borne microwave radiometers and scatterometers. Surv. Geophys., 29, 399–420, 2008.

Gruber, A.; Su, C.-H.; Zwieback, S.; Crow, W.; Dorigo, W.; Wagner, W. Recent advances in (soil moisture) triple collocation analysis. Int. J. Appl. Earth Obs. Geoinf., 45, 200–211, 2016.

Shamambo, D.C.; Bonan, B.; Calvet, J.-C.; Albergel, C.; Hahn, S. Interpretation of ASCAT Radar Scatterometer Observations Over Land: A Case Study Over Southwestern France. *Remote Sens.* 2019, *11*, 2842.

Wagner, W.; Lemoine, G.; Rott, H. A method for estimating soil moisture from ERS scatterometer and soil data. Remote Sens. Environ., 70, 191–207, 1999.

**2.3 [Line 198: Is "20 %" a relative error?]**

**Response to 2.3**

It is 20% of the LAI itself, this paragraph has been revisited to improve its understanding. Setting up the observed and modelled LAI standard deviation to 20 % of the LAI value is an empirical option coming from previous studies by Jarlan et al. (2008) and Rudiger et al. (2010), which have underlined the need for a variable LAI error definition. Barbu et al. (2011) further explored the impact of LAI model and background errors on the assimilation results by using diagnostics on model and observation errors (e.g. Desroziers and Ivanov, 2001) on different setups (see figure 2 of Barbu et al., 2011). They found that for small LAI values, it is necessary to use a fixed error standard deviation. This value was set to 0.04 $m^2m^{-2}$ for LAI values lower than 2 $m^2m^{-2}$ and is also used in this study.

The following sentence: "The standard deviation of errors for the observed LAI is assumed to be 20% and a similar assumption is made for the standard deviation of errors of the modelled LAI values higher than 2 m2m−2. For modelled LAI values lower than 2 m2m−2, a constant error of 0.4 m2m−2 is assumed (Barbu et al., 2011). More details can be found in Albergel et al, 2017 or Tall et al., 2019." as been reformulated and is now (P.7, Lines 220-224): "Based on previous results from Jarlan et al., 2008, Rüdiger et al., 2010, Barbu et al., 2011, observed and modelled LAI standard deviation errors are set to 20 % of the LAI value itself for values higher than 2$m^2m^{-2}$. For LAI values lower than 2 $m^2m^{-2}$, a fixed value of 0.04 $m^2m^{-2}$ has been used. More detailed can be found in Barbu et al., 2011 (section 2.3 on data assimilation scheme and figure 2)."

Reference (not added to the manuscript):
Desroziers, D. and Ivanov, S.: Diagnosis and adaptive tuning of observation-error parameters in a variational assimilation, Q. J. Roy. Meteorol. Soc., 127, 1433–1452, 2001.
Reference (added to the manuscript):

Jarlan, L., Balsamo, G., Lafont, S., Beljaars, A., Calvet, J.-C., and Mougin, E.: Analysis of leaf area index in the ECMWF land surface model and impact on latent heat on carbon fluxes: Application to West Africa, J. Geophys. Res., 113, D24117, doi:10.1029/2007JD009370, 2008.

Reference (already in the manuscript):

Rüdiger, C.; Albergel, C.; Mahfouf, J.-F.; Calvet, J.-C.; Walker, J.P. Evaluation of Jacobians for leaf area index data assimilation with an extended Kalman filter. J. Geophys. Res. 2010.

**2.4 [Section 2.2: Note that ASCAT SSM data are already assimilated in ERA5. Please discuss implications.]**

**Response to 2.4**

Thank you for your comment. ASCAT soil moisture is indeed assimilated in the ERA5 LDAS. However, previous studies showed that its impact is confined to the soil and that it is neutral on the IFS atmospheric analysis and forecasts (de Rosnay et al 2014, Munoz-Sabater et al 2019). In our study we use the ERA5 atmospheric analysis as forcing but we do not use any of the ERA5 soil analysis variables as input of our system. So, we consider the ASCAT SM contribution to the ERA5 atmospheric forcing to be negligible.

Reference (already in the manuscript):
de Rosnay, P.; Balsamo, G.; Albergel, C.; Muñoz-Sabater, J.; Isaksen, L. Initialisation of land surface variables for numerical weather prediction. Surv. Geophys., 35, 607–621, doi: 10.1007/s10712-012-9207-x, 2014.
Reference (not added to the revised version of the manuscript):
Muñoz-Sabater, J. , Lawrence, H. , Albergel, C. , de Rosnay, P. , Isaksen, L. , Mecklenburg, S. , Kerr, Y. and Drusch, M. (2019), Assimilation of SMOS brightness temperatures in the ECMWF Integrated Forecasting System. Q J R Meteorol Soc. Accepted Author Manuscript. doi:10.1002/qj.3577

**2.5 [Line 248: SWI is the Soil Water Index]**

**Response to 2.5**

Thanks, it has been corrected accordingly

**2.6 [Section 2.3: Describe also the masking of SSM]**

**Response to 2.6**

Thanks for your comment, the following sentence has been added to section 2.3 (P.9-10, Lines 299-301): "As in Albergel et al. (2018a, 2018b), pixels whose average altitude exceeds 1500 m above sea level as well as pixels with urban land cover fractions larger than 15% were discarded as those conditions may affect the retrieval of soil moisture from space."

**2.7 [Line 493: Only this sub-study focusses on severe conditions, but not "this study" overall.]**

**Response to 2.7**

We agree with Reviewer#2 and the sentence has been corrected accordingly, it is now: "As this subsection focuses [...]"

---

## Author Comment (AC3) · 22 Jan 2020

Dear Reviewer#3 many thanks for reviewing the manuscript and for highlighting its relevance and interest. Your comments and suggestions led to an improved version of the manuscript. Attached is a point by point answer to your specific comments, all your editorial and technical comments were accounted for in the revised version of the manuscript.

Response to Reviewer 3 are structured as follow: (1) 3.X: comments from Reviewer 3, (2) Response to 3.X: author's response and author's changes in manuscript when any. For sake of clarity, line and page numbering from the revised version are used.

[Figure]

Please also note the supplement to this comment:
https://www.hydrol-earth-syst-sci-discuss.net/hess-2019-534/hess-2019-534-AC3-
supplement.pdf

―――――――――――――――――――

**Supplement:**

Response to Reviewer 3 are structured as follow: (1) 3.X: comments from Reviewer 3, (2) Response to 3.X: author's response and author's changes in manuscript when any. For sake of clarity, line and page numbering from the revised version are used.

**Reviewer#3**

**[...] Overall, the LDAS-Monde system is great, but the paper needs a thorough revision [...]**

Dear Reviewer#3 many thanks for reviewing the manuscript and for highlighting its relevance and interest. Your comments and suggestions led to an improved version of the manuscript. Below is a point by point answer to your specific comments, all your editorial and technical comments were accounted for in the revised version of the manuscript.

**3.1 [Are the perturbations chosen to get an optimal data assimilation system? Please discuss]**

**Response to 3.1**

Yes, several studies have investigated the size of the perturbations within the ISBA LSM. In particular Draper et al., 2009, for soil moisture, Rüdiger et al., 2010, for LAI. The following sentence has been added to the revised version of the manuscript (as well as the new reference to Draper et al., 2009):
Section 2.1.3 on data assimilation
P.7, Lines 202-204:"Several studies (e.g. Draper et al., 2009; Rüdiger et al., 2010) have demonstrated that small perturbations lead to a good approximation of this linear behaviour, provided that computational round-off error is not significant."

References:
Draper, C. S., Mahfouf, J.-F., and Walker, J. P.: An EKF assimilation of AMSR-E soil moisture into the ISBA land surface scheme, J. Geophys. Res., 114, D20104, https://doi.org/10.1029/2008JD011650, 2009.
Rüdiger, C., Albergel, C., Mahfouf, J.-F., Calvet, J.-C., and Walker, J. P.: Evaluation of Jacobians for Leaf Area Index data assimilation with an extended Kalman filter, J. Geophys. Res., 115, D09111, https://doi.org/10.1029/2009JD012912, 2010.

**3.2 [How are the cross correlations between the errors in the various soil layers defined, and the error correlations between LAI and soil moisture?]**

**Response to 3.2**

In the SEKF, no covariance is directly prescribed between LAI and soil moisture or soil moisture between the various soil layers. The sensitivity of model variables to observations is entirely driven by the Jacobian of the observation operator, which is defined as the product of the model state evolution from t to t + 24h and the conversion of the model state into the observation equivalent (see paragraph 2.3.1 and supplementary material of Bonan et al. (2020)). The value of Jacobian has been heavily studied in previous publications such as Albergel et al. (2017) or Tall et al. (2019).

Within LDAS-Monde, cross correlations between the errors in the various variables (soil moisture of the different layers and LAI) will be investigated in a near future based on the Ensemble Square Root Filter (EnSRF) proposed by Bonan et al., 2020.

References:

Albergel, C., Munier, S., Leroux, D. J., Dewaele, H., Fairbairn, D., Barbu, A. L., Gelati, E., Dorigo, W., Faroux, S., Meurey, C., Le Moigne, P., Decharme, B., Mahfouf, J.-F., and Calvet, J.-C.: Sequential assimilation of satellite-derived vegetation and soil moisture products using SURFEX_v8.0: LDAS-Monde assessment over the Euro-Mediterranean area, Geosci. Model Dev., 10, 3889–3912, https://doi.org/10.5194/gmd-10-3889-2017, 2017.

Bonan, B., Albergel, C., Zheng, Y., Barbu, A. L., Fairbairn, D., Munier, S., and Calvet, J.-C.: An Ensemble Square Root Filter for the joint assimilation of surface soil moiture and leaf area index within LDAS-Monde: application over the Euro-Mediterranean region, Hydrol. Earth Syst. Sci. Discuss., https://doi.org/10.5194/hess-2019-391, accepted, 2020.

Tall, M.; Albergel, C.; Bonan, B.; Zheng, Y.; Guichard, F.; Dramé, M.S.; Gaye, A.T.; Sintondji, L.O.; Hountondji, F.C.C.; Nikiema, P.M.; Calvet, J.-C. Towards a Long-Term Reanalysis of Land Surface Variables over Western Africa: LDAS-Monde Applied over Burkina Faso from 2001 to 2018. *Remote Sens.*, *11*, 735, 2019

**3.3 [ASCAT has an approximate resolution of 25 km. How are these coarse data assimilated/downscaled into the 0.1ˆo model simulations?]**

**Response to 3.3**

The assimilated SWI product is provided by the Copernicus Global Land Service directly on a global 0.1° regular grid. Informations on how the SWI product is derived from ASCAT data at 25-km resolutions can be found in the Product User Manual (https://land.copernicus.eu/global/sites/cgls.vito.be/files/products/CGLOPS1_PUM_SWIV3-SWI10-SWI-TS_I2.60.pdf).

**3.4 [CDF matching ' refers to rescaling of the entire CDF, and is not a correct terminology when only rescaling the mean and variance.]**

**Response to 3.4**

We use in this paper a seasonal linear rescaling. Linear rescaling was introduced by Scipal et al. (2008) and has been shown giving results that are very similar to an exact CDF matching. Nevertheless, to avoid any confusion, we have rewritten the sentence as follows (P.9-10, Lines 299-301): "This is done through a linear rescaling as proposed by Scipal et al. (2007), where the mean and variance of observations are matched to the mean and variance of the modelled soil moisture from the second layer of soil (1-4 cm depth). This rescaling gives in practice very similar results to CDF (cumulative distribution function) matching. The linear rescaling is performed on a seasonal basis (with a 3-month moving window) as suggested by Draper et al., (2011), Barbu et al., (2014)." Further mentions of CDF matching in the manuscript have been replaced by "seasonal linear rescaling".

**3.5 [How exactly are the LAI data 'interpolated' from 1 km to 0.25 degree? Do you mean interpolation to bridge cloudy pixels and then aggregation (upscaling)?]**

**Response to 3.5**

Thanks for this suggestion. As in previous studies (e.g, Barbu et al., 2014, Albergel et al., 2019), observations are aggregated using an arithmetic average to the model grid points (0.25° or 0.10° in this study), if at least 50 % of the model grid points are observed (i.e. half the maximum amount).

Future work will focus on looking at the impact of cloud cover on the LAI upscaling process. Instead of 50%, a possibility could be to use an arithmetic average to the model grid point if at least

70% of the model grid point are observed. Then during the assimilation/evaluation ERA5 (or HRES IFS) total cloud cover field (tcc) could be use to mask out grid point if tcc is greater than 30%. This is already used when evaluating e.g. satellite land surface temperature to model data (e.g. Johannsen et al., 2019).

Reference:
Johannsen, F.; Ermida, S.; Martins, J.P.A.; Trigo, I.F.; Nogueira, M.; Dutra, E. Cold Bias of ERA5 Summertime Daily Maximum Land Surface Temperature over Iberian Peninsula. *Remote Sens., 11*, 2570, 2019.

**3.6 [Is the LAI also 'converted from the observation space to the model space' as is done for soil moisture? Please describe how? If there is no such rescaling, then the results may be trivial, i.e there will be more impact of a non-rescaled LAI assimilation than when doing a gentle nudging with rescaled soil moisture. However, since you use a KF variant, there probably is some rescaling for both (otherwise the KF assumptions would be violated).]**

**Response to 3.5**

Soil moisture is a very model-specific variable, precipitation, evapotranspiration, soil texture, topography, vegetation, and land use could either enhance or reduce the spatial variability of soil moisture depending on how it is distributed and combined with other factors (Famiglietti et al. 2008). In particular, differences in soil properties between the model grid points and reality could imply important variations in the mean and variance of soil moisture. Furthermore, vegetation effects are not completely corrected when going from the satellite measurement (e.g. radar backscatter in the case of ASCAT) to SSM, leading to potential seasonal biases (e.g. Shamambo et al., 2019). That is why we apply the linear rescaling to the ASCAT SWI. It also acts as an observation operator to go from the observational space (SWI, an index 0 and 1) to the model space (SSM in $m^3m^{-3}$).

For LAI, biases between the model and the observations are linked to the way processes are represented in the model as well as uncertainties on the atmospheric forcing (cumulated effect on modelled LAI). The assimilation sequentially removes bias in the modelled LAI (with respect to the observed LAI). This technical difference between SSM and LAI assimilation, combined with the longer memory of LAI compared to SSM contributes to the results presented in this study. See also response to comment 3.14.

References:
Famiglietti, J. S., D. Ryu, A. A. Berg, M. Rodell, and T. J. Jackson, 2008: Field observations of soil moisture variability across scales. *Water Resour. Res.*, 44, W01423, doi:10.1029/2006WR005804.
Shamambo, D.C.; Bonan, B.; Calvet, J.-C.; Albergel, C.; Hahn, S. Interpretation of ASCAT Radar Scatterometer Observations Over Land: A Case Study Over Southwestern France. *Remote Sens.* **2019**, *11*, 2842.

**3.6 [How exactly is the 'climatology' defined? Is it seasonally varying, how much smoothing is applied, etc?]**

**Response to 3.6**

The following sentence "This 9-yr global reanalysis was then used to provide a climatology for estimating anomalies of the land surface conditions." has been reformulated and is now (P.10, Lines 317-320) "This 9-yr global reanalysis was then used to provide a monthly climatology for

estimating anomalies of the land surface conditions. For each month (and variable considered) of 2018 we have removed the monthly mean and scaled by the monthly standard deviation of the 2010-2018 period"

**3.7 [The spinup period for the 0.1ˆo simulation seems unrealistically short. How was it initialized? Could you cycle over the short April-December period multiple times?**

**Response to 3.7**

Nine months can be perceived as a too short period to spin up the system. Unfortunately, HRES atmospheric forcing is only available from April 2016 and the LDAS-HRES experiment ends in December 2018. We have considered this 9 months period for the spin up in order to have the longest possible time series for land surface variables, thus giving more strength to statistics. We could have considered a longer period for spin up (April 2016 to December 2017) and studied only 2018. This gives very similar results on surface soil moisture and LAI (not shown). While not being fully spun-up, results obtained with LDAS-HRES can be considered as representative of the system response to data assimilation. Note that most initial values of the LDAS-HRES run are taken from the ECOCLIMAP-II database. For instance, initial LAI is set from a 1999-2005 climatology derived from MODIS

Another possibility to initialise LDAS-HRES could have been to downscale the state of LDAS-ERA5 run in April 2016 to 0.10°x0.10° spatial resolution. LDAS-ERA5 runs have been set to an equilibrium spinning up 20 times the first year (2010).

The following sentence: "The period 2017-2018 is presented, HRES is available at this spatial resolution from April 2016, only, and the time period from April to December 2016 is used as a short spinup." has been modified and is now (P.10, L.327-332): "HRES is available at a 0.1° x 0.1° resolution only from April 2016. April to December 2016 is used as a short period for spinup and results are presented for the period 2017-2018. Although a 9-month spinup period can be seen as rather short, evaluating LDAS-HRES on either 2017-2018 or 2018 (using instead a 21-month spinup) leads to similar results on surface soil moisture and LAI (not shown). While the system is not fully spun-up, it can be considered as representative of the system response to data assimilation."

**3.8 [Table II: An observation operator is a function, not a variable; also explain what you mean by control variable (updated variables) for readers who are new to the field. In fact, the control vector enters the observation operator, which in turn selects a subset of relevant variables to produce the observation prediction.]**

**Response to 3.8**

Agreed, in Table II "Observations operators" has been replaced by "Model equivalents" and the following sentences have been added to the revised version of the manuscript (section 2.1.3 on data assimilation, P.6, Lines 200-202): "The eight control variables are directly updated using their sensitivity to observed variables (i.e. defined by the Jacobians). Other variables are indirectly modified through biophysical processes and feedbacks from the model"

**3.9 [Why is there no skill evaluation in terms of anomalies? Would be interesting.]**

**Response to 3.9**

Thank your for your highly relevant comment. Following it and similar comments from the other Reviewers, it has been decided to revisit the soil moisture evaluation part of the study:
(1) we have added an evaluation of soil moisture from LDAS-Monde fourth layer of soil (10 to 20 cm) against in situ measurements of soil moisture at 20 cm depth when available (10 networks and 685 stations),
(2) for surface soil moisture (SSM), correlation values (R) were calculated for both absolute and anomaly time-series in order to remove the strong impact from the SSM seasonal cycle on this specific metric,
(3) a 95% Confidence Interval (CI) has been added to R values.
(4) we have added the number of stations for which correlations differences are significant (significant improvement or degradation from the analysis) as well as a map over North America for illustration.

It involves several changes in the revised version of the manuscript, they are listed below.

Methodology section, 2.5 Evaluation datasets and metrics

P.11, Lines 358-365:"In situ measurements of surface soil moisture from 19 networks across 14 countries available from the ISMN are also used to evaluate the performance of the soil moisture analysis. They represent 782 stations with at least 2 years of daily data over 2010-2018. Sensors at 5 cm depth (SSM) are compared with soil moisture from LDAS_ERA5 third layer of soil (4-10 cm), sensors at 20 cm depth with the fourth layer of soil (10-20 cm, 685 stations from 10 networks). Beside 11 stations located in 4 countries of Western Africa (Benin, Mali, Sénégal and Niger) and 21 stations in Australia, most stations are located in North America and Europe, see Table S3."

P.12, Lines 374-377: "For global estimates, Normalized RMSD (NRMSD, Eq.(2)) was used, also. Finally, for surface soil moisture, R was calculated for both absolute and anomaly time-series in order to remove the strong impact from the SSM seasonal cycle on this specific metric (see e.g. Albergel et al., 2018a, 2018b)."

Result section, 3..1.2 Ground-based datasets

P.17-18, Lines 548-582: "The statistical scores for soil moisture from LDAS_ERA5 open-loop and analysis (third and fourth layers of soil, 4-10 cm depth, 10-20 cm depth, respectively) over 2010-2018 when compared with ground measurements from the ISMN (5 cm depth and 20 cm depth) are presented in Table S2 for each individual network. Averaged statistical metrics (ubRMSD, R,$R_{anomaly}$ and bias) are similar for both LDAS_ERA5 analysis and open-loop even if local differences exist. For the analysis, averaged R ($R_{anomaly}$) values along with its 95% Confidence Interval (CI) using in situ measurements at 5 cm (782 stations from 19 networks) are 0.68±0.03 (0.53±0.04) (0.67±0.03(0.53±0.04) for the open-loop) with averaged-network values going up to 0.88±0.01 (0.58±0.04) for the analysis (SOILSCAPE network, 49 stations in the USA) and always higher than 0.55 except for one network, ARM (10 stations in the USA) presenting an averaged R value of 0.29±0.05. Averaged ubRMSD and bias (LDAS_ERA5 minus in situ) are 0.060 $m^3m^{-3}$ and 0.077 $m^3m^{-3}$ for the analysis, 0.060 $m^3m^{-3}$ and 0.076 $m^3m^{-3}$ for the open-loop, respectively. NIC (Eq.1) has also been applied to R values, 65% of the pool of stations present a neutral impact from the analysis (511 stations at NIC ranging between -3 and +3), 12% present a negative impact (91 stations at NIC < -3) and 23% present a positive impact at (180 stations at NIC > +3).
The number of stations where R differences between the analysis and the openloop are significant (i.e. their 95% CI are not overlapping) is 186 out of 782 (about 26%). There is an improvement from the analysis w.r.t. the openloop for 128 stations (out of 186, i.e. about 69%) and a degradation for 58 stations (about 31%). Figure 7 illustrates R differences between the analysis and the openloop runs. When differences (analysis minus openloop) are not significant stations are

represented by a small dot. When they are significant, large circles have been used, blue for positive differences (an improvement from the analsysis) and red for negative differences (a degradation from the analysis). For most of the stations where a significant difference is obtained, it represent an improvement from the analysis.

Averaged analysis R (95%CI), bias and ubRMSD for the fourth layer of soil (685 stations from 10 networks) are 0.65±0.03, 0.049 $m^3m^{-3}$ and 0.055 $m^3m^{-3}$, respectively. For the open-loop, they are 0.064±0.03, 0.048 $m^3m^{-3}$ and 0.056 $m^3m^{-3}$, respectively. For soil moisture at that depth, about 60% of the stations present a neutral impact from the analysis (410 stations at NIC ranging between -3 and +3), 28% a positive impact (189 stations at NIC > +3) and 12% a negative impact (86 stations at NIC < -3). Although differences between the openloop run and the analysis are rather small, these results underline the added value of the analysis with respect to the model run. Figure S6 represents the distribution of the scores values for LDAS_ERA5 open-loop and analysis using boxplots centred on the median value. They look very similar and from this figure, it is difficult to see either improvement or degradation from the analysis."

[Figure]

*Figure 7: Map of correlations (R) differences (analysis minus openloop) for stations available over North America. Small dots represent stations where R differences are not significant (i.e. 95% confidence intervals are overlapping), large circles where differences are significant.*

[Figure]

*Figure S6: a) Boxplots representing the distribution of the correlation values on absolute time-series and anomaly time-series ("Ano") between the stations with in situ measurements of soil moisture either 5cm depth or 20 cm depth and soil moisture from LDAS_ERA5 openloop and analysis over 2010-2018 (third and forth layer of soil, respectively). Correlation values are presented for surface soil moisture (5 cm depth measurements against third layer of soil), only. Distribution are centred on the median values. b) Distribution of the Bias values between the stations with in situ measurements of soil moisture either 5cm depth or 20 cm depth and soil moisture from LDAS_ERA5 openloop and analysis over 2010-2018 (third and forth layer of soil, respectively).c) Same as b) for ubRMSD.*

The following text has been added to the revised version of the manuscript: "Figure S6 represents the distribution of the scores values for LDAS_ERA5 open-loop and analysis using boxplots centred on the median value. They look very similar and from this figure, it is difficult to see either improvement or degradation from the analysis."

**3.10 [Which variable in LDAS-Monde output is related to SIF and how?]**

**Response to 3.10**

In ISBA, the fluorescence is not simulated directly, but the photosynthesis activity is simulated through the calculation of the GPP, which is driven by plant growth and mortality in the model. Sun et al. (2017) demonstrated that SIF and GPP were driven by the same environmental and biological factors and found that SIF observations from OCO-2 and GPP products from FLUXCOM were highly consistent in time and space. The modelled GPP values are expressed in $g(C){\cdot}m^{-2}{\cdot}day^{-1}$, whereas SIF is an energy flux emitted by the vegetation in units of $mW{\cdot}m^{-2}{\cdot}sr^{-1}{\cdot}nm^{-1}$. Thus, GPP and SIF cannot be directly compared as they do not represent the same physical quantities. However, several studies (including Zhang et al., 2016, Sun et al., 2017, Leroux et al., 2018) have found that their time dynamics and their spatial distributions can be investigated.

The following paragraph has been added to the revised version of the manuscript (Section 2.5 on evaluation datasets and metrics, P.13, Lines 400-406): "As for SIF, in ISBA the fluorescence is not simulated directly, however photosynthesis activity is simulated through the calculation of the GPP, which is driven by plant growth and mortality in the model. Modelled GPP values are expressed in $g(C) \cdot m^{-2} \cdot day^{-1}$, while SIF is an energy flux emitted by the vegetation ($mW \cdot m^{-2} \cdot sr^{-1} \cdot nm^{-1}$). Hence, GPP and SIF cannot be directly compared as they do not represent the same physical quantities. However, several studies (e.g, Zhang et al., 2016, Sun et al., 2017, Leroux et al., 2018) have found that their time dynamics investigated, highlighting the potential of SIF products to be used as a validation support for GPP models."

References:
Leroux, D.J.; Calvet, J.-C.; Munier, S.; Albergel, C. Using Satellite-Derived Vegetation Products to Evaluate LDAS-Monde over the Euro-Mediterranean Area. *Remote Sens.* **2018**, *10*, 1199.
Sun, Y.; Frankenberg, C.; Wood, J.D.; Schimel, D.S.; Jung, M.; Guanter, L.; Drewry, D.T.; Verma, M.; Porcar-Castell, A.; Griffis, T.J.; et al. OCO-2 advances photosynthesis observation from space via solar-induced chlorophyll fluorescence. Science **2017**, 358, 189.
Zhang, Y.; Xiao, X.; Jin, C.; Dong, J.; Zhou, S.; Wagle, P.; Joiner, J.; Guanter, L.; Zhang, Y.; Zhang, G.; et al. Consistency between sun-induced chlorophyll fluorescence and gross primary production of vegetation in North America. Remote Sens. Environ. **2016**, 183, 154–169.

**3.11 [Overall, it is a bit disconcerting that trivial design results are shown repeatedly. Assimilate a variable, and sure, the model will get closer the assimilated observations. The results need to be thoroughly revised (both text and figures) to eliminate the trivial results. They can be mentioned once, but then the focus needs to be on the independent evaluation. It is also not correct to say that results "improve" if they simply get closer to the assimilated observations (e.g. L. 375, L. 516,...). This holds both for the global assessment and for the case studies, e.g. all of L. 505-512 is 'trivial' and can be removed.]**

**Response to 3.11**

Verifying that the assimilation system works as intended is an important task. This is why several figures have been included for "sanity check". We have emphasized in the manuscript that several presented evaluations are carried out to check if the assimilation system is working properly.

Also, using SSM and LAI as an independent source of information to evaluate the forecast has been further discussed and added in the revised version of the manuscript. While LAI remains an independent source of information for the forecast evaluation (although constrained by the assimilation), ASCAT SWI has been rescaled to match the model climatology. The seasonal rescaling impacts both bias and correlation. In an attempt to have a more independent evaluation, an additional figure has been put in the revised version of the manuscript. It displays maps of correlations between modelled soil moisture (1-4 cm) from the four experiments (LDAS-HRES openloop, analysis, LDAS_fc4 and LDAS_fc8) and ASCAT SWI (i.e. ASCAT data prior rescaling) for the WEUR domain. Correlations are applied to both absolute values and to anomalies (to assess the short term variability of soil moisture).

*End of section 3.2.2*
P.22, Lines 703-724: "Similarly to Figures 13(a, b, c, d), panels of Figure 15 illustrate the impact of the analysis on SSM using correlations., To that end, ASCAT SWI (i.e. no rescaling) has been used. Figure 14 (top panels) shows map of R values based on absolute values while Figure 14 (bottom panels) shows R values on anomalies (short term variability) as defined in Albergel et al., 2018a. Figure 15 (a) and (e) represents R values and anomaly R values for LDAS_HRES, respectively. As expected R values are higher than anomaly R values. Maps of differences (panels b and f) of Figure

15 suggest that after assimilation, both scores are improved rather equally. While the 4 day and 8-day forecast still show an improvement from the initial condition on R values (panels c and d of Figure 15 dominated by positive differences, analysis minus openloop), maps of anomaly R values forecast don't show any negative or positive impact (panels g and h of Figure 15)."

[Figure]

*Figure 15: Top row, (a) R values between LDAS_HRES open-loop and ASCAT SWI estimates from the Copernicus Global Land Service (CGLS) over 2017-2018 for the WEUR domain, (b) R differences between LDAS_HRES analysis (open-loop) and ASCAT SWI. (c) and (d) same as (b) between LDAS_fc4 initialised by the analysis (open-loop) and LDAS_fc8. Bottom row, same as top row for R values based on anomaly time-series.*

Discussion and conclusion sections
P.23, Lines 749-754: "For SSM, the assimilation is done after a rescaling to the model climatology (see section 2.3), which removes bias. For LAI, however it is not the case and the assimilation process removes bias in the modelled LAI (w.r.t. to the observation). This technical difference between SSM and LAI assimilation, combined with the longer memory of LAI compared to SSM, contributes to the results presented in this section"

**3.12 [The snow cover results (Fig 7-8) can be removed. It is too trivial that there would be no impact on snow cover by assimilating soil moisture or LAI. Or else, explain in detail how either variable would affect the snow cover.]**

**Response to 3.12**
Agreed, both figures have been moved to the supplementary document (Figures S1 and S2) and it has been further emphasized that there is no snow data assimilation yet. Those results are presented to highlight areas of improvements in LDAS-Monde:
P.15, Lines 487-492: "As expected, the analysis has an almost neutral impact on snow as both SSM and LAI observations are filtered out from frozen/snow condition and as there is no snow data assimilation in LDAS_ERA5 (Figure S2 and panels (j), (k) and (l) of Figure S1). This clearly shows however an area of potential improvement of data assimilation within LDAS-Monde using satellite data such as the IMS one (as in e.g. de Rosnay et al., 2014)."

**3.13 [The independent validation (e.g against in situ SSM) shows no substantial improvement in any of the metrics due to data assimilation. Have the in situ data been thoroughly filtered to remove bad points? Why exactly do the authors see an advantage of LDAS_ERA5 for these variables relative to the open loop (L. 458)? There is some added value, but there is also significant degradation, i.e. I would say it is an equal game here.]**

**Response to 3.13**

Agreed, last paragraph of section 3.1.2 on ground based dataset has been modified and is now (P.18, Lines 582-587): "For evapotranspiration, river discharge and surface soil moisture, there is a slight advantage for LDAS_ERA5 analysis with respect to its open-loop counterpart. Even if the distribution of the averaged statistical metrics can be rather similar for both (particularly true for surface soil moisture evaluation), there are significant differences for some sites, which shows the added value of the analysis with respect to the openloop. Note that for fewer sites, a negative impact from the analysis can also be observed."

We have also revisited the soil moisture evaluation part of the manuscript, see response to comment 3.9.

**3.14 [L. 535 & L. 545: 'more sensitive to' is perhaps not the correct wording? Sensitivity would be quantified by something like the Jacobian. There is simply a larger update in LAI than in SSM by design, and this propagates in time differently due to the difference in memory for both variables (at this point in the paper, I am actually suspecting that LAI is assimilated with a bias, see comment above).]**

**Response to 3.14**

Agreed, "more sensitive" has been replaced by "relies more". We also agree on the larger updates allowed when assimilating LAI, and it has been stressed out by adding the following paragraph to the discussions and conclusion stection (see also response to 3.5)

P.23, Lines 749-754: "For SSM, the assimilation is done after a rescaling to the model climatology (see section 2.3), which removes bias. For LAI, however it is not the case and the assimilation process removes bias in the modelled LAI (w.r.t. the observation). This technical difference between SSM and LAI assimilation, combined with the longer memory of LAI compared to SSM, contributes to the results presented in this section."

**3.15 [Could you evaluate the impact of LAI and SSM assimilation in terms of runoff for the high-resolution simulation?]**

**Response to 3.15**
Thank you for this suggestion, we have added a figure to show the impact of the assimilation (together with the impact of the initialisation on 4-day and 8-day forecasts) on drainage and runoff over the WEUR domain.

The following paragraph and figure have been added to the revised version of the manuscript (section 3.2.2 on Case studies for assessing LDAS-Monde high resolutions (0.1° x 0.1°) experiments, P.22; Lines 713-724): "Top panels of Figure 16 illustrate the impact of the analysis on drainage monitoring and forecast over WEUR. Fig. 16 a) represents drainage from LDAS_HRES openloop varying between 0 and 1 kg.m$^{-2}$.day$^{-1}$, as seen in Fig.16 b) (drainage difference between LDAS_HRES analysis and openloop) the analysis impact is rather small, about ±3% and more pronounced in areas where the analysis has affected LAI more (see panels f), g) and h) of Figure 16). As seen on panels c) and d), there is also an impact from the initialisation in areas were the analysis was more effectively correcting LAI. Bottom panels of Figure 16 illustrate similar impact on runoff. As for drainage, this variable is affected by the analysis. Initial conditions have an impact on its forecast, also. Although we did not present a quality assessment of those two variables, our findings on river discharge analysis impact, but also those from Albergel et al., 2017, 2018a, suggest a neutral to positive impact, propagated from the analysis of SSM and LAI to river discharge through variables such as drainage and runoff."

[Figure]

*Figure 15: Top row, (a) drainage values for LDAS_HRES open-loop over 2017-2018 for the WEUR domain, (b) drainage differences between LDAS_HRES analysis and open-loop. (c), (d), same as (b) between LDAS_fc4 initialised by the analysis and LDAS_fc4 initialised by the open-loop, between LDAS_fc8 initialised by the analysis and LDAS_fc8 initialised by the open-loop. Bottom row, same as top row for runoff. Units are kg.m⁻².day⁻¹*

References:

Albergel, C., Munier, S., Leroux, D. J., Dewaele, H., Fairbairn, D., Barbu, A. L., Gelati, E., Dorigo, W., Faroux, S., Meurey, C., Le Moigne, P., Decharme, B., Mahfouf, J.-F., and Calvet, J.-C.: Sequential assimilation of satellite-derived vegetation and soil moisture products using SURFEX_v8.0: LDAS-Monde assessment over the Euro-Mediterranean area, Geosci. Model Dev., 10, 3889–3912, https://doi.org/10.5194/gmd-10-3889-2017, 2017.

Albergel, C.; Munier, S.; Bocher, A.; Bonan, B.; Zheng, Y.; Draper, C.; Leroux, D.J.; Calvet, J.-C. LDAS-Monde Sequential Assimilation of Satellite Derived Observations Applied to the Contiguous US: An ERA5 Driven Reanalysis of the Land Surface Variables. Remote Sens., 10, 1627, 2018a

---

## Author Comment (AC4) · 22 Jan 2020

Dear Reviewer 4 many thanks for reviewing the manuscript and for highlighting its relevance and interest. Your comments and suggestions led to an improved version of the manuscript. Attached is a point by point answer to your specific comments, all your editorial and technical comments were accounted for in the revised version of the manuscript.

Response to Reviewer 4 are structured as follow: (1) 4.X: comments from Reviewer 4, (2) Response to 4.X: author's response and author's changes in manuscript when any. For sake of clarity, line and page numbering from the first submission is used.

[Figure]

Please also note the supplement to this comment: https://www.hydrol-earth-syst-sci-discuss.net/hess-2019-534/hess-2019-534-AC4-supplement.pdf

———————————————————————

[Figure]

**Supplement:**

Response to Reviewer 4 are structured as follow: (1) 4.X: comments from Reviewer 4, (2) Response to 4.X: author's response and author's changes in manuscript when any. For sake of clarity, line and page numbering from the first submission is used.

**[...]I think the paper can be an important contribution and can eventually be suitable for publication in HESS, but at this point I recommend MAJOR revisions with consideration of the comments below. [...]**

Dear Reviewer#4 many thanks for reviewing the manuscript and for highlighting its relevance and interest. Your comments and suggestions led to an improved version of the manuscript. Below is a point by point answer to your specific comments, all your editorial and technical comments were accounted for in the revised version of the manuscript.

**Major**
**4.1 [1] Six of the thirteen figures that show results (i.e., not counting "data & methods" figures 1 and 2) are about evaluating the skill of the assimilation estimates \*exclusively\* against the assimilated observations (Figs 3 and 9-13). Comparisons against the assimilated observations are also included in Figs 4-6 (along with other variables) and Figs 14-15 (along with forecast estimates of SSM and LAI). While I agree that it is important to verify that the assimilation system works as intended, the authors overemphasize the comparison against the assimilated observations.]**

**Response to 4.1**

We agree with Reviewer #4, verifying that the assimilation system works as intended is an important task. Part of the figures mentioned are indeed dedicated entirely (Fig. 3) or partially (Figs. 4-6) to that validation. The other aforementioned figures play a different role. Fig. 9 allows us to identify potential hotspots for droughts and heat waves. Figs. 10-11 study the behaviour of LDAS-ERA5 in the context of droughts for the WEUR (Western Europe) and the MUDA (Murray-Darling) areas. Figs. 12-15 focus on the capacity of our system to forecast the evolution of land surface variables depending on how it is initialized.

Comment 4.6 on using SSM and LAI as an independent source of information to evaluate the forecast has been further discussed and added in the revised version of the manuscript. While LAI remains an independent source of information (although constrained by the assimilation as explained in Rewiewer#4 4.6), ASCAT SWI has been rescaled to match the model climatology. The seasonal rescaling impacts both bias and correlation. In an attempt to have a more independent evaluation an additional figure has been put in the revised version of the manuscript. It presents maps of correlations, between soil moisture (1-4 cm) from the four experiments (LDAS-HRES openloop, analysis, LDAS_fc4 and LDAS_fc8) and ASCAT SWI (i.e. ASCAT data prior rescaling) for the WEUR domain. Correlations are applied to both absolute values and to anomalies (to assess the short term variability of soil moisture).

*End of section 3.2.2*
P.22, Lines 703-724: "Similarly to Figures 13(a, b, c, d) panels of Figure 15 illustrates the impact of the analysis on SSM using correlations. This time, ASCAT SWI (i.e. no rescaling) has been used. Figure 15 (top panels) shows map of R values based on absolute values while Figure 15 (bottom panels) shows R values on anomalies (short term variability) as defined in Albergel et al. (2018a). Figure 15 (a) and (e) represents R values and anomaly R values for LDAS_HRES, respectively. As expected R values are higher than anomaly R values. Maps of differences (panels b and f) of Figure 15 suggest that after assimilation, both scores are improved rather equally. While the 4 day and 8-day forecast still show an improvement from the initial condition on R values (panels c and d of

Figure 15 dominated by positive differences, analysis-openloop), maps of anomaly R values forecast do not display any negative or positive impact (panels g and h of Figure 15)."

[Figure]

*Figure 14: Top row, (a) R values between LDAS_HRES open-loop and ASCAT SWI estimates from the Copernicus Global Land Service (CGLS) over 2017-2018 for the WEUR domain, (b) R differences between LDAS_HRES analysis (open-loop) and ASCAT SWI. (c) and (d) same as (b) between LDAS_fc4 initialised by the analysis (open-loop) and LDAS_fc8. Bottom row, same as top row for R values based on anomaly time-series.*

Discussion and conclusion sections
P.23, Lines 749-754: "For SSM, the assimilation is done after a rescaling to the model climatology (see section 2.3), which removes bias. For LAI, however, this is not the case and the assimilation process removes bias in the modelled LAI (w.r.t. the observation). This technical difference between SSM and LAI assimilation, combined with the longer memory of LAI compared to SSM, contributes to the results presented in this section"

**4.2 [2) The two figures about snow (Figs 7 & 8) could be simplified considerably because there is no meaningful difference between the assimilation estimates and the open-loop estimates, which is a rather trivial result (as the authors discuss).]**

**Response to 4.2**

Agreed, both figures have been moved to the supplementary document (Figures S1 and S2) and it has been further emphasized that there is no snow data assimilation yet. Those results are presented to highlight areas of improvements in LDAS-Monde:
P.15, Lines 487-492: "As expected, the analysis has an almost neutral impact on snow as both SSM and LAI observations are filtered out from frozen/snow condition and as there is no snow data assimilation in LDAS_ERA5 (Figure S2 and panels (j), (k) and (l) of Figure S1). This clearly shows, however, an area of potential improvement of data assimilation within LDAS-Monde using satellite data such as the IMS one (as in e.g. de Rosnay et al., 2014)."

**4.3 [3) There are no graphics in the main text (only in the supplement) about the validation of the results against *independent* in situ measurements (section 3.1.2). This independent validation should be reflected more prominently in the main paper.]**

**Response to 4.3**

Most of the in situ evaluation datasets involved in this study are available over North America and (western) Europe and two regional-scale studies assessing LDAS-Monde analysis impact have already been published (Albergel et al., 2017 over Europe, Albergel et al., 2018a over North America). To avoid redundancy with these previous studies, we preferred not to put too much of

those results in the main part of the study. However to better reflect the findings of this evaluation, last paragraph of section 3.1.2 on ground based dataset has been modified and is now (P.18, L.583-587): "For evapotranspiration, river discharge and surface soil moisture it can be stated that there is a slight advantage from LDAS_ERA5 analysis with respect to its open-loop counterpart. Even if the distribution of the averaged statistical metrics can be rather similar for both (particularly true for surface soil moisture evaluation), there are regional significant differences for some sites, which shows the added value of the analysis with respect to the open-loop. Note that for fewer sites, a negative impact from the analysis can also be observed."

Also, the whole evaluation against in situ measurements has been revisited and now includes such a figure, see response to comment 4.4.

**4.4 [4] The claim about "improvement" of the assimilation estimates vs. the open-loop estimates from the independent validation against in situ soil moisture estimates in section 3.1.2 (~line 460) is on shaky footing. For none of the networks listed in Table S3 is there a difference of more then 0.02 in the R values between the assimilation and the open loop. In some cases, the 0.02 difference is negative (ie., degradation). For most networks the R difference is 0 or 0.01, that is, there really isn't a meaningful change. Here, and also for at least the other in-situ based results, it is imperative that the authors provide some estimates of whether the differences are meaningful (e.g., by including statistical confidence intervals), and then honestly discuss the results. The claim in line 460 about significant improvements at some sites may be true, but given the network-average neutral results there must then also be sites with a significant degradation, which is not mentioned in the paper.]**

**Response to 4.4**

Thank your for your highly relevant comment. Following it and similar comments from the other Reviewers, it has been decided to revisit the soil moisture evaluation part of the study:
(1) we have added an evaluation of soil moisture from LDAS-Monde fourth layer of soil (10 to 20 cm) against in situ measurements of soil moisture at 20 cm depth when available (10 networks and 685 stations),
(2) for surface soil moisture (SSM), correlation values (R) were calculated for both absolute and anomaly time-series in order to remove the strong impact from the SSM seasonal cycle on this specific metric,
(3) a 95% Confidence Interval (CI) has been added to R values.
(4) we have added the number of stations for which correlations differences are significant (significant improvement or degradation from the analysis) as well as a map over North America for illustration.

It involves several changes in the revised version of the manuscript, they are listed below.

Methodology section, 2.5 Evaluation datasets and metrics

P.11, Lines 358-365: "In situ measurements of surface soil moisture from 19 networks across 14 countries available from the ISMN are also used to evaluate the performance of the soil moisture analysis. They represent 782 stations with at least 2 years of daily data over 2010-2018. Sensors at 5 cm depth (SSM) are compared with soil moisture from LDAS_ERA5 third layer of soil (4-10 cm), sensors at 20 cm depth with the fourth layer of soil (10-20 cm, 685 stations from 10 networks). Beside 11 stations located in 4 countries of Western Africa (Benin, Mali, Sénégal and Niger) and 21 stations in Australia, most stations are located in North America and Europe, see Table S3."

 "For global estimates, Normalized RMSD (NRMSD, Eq.(2)) was used, also. Finally, for surface soil moisture, R was calculated for both absolute and anomaly time-series in order to remove the strong impact from the SSM seasonal cycle on this specific metric (see e.g. Albergel et al., 2018a, 2018b)."

Result section, 3..1.2 Ground-based datasets

 "The statistical scores for soil moisture from LDAS_ERA5 open-loop and analysis (third and fourth layers of soil, 4-10 cm depth, 10-20 cm depth, respectively) over 2010-2018 when compared with ground measurements from the ISMN (5 cm depth and 20 cm depth) are presented in Table S2 for each individual network. Averaged statistical metrics (ubRMSD, R,$R_{anomaly}$ and bias) are similar for both LDAS_ERA5 analysis and open-loop even if local differences exist. For the analysis, averaged R ($R_{anomaly}$) values along with its 95% Confidence Interval (CI) using in situ measurements at 5 cm (782 stations from 19 networks) are 0.68±0.03 (0.53±0.04) (0.67±0.03(0.53±0.04) for the open-loop) with averaged-network values going up to 0.88±0.01 (0.58±0.04) for the analysis (SOILSCAPE network, 49 stations in the USA) and always higher than 0.55 except for one network, ARM (10 stations in the USA) presenting an averaged R value of 0.29±0.05. Averaged ubRMSD and bias (LDAS_ERA5 minus in situ) are 0.060 $m^3m^{-3}$ and 0.077 $m^3m^{-3}$ for the analysis, 0.060 $m^3m^{-3}$ and 0.076 $m^3m^{-3}$ for the open-loop, respectively. NIC (Eq.1) has also been applied to R values, 65% of the pool of stations present a neutral impact from the analysis (511 stations at NIC ranging between -3 and +3), 12% present a negative impact (91 stations at NIC < -3) and 23% present a positive impact at (180 stations at NIC > +3).
The number of stations where R differences between the analysis and the openloop are significant (i.e. their 95% CI are not overlapping) is 186 out of 782 (about 26%). There is an improvement from the analysis w.r.t. the openloop for 128 stations (out of 186, i.e. about 69%) and a degradation for 58 stations (about 31%). Figure 7 illustrates R differences between the analysis and the openloop runs. When differences (analysis minus openloop) are not significant stations are represented by a small dot. When they are significant, large circles have been used, blue for positive differences (an improvement from the analsysis) and red for negative differences (a degradation from the analysis). For most of the stations where a significant difference is obtained, it represent an improvement from the analysis.
Averaged analysis R (95%CI), bias and ubRMSD for the fourth layer of soil (685 stations from 10 networks) are 0.65±0.03, 0.049 $m^3m^{-3}$ and 0.055 $m^3m^{-3}$, respectively. For the open-loop, they are 0.064±0.03, 0.048 $m^3m^{-3}$ and 0.056 $m^3m^{-3}$, respectively. For soil moisture at that depth, about 60% of the stations present a neutral impact from the analysis (410 stations at NIC ranging between -3 and +3), 28% a positive impact (189 stations at NIC > +3) and 12% a negative impact (86 stations at NIC < -3). Although differences between the openloop run and the analysis are rather small, these results underline the added value of the analysis with respect to the model run. Figure S6 represents the distribution of the scores values for LDAS_ERA5 open-loop and analysis using boxplots centred on the median value. They look very similar and from this figure, it is difficult to see either improvement or degradation from the analysis."

*Figure 7: Map of correlations (R) differences (analysis minus openloop) for stations available over North America. Small dots represent stations where R differences are not significant (i.e. 95% confidence intervals are overlapping), large circles where differences are significant.*

[Figure]

*Figure S6: a) Boxplots representing the distribution of the correlation values on absolute time-series and anomaly time-series ("Ano") between the stations with in situ measurements of soil moisture either 5cm depth or 20 cm depth and soil moisture from LDAS_ERA5 openloop and analysis over 2010-2018 (third and forth layer of soil, respectively). Correlation values are presented for surface soil moisture (5 cm depth measurements against third layer of soil), only. Distribution are centred on the median values. b) Distribution of the Bias values between the stations with in situ measurements of soil moisture either 5cm depth or 20 cm depth and soil moisture from LDAS_ERA5 openloop and analysis over 2010-2018 (third and forth layer of soil, respectively).c) Same as b) for ubRMSD.*

The following text has been added to the revised version of the manuscript: "Figure S6 represents the distribution of the scores values for LDAS_ERA5 open-loop and analysis using boxplots centred on the median value. They look very similar and from this figure, it is difficult to see either improvement or degradation from the analysis."

**4.5 [5) The editing of the paper is rather careless. There are many small mistakes, and the organization of the text is lacking.]**

The 4 Reviewers have provided many editorial comments, corrected several mistakes. Thanks to their work we have an improved version of the manuscript.

**4.5a [a) The Introduction lacks a clear statement of the paper's objectives. The text in Lines 107-121 simply states what will be presented (with lots of references and details). It's hard to tell what the objectives might be.]**

**Response to 4.5a**

Agreed. In order to make the paper's objectives clearer, the following paragraph in the introduction has been revisited:

"In this study, stemming from previous works referenced above, this global, offline, joint integration of Surface Soil Moisture (SSM) and Leaf Area Index (LAI) EOs into the ISBA (Interaction between Soil Biosphere and Atmosphere) LSM (Noilhan and Planton, 1989, Noilhan and Mahfouf, 1996) are presented: [...]"
is now (P.4, Lines 108-114):
"In this study, stemming from previous works referenced above, it is shown that LDAS-Monde global, offline, joint integration of Surface Soil Moisture (SSM) and Leaf Area Index (LAI) EOs into the ISBA (Interaction between Soil Biosphere and Atmosphere) LSM (Noilhan and Planton, 1989, Noilhan and Mahfouf, 1996) can be used to detect, monitor and forecast the impact on extreme events on LSVs. Are presented in this study: [...]"

**4.5b [b) There are several instances in the Results section of text that belongs in the Methods section, incl: Lines 384-387 - IMS snow cover product description Lines 405-409 -Fluxnet description Lines 440-447 - ISMN description]**

**Response to 4.5b**

Agreed. When appropriate, those instances were moved to the section dedicated to methodology (description of IMS data; ISMN and FLUXNET-2015 networks, river discharge).

**Response to 4.5c [c) Section 3.2.2 is a \*single\* paragraph that stretches over nearly two pages. Really? There are several other paragraphs of excessive length.]**

**Response to 4.5c**

Section 3.2.2 has now been reshuffled with one paragraph per group of 2 figures.

**d) Graphics:**

**4.5d_f1 [Figure 1a:  Use different color for zero values and no-data value.  (currently, both are white, making it unclear whether there are data in, e.g., the western US, or whether those are screened, perhaps because of topography.]**

**Response to 4.5d_f1**

Agreed, see new figure below.

[Figure]

**4.5d_f2 [Figure 3: The label of the colorbar should read "RMSD of LAI [m2 m-2]", not just "LAI [m2 m-2]"]**

**Response to 4.5f_f2**
Agreed, see new figure below

[Figure]

**4.5d_f5 [Figure 5: Units are missing for RMSD panels. (This is particularly important because this information is needed to judge whether the differences are in fact meaningful.)]**

**Response to 4.5d_f5**
Thank you for this suggestion, for RMSD panels it has been decided to use normalized RMSD (% of improvement and/or degradation) so one can really see the impact on each evaluated variable, it also echoes Reviewer 4's comment 4.7 on analysis impact on GPP. Using similar x-axis limits provides a better information at a glance. For instance it minimizes the previous visual impact of the analysis on GPP, and as such addressing your comment 4.7. Also panels of new Figure 5 separate the assimilated and independent variables, see new figure below.

Also, in section 3.1.1 on gridded dataset:the following sentence "For SSM a noticeable improvement in both correlation and RMSD is found around 20°N corresponding mainly to an improvement in the Sahara desert (not shown). GPP is also improved across almost all latitude with a particularly positive impact below 20°N which is also true for EVAP. This variable is less impacted by the analysis and some parts of the world show a decrease in e.g. RMSD values." is now (P.14, Lines 436-441):
"For SSM a noticeable improvement in both correlation and RMSD is found around 20°N corresponding mainly to an improvement in the Sahara desert (not shown). Being linked to LAI, GPP is also improved across almost all latitudes (to a lesser extend than LAI) with a particularly positive impact below 20°N. As seen on Figure 5 d) and i), there is little impact on variable EVAP which can be considered negligible. It highlights the difficulty of land surface data assimilation to impact model fluxes by modifying model states."

[Figure]

**4.5d_f6 [Figure 6: Three panels only have a single tick & tick label on the y-axis. At least two are required to interpret the axis scale.]**

**Response to 4.5d_f6**

Agreed, it has been added in the revised version of the manuscript.

**4.5d_f7 [Figure 7: The color choices should be made consistent with Fig 4.]**

**Response to 4.5d_f7**

Agreed, Figure 7 is now in the supplementary.

**4.5d_f9 [Figure 9: I could not find out what the thin cyan lines depict.]**

**Response to 4.5d_f9**

The following sentence has been added to the caption of the considered figure's caption: "Solid red line, dashed red line and solid green line represent regions MUDA, WEUR and EAFR. Solid cyan lines represent all other boxes (see Table 1 and Figure 2)."

**4.5d_f10+11 [Figures 10+11: add "LAI" to plot title of c) and d); add "SSM" to plot title of g) and h)]**

**Response to 4.5d_f10+11**

Agreed, it has been added in the revised version of the manuscript.

**4.5d_fS2 [Figure S2: NSE should vary from -infinity to 1. The colorbar is from -20 to 20, and darker blue values would clearly be greater than 1. Either the colorbar is wrong or the values show something other than NSE.]**

**Response to 4.5d_fS2**

Thanks for spotting this issue resulting from a wrong call in a python script, it has been corrected in the revised version of the manuscript.

**4.5d_ts3 [Table S3: The column headings on the 2nd page of the table still include French words.]**

**Response to 4.5d_ts3**

Corrected, thanks for spotting this issue.

**4.6 [6) In section 3.2.2, the authors no longer make it clear that the verification is against the assimilated datasets. While verification of forecast data against the assimilated dataset can be viewed as independent validation because the verification data have not (yet) been assimilated, there is an important distinction here between SSM and LAI. For SSM, the assimilation is done after rescaling (cdf-matching), which removes bias. For LAI, however, the assimilation uses the raw LAI observations (I think). That is, the assimilation removes bias in the modeled LAI (w.r.t. the observed LAI). This technical difference between SSM and LAI assimilation, combined with the longer memory of LAI compared to SSM, should contribute to the results in section 3.1.2. Put differently, the LAI results of section 3.1.2 are not likely to hold if an independent LAI dataset had been used for validation that is itself biased against the assimilated LAI observations. (Different LAI datasets may not be as biased against each other as typical satellite SSM datasets, but there are considerable biases between LAI products.)]**

**Response to 4.6**

Verifying that the assimilation system works as intended is an important task. This is why several figures have been included for "sanity check". We have emphasized in the manuscript that several presented evaluations are carried out to check if the assimilation system is working properly.

Also, using SSM and LAI as an independent source of information to evaluate the forecast has been further discussed and added in the revised version of the manuscript. While LAI remains an independent source of information for the forecast evaluation (although constrained by the assimilation), ASCAT SWI has been rescaled to match the model climatology. The seasonal rescaling impacts both bias and correlation. In an attempt to have a more independent evaluation, an additional figure has been put in the revised version of the manuscript. It displays maps of correlations between modelled soil moisture (1-4 cm) from the four experiments (LDAS-HRES openloop, analysis, LDAS_fc4 and LDAS_fc8) and ASCAT SWI (i.e. ASCAT data prior rescaling) for the WEUR domain. Correlations are applied to both absolute values and to anomalies (to assess the short term variability of soil moisture).

*End of section 3.2.2*

P.22, Lines 703-724: "Similarly to Figures 13(a, b, c, d), panels of Figure 15 illustrate the impact of the analysis on SSM using correlations., To that end, ASCAT SWI (i.e. no rescaling) has been used. Figure 14 (top panels) shows map of R values based on absolute values while Figure 14 (bottom panels) shows R values on anomalies (short term variability) as defined in Albergel et al., 2018a. Figure 15 (a) and (e) represents R values and anomaly R values for LDAS_HRES, respectively. As expected R values are higher than anomaly R values. Maps of differences (panels b and f) of Figure 15 suggest that after assimilation, both scores are improved rather equally. While the 4 day and 8-day forecast still show an improvement from the initial condition on R values (panels c and d of Figure 15 dominated by positive differences, analysis minus openloop), maps of anomaly R values forecast don't show any negative or positive impact (panels g and h of Figure 15)."

[Figure]

*Figure 15: Top row, (a) R values between LDAS_HRES open-loop and ASCAT SWI estimates from the Copernicus Global Land Service (CGLS) over 2017-2018 for the WEUR domain, (b) R differences between LDAS_HRES analysis (open-loop) and ASCAT SWI. (c) and (d) same as (b) between LDAS_fc4 initialised by the analysis (open-loop) and LDAS_fc8. Bottom row, same as top row for R values based on anomaly time-series.*

Discussion and conclusion sections

P.23, Lines 749-754: "For SSM, the assimilation is done after a rescaling to the model climatology (see section 2.3), which removes bias. For LAI, however it is not the case and the assimilation process removes bias in the modelled LAI (w.r.t. to the observation). This technical difference between SSM and LAI assimilation, combined with the longer memory of LAI compared to SSM, contributes to the results presented in this section"

**4.7 [7) Figure 3c suggests that the change in GPP is negligible, at least in the zonal mean sense although Figure 4f suggests that GPP does change in terms of RMSD. Given the considerable change in the (zonal mean) LAI (Fig 3a), I would have expected a lot more change in the mean GPP. I suspect that the disconnect between the LAI and GPP changes is rooted in how these variables are connected in ISBA and how exactly the assimilation system goes about updating LAI. This rather counter-intuitive result requires clarification in the paper.]**

**Response to 4.7**

We believe that Figure 5 was rather confusing and that the new Figure proposed (see Response 4.5d_f5, also) permits to clarify this point. In section 2.1.1 on ISBA land surface model, the following sentence is now "In the $CO_2$-responsive versions of ISBA, photosynthesis is in control of the evolution of vegetation variables." is now (P.5, Lines 157-160) "In the $CO_2$-responsive versions of ISBA, ISBA-A-gs, the model can simulate the $CO_2$ net assimilation and GPP by considering the functional relationship between the photosynthesis rate (A) and the stomatal aperture (gs) based on the biochemical A-gs model proposed by Jacob et al. (1996). Photosynthesis is in control of the evolution of vegetation variables."

References:
Jacobs, C.M.J.; van den Hurk, B.J.J.M.; de Bruin, H.A.R. Stomatal behaviour and photosynthetic rate of unstressed grapevines in semi-arid conditions. Agric. For. Meteorol. 80, 111–134, 1996.

**4.8 [8] Fig 5h:  The changes in EVAP are with +/- 0.02 (mm/d???).  If my guess about the units is correct, this would amount to only a few mm per year, which is well within the uncertainty of in situ measurements. That is, the EVAP changes are not likely to be meaningful in a practical sense. This should be discussed more explicitly.]**

**Response to 4.8**

Agreed, the new figure 5 also helps to clarify that the impact on variable EVAP is rather negligible. See also Responses to 4.5d_f5, 4.13

**Minor**
**4.9 [9] Line 167: typo "bale" –> "able"]**

**Response to 4.9**
Typo corrected in the revised version of the manuscript, thanks.

**4.10 [10] Line 209: "fifth generation of European reanalyses produced by ECMWF" I recommend phrasing this differently to avoid the misunderstanding that the reanalyses are just for the European domain. E.g.,: "fifth generation of global reanalyses produced by ECWMF"]**

**Response to 4.10**
Rephrased in the revised version of the manuscript, thanks.

**4.11 [11] Lines 293-295:  How did you address the heterogeneity within the 0.25-deg grid cells during spin-up? It is not obvious that the short spin-up period from April 2016 suffices for properly spinning up grid cells with strong heterogeneity at the sub-0.25-degree scale.]**

**Response to 4.11**

The global LDAS-ERA5 runs were spun-up by running 20 times the first year (2010). For LDAS_HRES, nine months can be perceived as a too short period to spin up the system. Unfortunately, HRES atmospheric forcing is only available from April 2016 and the LDAS-HRES experiment ends in December 2018. We have considered this 9 months period for the spin up in order to have the longest possible time series for land surface variables, thus giving more strength to statistics. We could have considered a longer period for spin up (April 2016 to December 2017) and studied only 2018. This gives very similar results on surface soil moisture and LAI (not shown). While not being fully spun-up, results obtained with LDAS-HRES can be considered as representative of the system response to data assimilation. Note that most initial values of the LDAS-HRES run are taken from the ECOCLIMAP-II database. For instance, initial LAI is set from a 1999-2005 MODIS climatology.

Another possibility to initialise LDAS-HRES could have been to downscale the state of LDAS-ERA5 run in April 2016 to 0.10°x0.10° spatial resolution. LDAS-ERA5 runs have been set to an equilibrium spinning up 20 times the first year (2010).

The following sentence: "The period 2017-2018 is presented, HRES is available at this spatial resolution from April 2016, only, and the time period from April to December 2016 is used as a

short spinup." has been modified and is now (P.10, Lines 327-332): "HRES is available at a 0.1° x 0.1° resolution only from April 2016. April to December 2016 is used as a short period for spinup and results are presented for the period 2017-2018. Although a 9-month spinup period can be seen as rather short, evaluating LDAS-HRES on either 2017-2018 or 2018 (using instead a 21-month spinup) leads to similar results on surface soil moisture and LAI (not shown). While the system is not fully spun-up, it can be considered as representative of the system response to data assimilation."

**4.12 [12) Line 379: Do you mean a decrease in RMSD or a decrease in skill?]**

**Response to 4.12**
This sentence has been revised and "[...] shows a degradation" is now "[...] shows a decrease in skill"

**13) Line 412: If I'm reading this correctly RMSD decreases while both bias and ubRMSD increase. This is quite counter-intuitive and requires a rather odd distribution of the metrics across the sites or networks included in the average. In any case, since bias and ubRMSD get worse, I do not think that the statement about "a small advantage of the analysis over the open-loop" is justified.**

**Response to 4.13**
Agreed, the considered sentence has been reformulated and is now: "If these numbers depict a small advantage of the analysis over the open-loop configuration, it is worth mentioning that differences are rather small and likely to fall within the uncertainty of the in situ measurements."

**14) Line 429: "NSE values below -2 were discarded" requires a justification, otherwise it reads like cherry-picking.**

**Response to 4.14**
Agreed, this threshold has also been used for previous studies at CNRM as we did not want to look at river discharges we do not represent well. The pool of stations we have used are monitoring all types of rivers and streams including those where human impacts (dams and reservoirs, irrigation, water uptake, not represented in ISBA yet) is affecting the natural flow of rivers. As we expect the impact of the analysis on river discharge to be small (based on previous work), we did not find necessary to include stations we badly represent in ISBA, possibly for known reasons. Futur work will focus on preparing a more robust in situ pool of station, separating e.g. managed and unmanaged rivers and stream.

The following paragraph has been added to the methodology section (P.12-13, Lines 394-399): "Stations with NSE values lesser that -2 were discarded. A similar threshold has already been used in previous studies evaluating LDAS-Monde (e.g. Albergel et al., 2017, 2018a). Many processes, most of them linked to water management such as the presence of dams and reservoirs, irrigation, water uptake in urban areas, are not yet represented in ISBA leading to a poor representation of river discharges. As previous evaluations studies have suggested a neutral to positive impact from the assimilation, only, it has been decided to focus on stations with reasonable NSE values."

**4.15 [15) Line 535: "the analysis is of better quality" Given the numbers, I see at best "slightly better quality"]**

**Response to 4.15**
Emphasized in the revised version of the manuscript, "Note however that for the MUDA area, a 4-d forecast of surface soil moisture initialised by the analysis is of better quality than a 4-d forecast

initialised by the open-loop" is now (P.21, Lines 664-666): "Note, however, that, for the MUDA area, there is a small positive impact of the initialisation on the 4-d and 8-d forecast of surface soil moisture (blue areas on Figure 13 c) and d))."

**4.16 [4.16 [16) Line 592:  "surface (0-1 cm)" In section 3.2.2 the discussion was about the "(1-4cm)" layer. Which is it?]**

**Response to 4.16**
Thanks, it should read 1-4cm, it is now corrected in the revised version of the manuscript

---

## Author Response (AR2)

**Responses to Reviewer are structured as follow: (1) 1.X: comments from Reviewer, (2) Response to 1.X: author's response and author's changes in manuscript when any. For sake of clarity, line and page numbering from the submitted version is used when appropriate.**

We thank the Reviewer for her/his second review of our paper. We acknowledge that, while we positively answered to most points raised by the Reviewer in her/his first review, some others still required further work. Comments raised by the Reviewer in this second review can be summarized in three key points as follows:

A) *"[…] de-emphasizing the "validation" of the LDAS-Monde output against the assimilated observations and making it clear throughout the paper when "validation" is against the assimilated obs."*
    B) *"[…] emphasize the validation against independent (in situ) measurements by moving the relevant graphics from the Supplement to the main text."*
C) *"[…] lack of care in the editing of English style and grammar […]"*

    Please find below our answers to the Reviewer's comment. They should complement the work previously achieved during the first review of our manuscript.

**This is my second review of the paper, after the reviewers revised the originally submitted manuscript. The authors fixed a number of obvious errors and confusing statements, including my original "minor" comments (#9-16) as well as my major comments #2 (re. Snow variables) and #8 (re. Evaporation).**

**1.A) [A1 My original major comments #1 and # suggested de-emphasizing the "validation of the LDAS-Monde output against the assimilated observations and making it clear throughout the paper when "validation" is against the assimilated obs.**

    **A2 In response, the authors \*added\* more "validation" results against the assimilated**
**observations. The new figure 15 shows correlations against ASCAT SWI prior to rescaling. Simply omitting the rescaling of the ASCAT SWI does \*not\* make the ASCAT SWI data independent of the assimilated observations.]**

    Response to 1.A1
    We have now made clear in the paper that the evaluation of LDAS-Monde is performed using either the assimilated observations or independent datasets. We have also de-emphasized the "validation" against the assimilated observations through the text and clarified the two main objectives of paper. These improvements has led to several changes in the manuscript:
    • the abstract has been rewritten,
    • the structure of the paper has been modified,
    • and some figures have been either partially deleted or merged.

Below are presented all the modifications related to Reviewers' comment 1.A.

    1.A1.1 The new abstract is (2 main objectives in bold):

    "This study demonstrates that LDAS-Monde, a global and offline Land Data Assimilation System
(LDAS), that integrates satellite Earth Observations into the ISBA (Interaction between Soil

Biosphere and Atmosphere) Land Surface Model (LSM), is able to detect, monitor and forecast the impact of extreme weather on land surface states. LDAS-Monde jointly assimilates satellite derived Earth observations of Surface Soil Moisture (SSM) and Leaf Area Index (LAI). First, LDAS-Monde is run at a global scale forced by the latest atmospheric reanalysis from the European Centre for Medium Range Weather Forecast (ECMWF), ERA5 (ECMWF fifth global reanalysis, LDAS_ERA5 hereafter) over 2010-2018, leading to a 9-yr, ~0.25° x 0.25° spatial resolution reanalysis of Land Surface Variables (LSVs). **The quality of this global analysis is evaluated using several satellite-based datasets: assimilated SSM and LAI, but also independent datasets of evapotranspiration, Gross Primary Production, Sun Induced Fluorescence and snow cover. In addition, in situ measurements of SSM, evapotranspiration and river discharge are also employed for the evaluation. This assessment is conducted by comparing LDAS-Monde analysis with a model simulation (open-loop, no assimilation). Secondly, the global analysis is used to (i) detect regions exposed to extreme weather such as droughts and heatwave events and (ii) address specific monitoring and forecasting requirements of LSVs for those regions.** This is performed by computing anomalies of the land surface states. They display strong negative values for LAI and SSM in 2018 for two regions experiencing severe heatwave and/ or droughts: North Western Europe and the Murray-Darling basin in South Eastern Australia. For those two regions, monitoring and forecasting LSVs under extreme conditions are examined by forcing LDAS-Monde with ECMWF Integrated Forecasting System (IFS) high resolution operational analysis (LDAS_HRES, ~0.10° x 0.10° spatial resolution) over 2017-2018. Monitoring capacities are studied by comparing open-loop and analysis experiments again against the assimilated observations. Forecasting abilities are assessed by initializing 4- and 8-day LDAS_HRES forecasts of the LSVs with the LDAS_HRES assimilation run compared to open-loop experiments. The impact of initialization in forecast mode is particularly visible for LAI that evolves at a slower pace than SSM and is more sensitive to initial conditions than to atmospheric forcing, even at an 8-day lead time. This highlights the importance of initial conditions to forecast LSVs and it confirms that LDASs should jointly analyse both soil moisture and vegetation states."

1.A1.2 The  structure of the paper:

**3 Results**
*3.1 Global assessment of LDAS_ERA5*
3.1.1 Gridded datasets
3.1.2 Ground-based datasets
*3.2 Monitoring and forecasts for areas under severe/extreme conditions*
3.2.1 Case studies for assessing LDAS-Monde medium resolutions (0.25° x 0.25°) experiments
3.2.2 Case studies for assessing LDAS-Monde high resolutions (0.1° x 0.1°) experiments has been modified as follows :

**3 Global assessment of LDAS_ERA5**
3.1 Gridded datasets
3.2 Ground-based datasets
**4 Monitoring and forecasts for areas under severe/extreme conditions**
4.1 Selection of two regional case studies
4.2 Case studies presentation: LDAS-Monde medium resolution (0.25° x 0.25°) experiments
4.3 Case studies for assessing LDAS-Monde high resolutions (0.1° x 0.1°) analysis and forecast

This new structure is detailed at the end of the introduction: "The paper is organised in five sections: section 2 details the various components constituting LDAS-Monde (the ISBA LSM, the data assimilation scheme and the EOs assimilated as well as the different atmospheric forcing datasets used), followed by the experimental and evaluation setup. Section 3 describes and discusses the impact of the analysis on the representation of the LSVs. Section 4 details the identification of 2 case studies over regions particularly affected by extreme events during 2018 and their detailed monitoring at higher spatial resolution combined with land surface forecasting activities is also presented. Finally section 5 provides conclusions and prospects for future work."

1.A1.3 The first objective of the study described at the end of the introduction (P.4, L.110-120):
"*An evaluation at global scale using diverse and complementary datasets such as evapotranspiration from the GLEAM project (Miralles et al., 2011, Martens et al., 2017), Gross Primary Production (GPP) from the FLUXCOM project (Tramontana et al., 2016, Jung et al., 2017), Solar Induced Fluorescence (SIF) from the GOME-2 (Global Ozone Monitoring Experiment-2) scanning spectrometer (Munro et al., 2006, Joiner et al., 2016) and snow cover data from the Interactive Multi-sensor Snow and Ice Mapping System (or IMS, https://www.natice.noaa.gov/ims/, last accessed June 2019). It is also validated using reference observations including in situ evapotranspiration from the FLUXNET 2015 synthesis data set (http://fluxnet.fluxdata.org/, last accessed June 2019), soil moisture from the International Soil Moisture Network (ISMN, https://ismn.geo.tuwien.ac.at/en/, last accessed June 2019) as well as river discharge from several networks across the world.*"
has been rewritten as follows  (in bold are the specific modifications to clarify  when the evaluation is performed  against the assimilated observations or independent datasets):
"An evaluation of LDAS-Monde at a global scale is carried out. **This assessment involves the assimilated observations to demonstrate that the system is working as intended. But more fundamentally, LDAS-Monde global analysis is appraised using diverse, independent and complementary** satellite-derived datasets of evapotranspiration (EVAP) from the GLEAM project (Miralles et al., 2011, Martens et al., 2017), Gross Primary Production (GPP) from the FLUXCOM project (Tramontana et al., 2016, Jung et al., 2017), Solar Induced Fluorescence (SIF) from the GOME-2 (Global Ozone Monitoring Experiment-2) scanning spectrometer (Munro et al., 2006, Joiner et al., 2016) and snow cover data from the Interactive Multi-sensor Snow and Ice Mapping System (or IMS, https://www.natice.noaa.gov/ims/, last accessed June 2019).  This evaluation is additionally performed with in situ measurements of evapotranspiration from the FLUXNET 2015 synthesis data set (http://fluxnet.fluxdata.org/, last accessed June 2019), soil moisture from the International Soil Moisture Network (ISMN, https://ismn.geo.tuwien.ac.at/en/, last accessed June 2019) and river discharge from several networks across the world."

1.A1.4 The following paragraph has been added at the beginning of section 3.1 on gridded datasets:
"In this sub-section, LDAS-Monde open-loop and analysis are first compared to the assimilated observations (SSM and LAI) to demonstrate that the assimilation system is working as intended. Both experiments are also compared to independent sources of information to evaluate the analysis impact (GPP, EVAP and SIF)."

1.A1.5 The following sentences  have been modified: "*From Figure 4a it is possible to see the positive impact of the analysis compared to the open-loop, with the former being closer to the observations. Improvement from the analysis occurs from nearly 80°North to about 55° South, areas around the equator are particularly improved.*" and are now: "From Figure 4a it is possible to see the positive impact the analysis has on LAI compared to the open-loop, with the former being closer to the observations. Improvements from the analysis occurs from nearly 80°North to about 55° South, areas around the equator are particularly improved. This demonstrates that the data assimilation system is working as intended."

1.A1.6 The following sentence has been added at the beginning of section 3.2 on ground based datasets: "LDAS_ERA5 analysis and open-loop are also evaluated using independent in situ measurements of evapotranspiration, river discharge and surface soil moisture across the world."

1.A1.7 In order to de-emphasize the evaluation against assimilated observations, panels c), d), g) and h) of Figures 9 and 10 from section 3.2.1 (now 4.2 on regional evaluation of LDAS_ERA5 openloop and analysis against assimilated observations) have been removed and the remaining panels merged into 1 figure (new figure 11 of 4.2, see below). Figure 11 description has been modified accordingly and is now: "Figure 11 illustrates seasonal cycles of observed LAI (Figure

11a) and SWI (Figure 11e), LDAS_ERA5 analysis and open-loop LAI (Figure 11b) and SSM (Figure 11f) for the WEUR domain. 2018 is compared to an average of the period 2010-2017. From Figure 11a, one may see the heatwave impact with a sharp drop in observed LAI values from June to November 2018 (solid green line). Such low LAI values have never been observed over the eight previous years (dashed green line for the 2010-2017 averaged along with the 2010-2017 minimum and maximum observations in shaded green). A similar behaviour is also visible in the ASCAT SWI dataset in Figure 11e with the lowest values ever reached in this 2010-2018 period. Over WEUR, LDAS_ERA5 open-loop overestimates LAI in the second part of the year as already highlighted by several studies (e.g. Albergel et al., 2017, 2019). LDAS_ERA5 analysis has a positive impact, reducing LAI values, as seen on Figure 11b (LAI open-loop in blue, analysis in red) Panels c), d) g)

and h) of Figure 11) depict a similar situation for the MUDA area, almost every month of 2018 presents the lowest values for both SSM and LAI. For both MUDA and WEUR, the smaller differences for LAI and SSM between LDAS_ERA5 analysis and open-loop in 2018 compared to 2010-2017 also suggest that both extreme events were well captured in the atmospheric forcing used to drive LDAS_ERA5."

[Figure]

*Figure 11 :Upper panels represent seasonal cycles of a) observed GEOV1 LAI from CGLS, b) LAI from the open-loop (in blue) and the analysis (in red) for the WEUR area (see Table I for geographical extent). c) and d) panels are similar to a) and b) for the MUDA area . Lower panels represents seasonal cycles of e) ASCAT SWI from CGLS, f) SSM from the open-loop (in blue) and the analysis (in red) for the WEUR area. Panels g) and h) are similar to e) and f) for the MUDA area. For each panels dashed line represents the averaged over 2010-2017 along with the minimum and maximum values, the solid lines are for the year 2018.*

1.A1.8 First paragraph of section 3.2.2 (now 4.3) is now: "For these two specific areas (WEUR and MUDA), LDAS-Monde is also run forced by HRES (LDAS_HRES) at 0.1° x 0.1° spatial resolution over April 2016 to December 2018. Additionally to LDAS_HRES analysis, forecast experiments with a lead time of 4-days and 8-days, initialised by either LDAS_HRES analysis or open-loop are presented for 2017-2018 (for SSM and LAI) in order to assess the impact of the initial conditions on the forecast of the LSVs. **In this subsection, this new set of six experiments is verified against the assimilated observations.**"

1.A1.9 We have added a new column to Table III presenting the evaluation datasets to make it clear either or not they are an independent source of information.

1.A.2 We agree with the Reviewer's comment that the rescaling of the ASCAT SWI product does not make the ASCAT SWI data independent of the assimilated observations, we have now discussed that point in the first paragraph of section 3.2.2 (now 4.3):"Verification of the forecast experiments can be viewed as an independent validation as those observations are not assimilated yet. It is worth mentioning that there is a difference between the use of SSM and LAI observations to evaluate the forecast. For SSM, the assimilation is done after a rescaling to the model climatology (see section 2.3), which removes bias. For LAI, however this is not the case and the assimilation process unbiases the modelled LAI (w.r.t. the observation). This difference, together with the longer memory of LAI (compared SSM), contributes to the results presented in this subsection. Statistical scores for LDAS_HRES open-loop and analysis are presented, also, to serve as a benchmark of the forecast experiments." The diagnostic presented by figure 15 (now figure 16) on SWI anomaly remains useful and is kept in the text.

**1.B [B1) My original major comment #3 asked to emphasize the validation against independent (in situ) measurements by moving the relevant graphics from the Supplement to the main text. The authors reply that much of this has already been published in earlier papers (Response to 4.3), which is acceptable if explained and referenced properly.**

**However, this was not clear from the originally submitted manuscript, and rather than clearly referencing the earlier validation results in the revised version, the authors chose to keep the section on in situ validation results (section 3.1.2) without a single reference within this section to the earlier publications, where much of the same results have already been published, according to the authors. f that is indeed the case, then the section has to be deleted. If there are new results here (perhaps a more global set of validation sites), then there need to be graphics in the main text.**

**B2) The authors added 95% confidence intervals for correlations, results in terms of raw and anomaly R values, and metrics for the 4th soil layer (10-20cm). None of this changed the fact that the in situ validation results indicate no statistically significant improvement or degradation in soil moisture skill. The numbers in Table S3 show that this is true for each network individually, in terms of R, anomaly R, and ubRMSE. The results in Fig 7 over CONUS only do not seem consistent with the numbers quoted in the text that talk about significant improvements at 186 stations and significant degradation at 58 stations (Lines 505-510). The results in Table S3 simply do not support the language of "added value of assimilating [..] soil moisture" (e.g., Line 26).]**

Response to 1.B

The authors would like to apologize for the absence of reaction on the validation of LDAS-Monde with in situ measurements. We agree with the Reviewer that the validation step should have included in situ observations in the main document rather than in the supplement. To that end, the sub-section on evaluating LDAS-Monde with in situ measurements has been reshuffled and 2 figures from the supplementary material have been inserted and discussed in the main body of the manuscript: Figure 7 (evaluation using FLUXNET data) and figure 8 (evaluation using river discharge). Figure 9 (old figure 7) over the CONUS area was not consistent with the text as the text was discussing the whole pool of stations (not only those over CONUS). This point has now now been clarified and the numbers for this specific area have also been provided.

[revised manuscript text omitted]

**1.C) [My major comment #5 identified more than a dozen instances of careless editing. The authors mostly fixed these issues, but the manuscript still shows a considerable lack of care in the editing of English style and grammar, to the point where the text is difficult to understand at times.]**

Response to 1.C

We have performed a careful editing work through the whole manuscript and, to the best of our knowledge, we believe that we have fixed most (hopefully all) English style and grammar errors. The track-change version of the revised manuscript permits to appreciate every small change carried out in the document. Finally, i**n the case of acceptance, the final revised paper will be typeset and proofread. This should hopefully remove any potential remaining misprints.**

[revised manuscript text omitted]

---

## Author Response (AR3)

Dear Editor,

In their latest review, the two (original) reviewers insisted that language in the paper needs to be improved. Our manuscript has now been carefully scrutinised leading to several changes. We have not added any new materials but rewritten several parts of the manuscript. Please find below a marked-up version of our manuscript highlighting those changes. Reviewer#1 has also asked to revise the legend of figure 7a (cutting of the colour-bar at -20 and 20) to make it more visible, please find the new figure below.

Sincerely

Clément Albergel, on behalf of the co-author

[revised manuscript text omitted]